# Achieving water budget closure through physical hydrological processes modelling: insights from a large-sample study

Xudong Zheng[1], Dengfeng Liu[1*], Shengzhi Huang[1*], Hao Wang[2], Xianmeng Meng[3]

[1]State Key Laboratory of Eco-hydraulics in Northwest Arid Region of China, School of Water Resources and Hydropower, Xi'an University of Technology, Xi'an, 710048, China
[2]State Key Laboratory of Simulation and Regulation of Water Cycle in River Basin, China Institute of Water Resources and Research, Beijing, 100038, China
[3]School of Environmental Studies, China University of Geosciences, Wuhan, 430074, China

*Correspondence to*: Dengfeng Liu (liudf@xaut.edu.cn)

**Abstract.** Modern hydrology is embracing a data-intensive new era, information from diverse sources is currently providing support for hydrological inferences at broader scales. This results in a plethora of data reliability-related challenges that remain unsolved. The water budget non-closure is a widely reported phenomenon in hydrological and atmospheric systems. Many existing methods aim to enforce water budget closure constraints through data fusion and bias correction approaches, often neglecting the physical interconnections between water budget components. To solve this problem, this study proposes a Multisource Datasets Correction Framework grounded in Physical Hydrological Processes Modelling to enhance water budget closure, termed PHPM-MDCF. The concept of decomposing the total water budget residuals into inconsistency and omission residuals is embedded in this framework to account for different residual sources. We examined the efficiency of PHPM-MDCF and the residuals distribution across 475 CONUS basins selected by hydrological simulation reliability. The results indicate that the inconsistency residuals dominate the total water budget residuals, exhibiting highly consistent spatiotemporal patterns. This portion of residuals can be significantly reduced through PHPM-MDCF correction and achieved satisfactory efficiency. The total water budget residuals have decreased by 49% on average across all basins, with reductions exceeding 80% in certain basins. The credibility of the correction framework was further verified through noise experiments and comparisons with existing methods. In the end, we explored the potential factors influencing the distribution of residuals and found notable scale effects, along with the key role of hydro-meteorological conditions. This emphasizes the importance of carefully evaluating the water balance assumption when employing multisource datasets for hydrological inference in small and humid basins.

## 1 Introduction

Advances in measurement and monitoring techniques have revolutionized the hydrology research through providing an unprecedented opportunity to detect hydrology process (Sivapalan and Blöschl, 2017). Data availability is no longer the key constraint for conducting large-scale research as it once was. Approaches that works with large samples and multisource data

are now more attractive for hydrological studies (Nearing et al., 2021). In the absence of satisfactory in-situ observation, we can freely access data from different sources as complement, such as satellite remote sensing, radar, model simulation and reanalysis (Refsgaard et al., 2022). As such, whether at the watershed scale or the modelling scale (e.g., grid cells), we have multiple choices to represent water budget components, thereby facilitating hydrological inferences. This reality is also referred
to as the fourth paradigm of hydrology (Peters-Lidard et al., 2017).

However, every coin has two sides, the abundance of available data has brought challenges in data selection, confronting contemporary hydrologists with the task of filtering datasets. After excluding datasets that do not match the research scale and spatiotemporal coverage, we still have no idea about how to select the most suitable one from remaining datasets. In the past
decades, extensive efforts have been made to evaluate the accuracy of datasets by referencing in-situ observation or ensemble of multisource data (Sahoo et al., 2011; Tang et al., 2020; Ansari et al., 2022). However, the fact remains that the "true value" is perpetually unattainable, rendering any form of reference data uncertain. For example, the undercatch phenomenon in rainfall measurements is well known, and it is difficult to eliminate the bias even with the application of undercatch corrections (Robinson and Clark, 2020). The issue of scale mismatches and the availability of site data in certain regions also pose
challenges for data evaluation. Therefore, we argue that the evaluation based on reference data lakes sufficient reliability, highlighting the need for more widely applicable criteria in evaluating and correcting datasets from various sources.

The law of mass conservation, typically represented in hydrology by the water balance, constitutes a fundamental principle applicable universally across time and space. Thus, the terrestrial water budget describes the physical consistency among
different components of the water balance, which can serve as a criterion for evaluating and correcting datasets. For a closed basin, the water budget can be mathematically expressed as (Lehmann et al., 2022),

$$\frac{\mathrm{d}TWS}{\mathrm{d}t} = P - ET - R, \tag{1}$$

where $\frac{\mathrm{d}TWS}{\mathrm{d}t}$ is change in terrestrial water storage, $P$ is precipitation, $ET$ is evaporation, $R$ is runoff at the outlet. By incorporating data from different sources into Eq. (1), we can assess whether these data achieve closure of the water budget,
thereby evaluating their reliability in depicting hydrological processes. If Eq. (1) is not satisfied, the residual term, known as water budget residuals, can quantify the extent of physical inconsistency among multiple datasets. A comprehensive review of the terrestrial water budget closure examination is given in Lv et al. (2017), interested readers are encouraged to refer to this work. The consensus in the recent scientific literature is that data inconsistency is widespread, attributed to different production processes among various datasets, and no single combination of datasets can fully close the water budget across all basins.
Such inconsistency poses an obstacle to robust hydrological inferences (Beven, 2002). As an example of this, physically inconsistent forcing and evaluation data can mislead hydrological modelling and introduce significant uncertainty to model

inferences (Kauffeldt et al., 2013). To mitigate the impact of data inconsistency, it is essential to properly correct datasets and improve water budget closure.

The pioneering work in enhancing water budget closure across different data sources through data correction was conducted by Pan and Wood (2006), who integrated a Constrained Ensemble Kalman Filter (CEnKF) to impose constrains on terrestrial water budget. This technique was subsequently developed and applied in several studies (Sahoo et al., 2011; Zhang et al., 2016). Similar extension methods include Multiple Collocation (MCL) and Proportional Redistribution (PR) method (Abolafia-Rosenzweig et al., 2020; Abhishek et al., 2022; Luo et al., 2023). These methods are all grounded in the data fusion

process, deriving uncertainties for each water budget component from multiple data sources. Estimated uncertainties facilitate the determination of weights for allocating closure residuals, ultimately achieving a zero residual. Overall, these methods can be collectively referred to as data fusion-based closure correction approaches. Another recently developed method to constrain water balance employs an optimization-based strategy, exhibiting improved performance in long-term consistency with GRACE terrestrial water storage change (Petch et al., 2023). Other approaches, such as post-Processing Filtering technique

(PF) and bias correction method (Munier et al., 2014; Weligamage et al., 2023), can also be helpful in closing water budget. However, the closure constraints imposed by the above methods (hereafter referred to as traditional methods) have been questioned, with Abolafia-Rosenzweig et al. (2020) arguing about the potential incorrect assignment of residuals. If a component in the water budget exhibits a bias, closure correction algorithms may mistakenly apply the bias closure constraint to other components. The intrinsic attribution of this issue lies in the algorithms neglecting the physical correlations among

components and imposing strict constraints on water budget closure by integrating uncertainties from multisource data. Or in other words, assigning closure residuals exclusively based on the magnitude of priori data uncertainty, without accounting for the distribution of components in hydrological processes, such as the partitioning of precipitation, may be unrealistic and could lead to erroneous allocation of closure residuals. In the context of applying such closure constraint, it becomes evident that the precision of certain individual components may notably deteriorate, particularly when uncertainties are challenging to quantify

(Luo et al., 2023).

    As is well-known, hydrological models, whether data-driven or physics-based, aim primarily to characterize hydrological processes by accurately allocating water quantities among components such as precipitation, evaporation, runoff, and soil moisture. In abstract terms, hydrological models can be regarded as directed graphs of fluxes, with nodes representing state

variables and edges symbolizing fluxes or transitions (Wang and Gupta, 2024). Such directed graph is computationally closed, indicating that hydrological models inherently exhibit the essential characteristic of water budget closure. A clear piece of evidence comes from the data consistency evaluation conducted by Penning De Vries et al. (2021), who found that the dataset from the same model (i.e., precipitation and evaporation from ERA5 coupled model) manifested a well-closed system. In this sense, hydrological models appear capable of guiding the allocation of closure residuals to enhance water budget closure.

Another distinctive feature of hydrological models, known as error adaptability or calibration compensation capability,

underscores their pivotal role as innovative solutions for addressing challenges in achieving water budget closure. The feature emphasizes that hydrological models can, to some extent, compensate for biases in model inputs, outputs and structure, allowing satisfactory performance even when the utilized datasets exhibit certain inaccuracies (Wang et al., 2023). This provides hydrological models with the potential to integrate forcing and evaluation datasets into a unified water balance system under the soft constraint paradigm.

Here we propose another critical question regarding achieving water budget closure: Is the terrestrial water budget described by Eq. (1) fully comprehensive? This issue came to our attention through a recent study by Gordon et al. (2022), who examined the widespread validity of the Closed Water Budget (CWB) hypothesis (i.e., formulated by Eq. (1)) across 114 highland catchments using multiple data sources. Surprisingly, their results revealed that the CWB hypothesis failed to hold in 75% to 100% of the catchments. They highlighted that such failure of the CWB hypothesis could propagate widely in hydrological inferences relying on it, potentially leading to erroneous conclusions. To provide a physical explanation for the invalidity of the CWB, they extended Eq. (1) by introducing an error term $e$ and additional term $G$, as depicted in Eq. (2).

$$e + G = P - ET - R - \frac{dTWS}{dt}, \tag{2}$$

The term $G$ accounts for the inter-basin groundwater fluxes that were not considered in the original formulation, while the term $e$ addresses inconsistencies among the original datasets. Clearly, when applying the CWB hypothesis for data evaluation or correction, there is a tendency to prematurely assume the completeness of the applied formulas, potentially leading to significant biases in the final results. Furthermore, in practical application, besides groundwater, the Eq. (1) may inadvertently omit other water fluxes and storages. For instance, utilizing gravity changes observed by GRACE to estimate TWS may encompass inter-basin water transfers or irrigation, which can have substantial influence in studies conducted at relatively small scales (Lv et al., 2017). Partial observations of precipitation, evaporation and runoff can also introduce biases into this equation. To distinguish the omission from total water budget residuals among the original datasets, we further extend Eq. (2) to obtain the generalized form as follows:

$$Res = Res_i + Res_o = P - ET - R - \frac{dTWS}{dt}, \tag{3}$$

where $Res$ is the total water budget residuals; $Res_i$ is the inconsistency residuals, accounting for the fraction of water non-closure due to physical inconsistencies among the original datasets; $Res_o$ is the omission residuals, explaining the fraction resulting from omitted fluxes and storages in the original equation. We assume that Eq. (3) offers a comprehensive description of the terrestrial water budget and can be examined using multisource datasets. This advancement, compared to previous studies, breaks down the sources of water budget residuals, offering guidance for data evaluation and correction.

Given the current increase in data availability but concerns over reliability, this study aims to address the following scientific questions through physical hydrological processes modelling: (a) How can the total water budget residuals be quantitatively

decomposed into inconsistency and omission residuals based on Eq. (3)? (b) From a large-sample perspective, what are the distribution patterns of these residuals? (c) What strategies can be employed to achieve water budget closure through physical

hydrological processes modelling while strengthening the physical coherence among datasets from different sources? By addressing these questions, we highlight the necessity for a comprehensive description of the water budget equation to effectively evaluate and correct water non-closure. Furthermore, we developed a multisource datasets correction framework based on decomposition of water budget residuals and multi-objective calibration within hydrological modeling. The presented framework, providing the capability to enhance the water budget closure and hydrological connections among multisource

datasets, was applied to a large-sample basins dataset across CONUS.

The remainder of this paper is organized as follows. Sect. 2 describes the main datasets used in this research. Sect. 3 then details the methods for decomposing water budget residuals and the multisource data correction framework with a hydrological model. The results are presented and discussed in Sect. 4 and Sect. 5. Sect. 6 provides the main conclusions and outlook of

this study.

## 2 Data

### 2.1 The CAMELS dataset

Motivated by Gupta's call for large sample hydrological studies to strike a balance between depth and breadth (Gupta et al., 2014), in this study, we attempt to carry out analysis on a widely used large sample dataset, i.e., the Catchment Attributes and

Meteorology for Large-sample Studies (CAMELS) community dataset. This dataset, developed by Newman et al. (2015) and Addor et al. (2017), encompasses daily forcing, hydrologic response, and basin attributes for 671 basins across the contiguous United States (CONUS), characterized by minimal human disturbance. Drawing upon this dataset, a substantial body of experimental studies have been conducted, covering model intercomparison, analyses of hydrological scale effects, evaluations of model performance metrics, parameter estimation and exploration of machine learning models (Knoben et al., 2020; Beven,

2023). Grounded in large sample inquiries, these studies systematically explore the prevalent heterogeneity from different perspectives, yielding more robust and widely applicable conclusions.

In the original work proposed CAMELS dataset by Newman et al. (2015), a widespread physical inconsistency behaviors were observed, characterized by an imbalance between precipitation and runoff. In the spatial depiction within the Budyko

framework, certain basins exhibited plotting points exceeding the water limit line, indicating a surplus of runoff relative to precipitation. They emphasized the necessity for corrections to be applied to datasets. For the aforementioned reasons, investigation of the decomposition and reconciliation of water budget residuals on the CAMELS dataset is both necessary and feasible. In practice, the in-situ runoff data observed by USGS National Water Information System server was used. Considering the availability of data products, our analysis is conducted over a common overlapping period spanning from

1998 to 2010. During this period, eighteen basins with missing runoff observations were excluded in advance. Figure 1 presents a regional profile and the detailed information on the excluded basins is provided in supplemental Table S1.

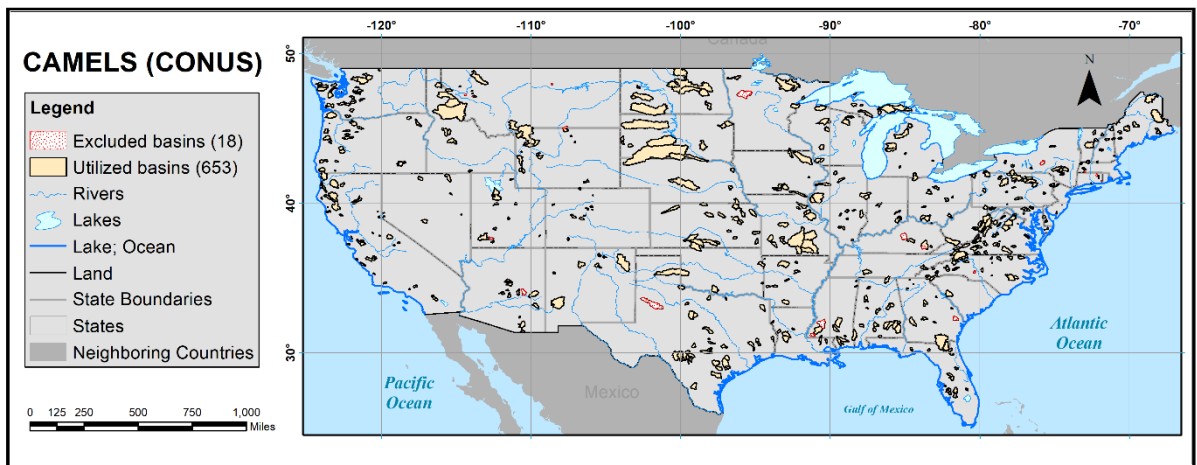

**Figure 1.** Geographic representation of the CAMELS Basins Dataset (Newman et al., 2015 and Addor et al., 2017). Eighteen basins excluded from the analysis are denoted by red dots, whereas the study incorporates the remaining 653 basins, emphasized with yellow shading. The
copyright of the background map belongs to Esri (Gray Canvas Basemap).

### 2.2 Datasets for constructing water budget equation

One of the main aims of this study is to investigate the decomposition of water budget residuals and correction to datasets, rather than comparing the differences and rankings of closure residuals across different dataset combinations. In line with this objective, referring to the work of Petch et al. (2023), we strategically selected single product for each water component to
construct water budget equation, thereby laying the foundation for further research. In making this selection, we considered not only the resolution and spatiotemporal coverage of the products but also took into account recommendations from previous data evaluation studies regarding data accuracy (Kittel et al., 2018; Lehmann et al., 2022). All datasets used are summarized in Table 1. Notably, the term "measurements" referred in this work are derived from multisource datasets and do not specifically refer to in-situ measurements.


Specifically, daily precipitation estimation derived from the Tropical Rainfall Measuring Mission (TRMM 3B42V7) is used in this study. The well-known international NASA project aims to comprehensively estimate all forms of precipitation, including rain, drizzle, snow, graupel, and hail, through the integration of satellite data and ground-based rain gauge measurements (Huffman et al., 2016). The accuracy of TRMM dataset has validated by many studies through comparisons
with observation data and other reanalysis datasets (Kittel et al., 2018; Villarini et al., 2009). For evaporation, we utilized the third version of Global Land Evaporation Amsterdam Model (GLEAM v3) product (https://www.gleam.eu/), which employs

a set of algorithms to separately estimate the different components of land evaporation (Miralles et al., 2011). Several studies have demonstrated that this product aligns well with flux measurements and multisource product ensemble (Munier et al., 2014; Robinson and Clark, 2020). And, as mentioned above, the runoff measurements on a basin scale are provided by the CAMELS dataset, which is derived from site observations.

Finally, the most challenging component to estimate in the water budget equation is the Terrestrial Water Storage Change (TWSC) as it includes water both on and below the Earth's surface. In the previous studies, the measurement of gravity field changes, as provided by the Gravity Recovery And Climate Experiment (GRACE) product, has been frequently employed for the estimation of the TWSC (Luo et al., 2020; Kabir et al., 2022). This approximation is based on the assumption that, for a given large-scale basin, variations in mass are primarily attributed to changes in TWSC. However, the assumption is fragile when applied to small basin, leading to significant uncertainty in estimating TWSC for basins with areas less than 63,000 km$^2$ (Lehmann et al., 2022). This study focuses on the basins dataset from the CAMELS, with most basin areas being smaller than this threshold. To avoid introducing additional uncertainty into the analysis, we need alternative methods to estimate TWSC.

Assuming that TWSC can be retrieved through a combination of different water storages, we obtained the four-layer soil moisture from ERA5 Land and Snow Water Equivalent (SWE) from GlobSnow to estimate overall TWSC. This approach has been implemented in the investigation of Hoeltgebaum and Dias (2023), yield a high consistency between estimated TWSC and GRACE observation (i.e., correlation coefficient exceeding 0.71). Another consideration in this method is that the decomposed TWSC products (i.e., soil moisture and SWE) can correspond to the results simulated by hydrological model, thereby allowing us to correct water budget residuals, as discussed later.

Overall, all datasets were resampled to a daily time step, and then aggregated over basins through simple averaging to perform analysis of water budget closure on a basin scale from 1998 to 2010. Including the observed runoff from CAMELS, all data were converted to water depth (mm) to construct a unified water budget equation. It is noteworthy that there are certain missing data in GlobSnow SWE varying across basins. To fill these data gaps, we set a window of length 5 centred on missing data. We applied linear interpolation within the window for gap filling. If linear interpolation was not feasible due to, for instance, the absence of valid values within the window, mean climatology was employed to fill the missing data. To illustrate this, we randomly selected nine basins and visually depicted the gap filling process in supplemental Fig. S1.

**Table 1.** Overview of the products for constructing water balance equation used in this study.

| Variable | Product | Original Resolution | | Original Period | Reference |
| --- | --- | --- | --- | --- | --- |
| | | Spatial | Temporal | | |
| Precipitation | TRMM 3B42V7 | 0.25°×0.25° | Daily | 1998-2019 | *Huffman et al. (2016)* |
| Evaporation | GLEAM v3.8a | 0.25°×0.25° | Daily | 1980-2022 | *Martens et al. (2017)* |
| Soil moisture layer 1/2/3/4 | EAR5 Land | 0.1°×0.1° | Hourly | 1950-present | *Muñoz Sabater et al. (2021)* |
| Snow water equivalent | GlobSnow v3.0 | 25km×25km | Daily | 1979-2018 | *Luojus et al. (2021)* |
| Runoff | CAMELS USGS | Basin scale | Daily | 1980-2010 | *Newman et al. (2015)* |

## 3 Methods

To leverage physical hydrological processes modelling for the decomposition and correction of water budget residuals, the following assumptions are necessary: (1) the hydrological model provides a reliable representation of hydrological processes, ensuring an accurate partitioning of input precipitation; (2) the uncertainties associated with the model forcing and structure can be considered negligible during the modelling process. These two hypotheses form the foundation of this work. To ensure the validity of Hypothesis 1, we employed multiple evaluation variables and corresponding metrics to guarantee the overall reliability of the model, which will be detailed in the model setup section. Additionally, it is pertinent to acknowledge the Hypothesis 2 represents a strong assumption, carrying inherent uncertainties. Despite this, it is necessary for the feasibility of the overall work, and we will further explore the influence of this hypothesis on the results in the discussion section.

### 3.1 Decomposition of water budget residuals: inconsistency and omission residuals

Our strategy for decomposing water budget residuals is grounded in the computational closure of the hydrological model. As previously discussed, conceptualized as a closed directed graph, the difference between the inputs and outputs of the model must necessarily equal the change in state variables. Stated differently, there is a water balance between the forcing and simulated variables of the model, with no physical inconsistency residuals present. Therefore, setting the inconsistency residuals in Eq. (3) to zero allows us to derive the water budget equation of the hydrological model as follows:

$$Res_o = P_{forcing} - ET_{sim} - R_{sim} - \frac{\mathrm{d}TWS_{sim}}{\mathrm{d}t}, \tag{4}$$

where the subscripts "forcing" and "sim" denote the forcing and simulation values, respectively. It is crucial to clarify that all variables in Eq. (4) are derived from the model itself, rather than from measurement, and can therefore be considered physically consistent. On the other hand, integrating the multisource datasets described in Sect. 2.2 into Eq. (3) yields the total water budget residuals (i.e., $Res$). For convenience, we refer to the water budget characterized by the hydrological model as the simulation system and the one constructed by multisource datasets as the measurement system. When the hydrological model

calibrated against multiple variables measured by the multisource datasets and achieves reliable performance, we consider the water budget represented by the simulation and measurement systems to be comparable. At this point, the difference between Eq. (3) and (4) represents the omission residuals (i.e., $Res_i = Res - Res_o$), indicating the water fluxes or storages omitted by the original equation. Thus, the total water budget residuals can be decomposed into inconsistency and omission residuals. It is noteworthy that while the inconsistency residuals are absent in the simulation system—a physical consistent system—omission residuals may still exist due to inherent omissions in the original equation. Hence, the left-hand side of the Eq. (4) may not be zero.

Considering the comparability of available datasets and model simulations, we have developed more specific expressions for Eq. (3) and (4), as depicted below.

$$Res = Res_i + Res_o = P_{TRMM} - ET_{GLEAM} - R_{USGS} - \frac{\mathrm{d}SWE_{GlobSnow} + \mathrm{d}SM_{ERA5}^{0\sim50cm} + \mathrm{d}SM_{ERA5}^{50\sim289cm}}{\mathrm{d}t}, \tag{5}$$

$$Res = Res_o = P_{TRMM} - ET_{sim} - R_{sim} - \frac{\mathrm{d}SWE_{sim} + \mathrm{d}SMS_{sim} + \mathrm{d}GRS_{sim}}{\mathrm{d}t}, \tag{6}$$

where the subscripts indicate variable sources, such as measurements and simulated values, and superscripts for $SM$ denote the depth of soil layers to be aggregated. The above water budget equations are discretized employing a simple central difference scheme with a two-day time step at the daily scale (Petch et al., 2023). Then, the residuals are calculated at daily scale and subsequently aggregated to the monthly and annual scales for further analysis.

It is important to further clarify that the hydrological model used in this study (see below) divides total soil moisture into soil water storage ($SMS_{sim}$, hereafter SMS) and groundwater reservoir storage ($GRS_{sim}$, hereafter GRS). The soil moisture measurements, ERA5, on the other hand, employs the H-TESSEL (Hydrology Tiled ECMWF Scheme for Surface Exchanges over Land) land surface scheme to characterize land surface hydrological processes (Balsamo et al., 2009), dividing soil into four layers (i.e., 0~7 cm, 7~28 cm, 28~100 cm and 100~289 cm). In the H-TESSEL model, the upper 50 cm of soil column is defined as the effective depth for generating surface runoff. To ensure consistency between the simulation and measurement systems, we match the top 50 cm of ERA5 soil moisture with the soil water storage in the hydrological model used, while the depth range of 50 cm to 289 cm corresponds to the groundwater reservoir storage in the same model.

**3.2 Multisource datasets correction framework for achieving water budget closure**

Here, we introduce an innovative Multisource Datasets Correction Framework grounded in Physical Hydrological Processes Modelling to enhance water budget closure, termed PHPM-MDCF. Unlike traditional correction methods that use uncertainty (typically derived from the variance of multisource datasets for the same variable or priori estimation) as a weight for allocating water budget residuals, this framework leverages the hydrological model—a physical consistent system—as a constraint to

correct the measurement system. Figure 2 indicates the flowchart for the correction framework and the procedure is described as follows:

- Step 1: Initialization of the basic computing unit. Calibrate hydrological model, calculate the total water budget residuals from the original datasets, and then decompose them into inconsistency and omission residuals following the method outlined in Sect. 3.1. This step is denoted as iteration 0.

- Step 2: Correction for the inconsistency residuals. Allocate inconsistency residuals based on the magnitude of differences (i.e., the distance between simulation and measurement systems) between simulated and measured values for each
variable in Eq. (5) and (6). This difference indicates the correction direction and magnitude for each variable, which facilitates the convergence of the measurement system toward the simulation system. Here, an initial correction rate of 0.5 is set to gradually correct the multisource datasets, thereby avoiding potential uncertainties that arise from excessive correction. Formally, the allocation of inconsistency residuals can be described by the following equation:

$$M_c^v = M_o^v - Res_i \times \frac{d_v}{d_{all}} \times \alpha, \tag{7}$$

where $M_c^v$ is the corrected measurements of variable $v$, and $M_o^v$ is the original measurements; $d_v$ is the difference between simulation and measurement of variable $v$, and $d_{all}$ represents the aggregate of differences for all variables; $\alpha$ is the correction rate, with an initial value of 0.5.

- Step 3: Calibration and evaluation of the model. Recalibrate and evaluate the hydrological model using the datasets corrected in the previous step to assess the reliability of this correction. If the recalibrated model yields unreliable
simulations, consider this correction excessive, halve the correction rate, and repeat Step 2. Otherwise, maintain the correction rate and proceed with the next iteration of correction. The consideration behind this step is that excessive correction may lead to the measurement system going out of bounds, preventing further convergence of the two systems. This is to say, the iterative process involves continual trial and error, with each error prompting us to approach the next correction more cautiously.

- Step 4: Iteration and termination of correction. Iterate through Steps 2-3 to gradually correct the datasets until the inconsistency residuals decreases to 10% of its initial value or the correction rate falls below 4%.

The design goal of the PHPM-MDCF is to impose soft constraint on multisource datasets through the calibration compensation capability and the physical consistency feature of the hydrological model. Such a constraint is referred to as "soft" because,
unlike traditional methods that import "hard" constraints, the correction process does not strictly require residuals to be zero immediately. Instead, it aims to advance the convergence between the simulation and measurement systems, as illustrated in Fig. 3. In extreme case, when the measurement system is corrected to be identical to the simulation system, all measurements would become physically consistent. This process can be seen as a collapse from Eq. 5 to Eq. 6. The efficiency of ultimately

closing residuals depends on the ability of model to accurately characterize real world, and this can vary across different

locations.

Notably, the correction is performed at the daily scale, aligning with the model step. In the subsequent application of the PHPM-MDCF, the measurements are derived from the data provided in Sect. 2.2. In addition, through experimentation, the parameter settings in the PHPM-MDCF (i.e., initial correction rate, decay rate of the correction rate, and correction termination

threshold) have been tailored to suit the current study area (Table S2). When applying this framework to other regions, additional adjustments and testing may be required.

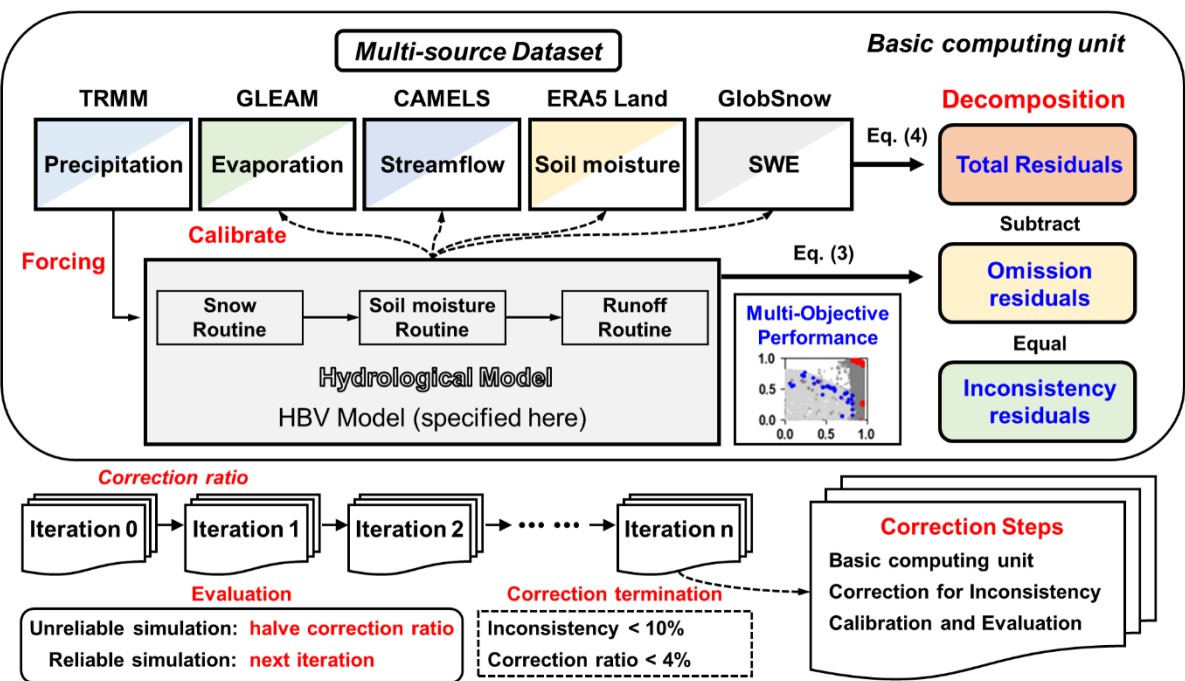

**Figure 2.** Flowchart of the multisource datasets correction framework grounded in physical hydrological processes modelling, PHPM-MDCF.

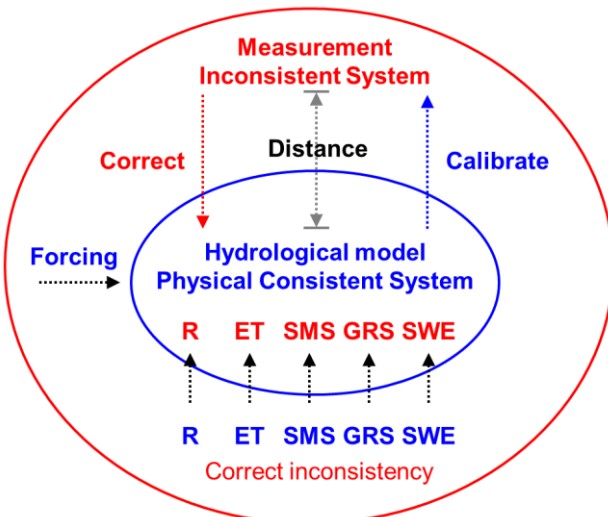

**Figure 3.** Illustration of the correction process advancing convergence between the simulation and measurement systems. The measurement system is corrected to approach the simulation system, while the simulation system is refined via parameter calibration to better approximate the measurement system. As a result, the distance between the two systems is reduced, leading to better physical consistency in the corrected measurement system.

### 3.3 Model setup and calibration

In the present investigation, we employed the Hydrologiska Byråns Vattenbalansavdelning (HBV) model, to implement our correction framework. The conceptual HBV model was developed by the Swedish Meteorological and Hydrological Institute (SMHI) in the 1970s (Bergström, 1976). Given its straightforward yet effective design and minimal input requirements, this model has attained broad recognition and application within the global hydrological modelling scientific community, which has also been tested in the CAMELS basins (Feng et al., 2022). Here we provide brief details and refer the reader to the above references for a fuller description.

The basic structure of the HBV model comprises three main modules: the snow routine, soil moisture routine, and runoff routine, as illustrated in Fig. A1. Starting with precipitation forcing, water flux traverses through the three modules, accumulating in various state variables such as snow and soil water. Ultimately, water is released through three reservoirs—soil moisture, upper zone, and lower zone reservoirs—as quick runoff, interflow, and base flow. Thus, the overall soil moisture can be divided into soil water storage (i.e., the first reservoir) and groundwater reservoir storage (i.e., the combination of the latter two reservoirs). In the current study, the HBV model is configured to run of a daily basis, aligning with both the forcing and evaluation datasets, ensuring the feasibility of subsequent correction. Table A1 lists the free parameters slated for calibration in the HBV model, providing their descriptions and respective ranges.

Here, a multi-objective global optimization algorithm, the Non-dominated Sorting Genetic Algorithm II (NSGA-II), is applied for parameter calibration of the HBV model. Owing to its optimization efficiency, this algorithm has been extensively used in hydrological modelling practices around the world (Mostafaie et al., 2018). For more details about the algorithm, see Deb et

al. (2002). We implemented the calibration framework using the NSGA-II algorithm in a Python environment with the DEAP package (Fortin et al., 2012). Five calibration objectives are considered, including R (runoff), ET (evaporation), SMS (soil moisture storage), GRS (groundwater reservoir storage) and SWE (snow water equivalent). Meanwhile, the Kling-Gupta Efficiency (KGE) metric (Gupta et al., 2009) is utilized to evaluate the simulation performance of R and ET, while the Pearson correlation coefficients (r) is employed to evaluate the performance of SMS and GRS, considering potential discrepancies in

their magnitudes arising from differences in soil layer depth. Finally, the Root Mean Square Error (RMSE) is applied to evaluate the simulation performance of SWE. Ideally, the optimal simulation is characterized by values of 1 for the first two metrics and 0 for the last one. The detailed description of the evaluation metrics is provided in Appendix B.

## 4 Result

### 4.1 Distribution of water budget residuals and its components across the CAMELS basins

In this section, we investigate the spatiotemporal distribution of water budget residuals for each component decomposed using the method proposed in Sect. 3.1 across the large sample of the CAMELS basins. This result provides insights into the two primary sources of non-closure issue in water budget equation—physical inconsistencies among the original datasets and water fluxes or storage omitted in the original equation. To ensure the robustness of the results, as mentioned previously, it is essential that hydrological model reliably represent hydrological processes. With reference to previous studies (Knoben et al. 2019;

Clark et al., 2021; Aerts et al., 2022), we have adopted KGE $\geq -0.41$ and r statistically significant at the 5% level as criteria for guaranteeing reliable simulations. The multi-objective simulation performances of the HBV model are detailed in Appendix C. In general, the majority of basins (475, accounting for 72.24% of the total basins) achieved reliable simulations across all variables. Among them, we have observed that the central and western CONUS present relatively greater challenges for modelling. This pattern and its potential causes will be further explored in the ensuing discussion.


Within the 475 basins demonstrating reliable simulations, in Fig. 4 we plotted the spatial distribution of the long-term monthly mean water budget residuals ($Res$), inconsistency residuals ($Res_i$), and omission residuals ($Res_o$). An important observation from comparing different rows of Fig.4 is that $Res$ shares a similar spatial pattern with $Res_i$, whereas $Res_o$ exhibits some differences. This pattern exists across different quantile ranges of the residuals. For instance, $Res$ and $Res_i$ both present an

east-west gradient for three statistical measures (i.e., min, median, max), with low values occur along the western coastline and high values primarily concentrated in eastern inland basins. The exception is a cluster of low median values located in the central CONUS. Interestingly, the minimum values of $Res_o$ display a contrasting spatial pattern, with higher values in the west and lower values in the east. The spatial difference in median and maximum values of $Res_o$ are not pronounced. These patterns

lend support to the underlying assumption that the drivers of inconsistency residuals and omission residuals are fundamentally
different, and thus can be decomposed from the total water budget residuals.

Figure 5 further illustrates the temporal distribution patterns of the three residuals in terms of seasonality. It is readily
discernible in the figure that the similarity between $Res$ and $Res_i$ reappears, manifesting distinct seasonal patterns with more
pronounced negative trends during the cold seasons (i.e., October to the following April) and positive trends during warm
seasons (i.e., May to September). On the contrary, $Res_o$ tends to be mainly positive except from September to November; its
extent of variability is also significantly smaller than that of the other two residuals. In regard to magnitude, $Res_i$ is much
greater than $Res_o$, whether considering positive or negative bias. From the above results, we can conclude that $Res_i$
predominates within $Res$, exhibiting significant spatiotemporal difference from $Res_o$. These two residuals may combine or
offset each other to collectively form the total water budget residuals. The potential factors affecting the spatiotemporal
distribution and proportion of $Res$ will be further investigated in Sect. 4.4.

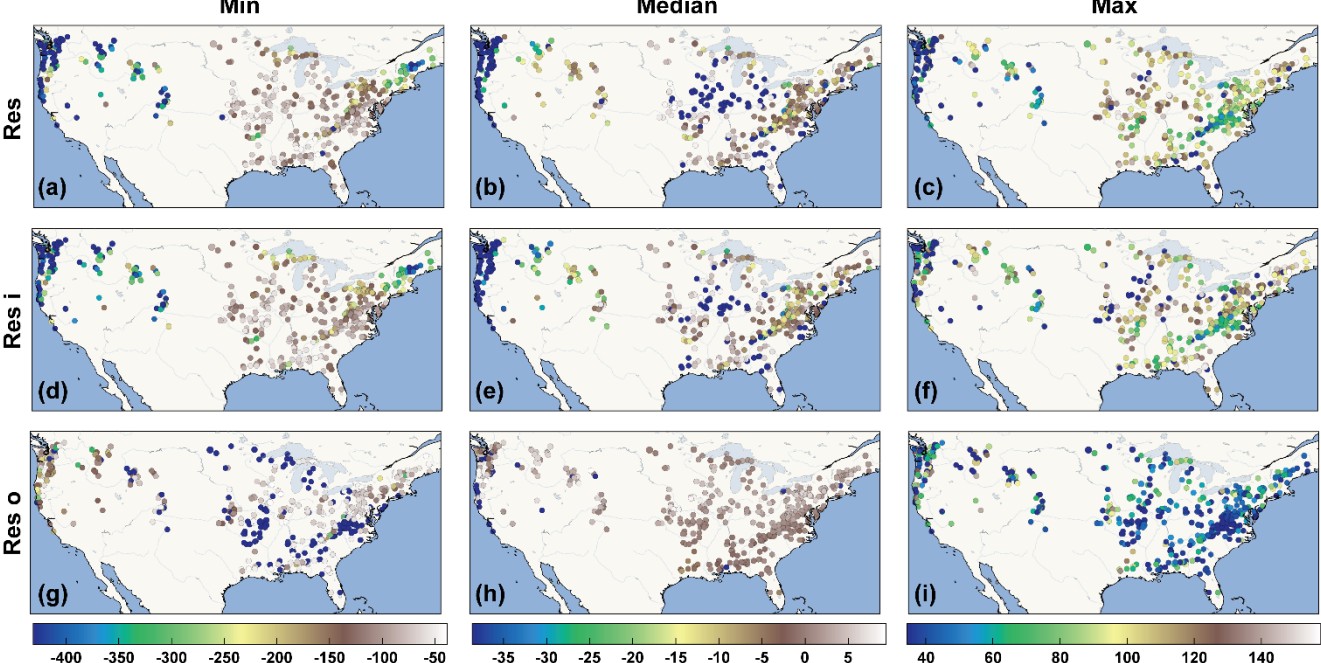

**Figure 4.** Spatial distribution of long-term monthly mean water budget residuals ($Res$), inconsistency residuals ($Res_i$), and omission
residuals ($Res_o$) across 475 CAMELS basins with reliable simulations. The unit of residuals is "mm".

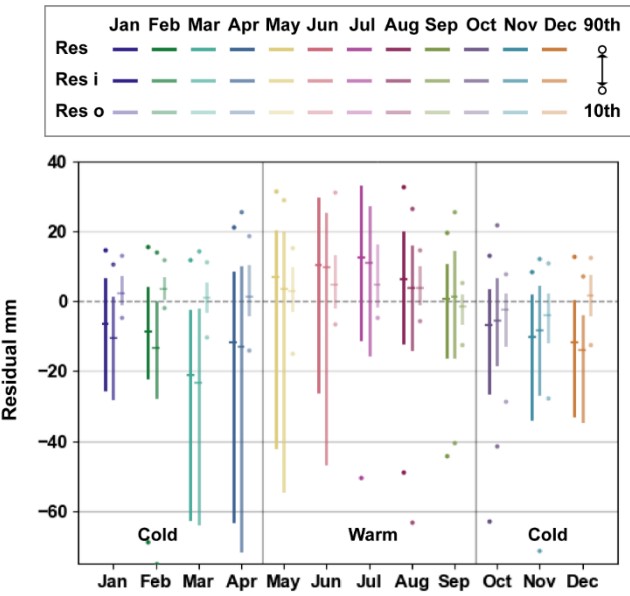

**Figure 5.** Temporal distribution of monthly water budget residuals ($Res$), inconsistency residuals ($Res_i$), and omission residuals ($Res_o$) across 475 CAMELS basins with reliable simulations. Boxplot-like diagrams describe variability across catchments, and outliers represent the 10th and 90th percentiles. The unit of residuals is "mm".

## 4.2 Efficiency of the PHPM-MDCF

We are now tackling the third question through the proposed multisource datasets correction framework (PHPM-MDCF) across the 475 CAMELS basins with reliable simulations. For illustration, several case basins have been selected to demonstrate the correction process and its efficiency.

Figure 6 shows the correction results at the case basin numbered 1013500 (for more details about the basin number, see Newman et al., 2015). As expected, the time series of $Res$ and $Res_i$ after correction (red lines) tend to be flatter and closer to zero compared to their uncorrected counterparts (blue lines). This becomes more apparent as the timescale increases. However, despite recalibrating the model with corrected datasets, $Res_o$ driven by the omission in water budget equation exhibited no substantial changes before and after correction (e.g., the monthly mean absolute values maintain around 6.5 mm, see Fig 6f). This phenomenon occurs because we only corrected the inconsistency residuals with reference to the simulation system, while the omission accounting for addition water terms should not be corrected in the existing datasets.

To get an impression of the PHPM-MDCF correcting water budget residuals, the bottom row of Fig. 6 shows the variation of mean absolute values of three residuals with increasing correction iterations at the monthly scale. The results indicated that the correction process led to a significantly reduction in $Res$ and $Res_i$, decreasing from 42.8 mm and 44.3 mm to 6.9 and 8.6 mm

(approximately 83.9% and 80.7% reduction). Although water budget residuals cannot be fully corrected to zero in this framework (as they do in traditional methods), we argue that this correction efficiency is satisfactory enough. It is rooted in physical hydrological process modelling, thus potentially strengthening the physical relationships among the components of the water balance. The final corrected result for this case basin are presented in Fig. S2, depicting the time series of multisource datasets before and after correction. In the following sections, we will provide further evidence of the credibility of this correction framework.

The correction results for several other case basins (i.e., numbered as 1137500, 2177000, 6311000 and 14092750) are presented in Fig. S3-6. Their absolute mean monthly residuals decreased by 70.4%, 58.1%, 40.3%, and 54.0%, respectively, providing evidence for the effectiveness of the PHPM-MDCF. To have a clearer idea of the ability of the correction framework to reduce water budget residuals across all the CAMELS basins, Fig. 7 shows the map of the percentage reduction in monthly total water budget residuals after corrections. In general, the PHPM-MDCF demonstrated robust performance across most basins, with an averaged reduction percentage of 49% across all basins. The correction efficiency exhibits a latitudinal-dependent decline pattern, which primarily due to the small initial residuals in low latitude regions (Fig. 4). In high-latitude regions, such as the western coastline and eastern inland basins, the potential correction space is much larger, leading to higher correction efficiency (in terms of absolute value).

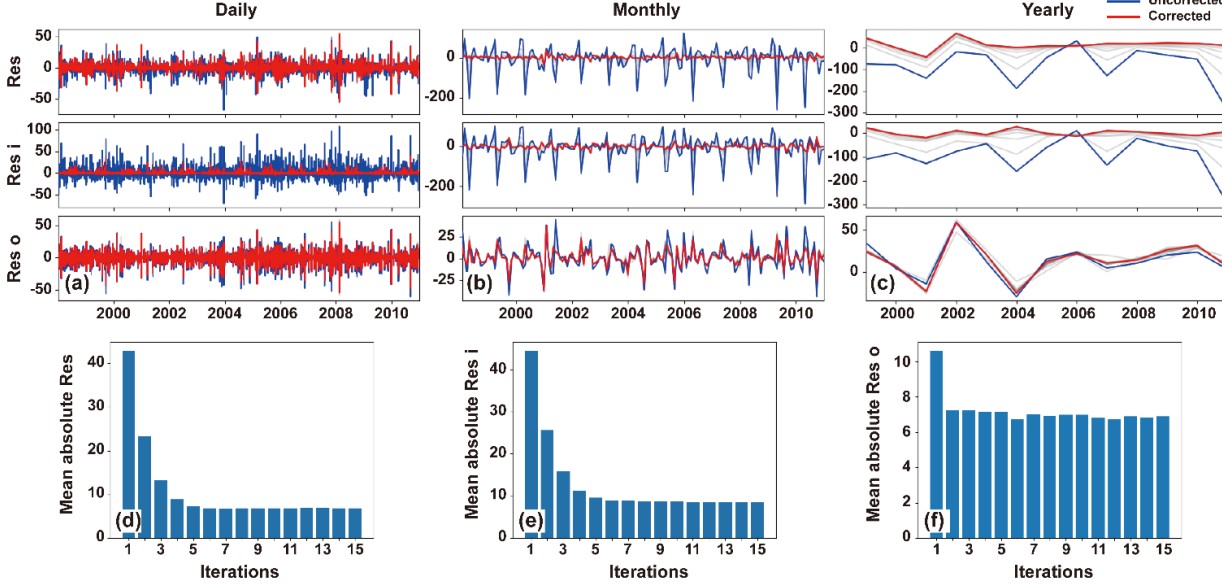

**Figure 6.** Correction results of water budget residuals for multisource datasets at basin 1013500. (a-c) Time series of water budget residuals ($Res$), inconsistency residuals ($Res_i$), and omission residuals ($Res_o$) at daily, monthly and yearly scales, grey line represents residuals during the correction process. (d-f) Variation of long-term mean absolute values of three residuals with correction iterations at the monthly scale. The unit of residuals is "mm".

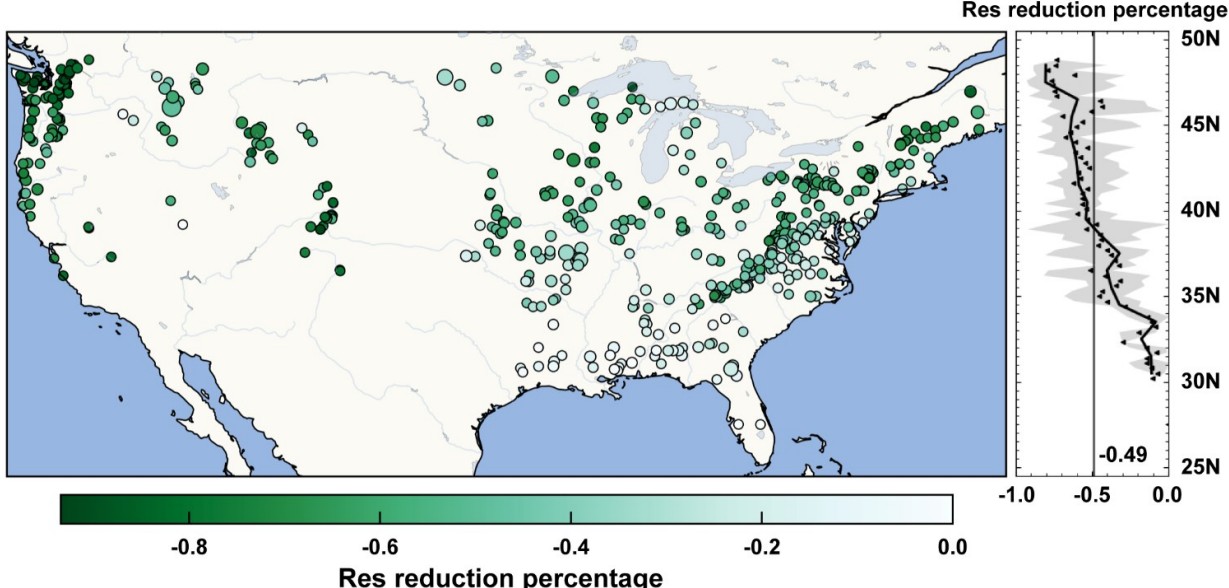

**Figure 7.** The percentage reduction of monthly total water budget residuals after correction through the PHPM-MDCF. Zonal means (right panel) include mean (black scatters), median (black line) and range (gray shading). The vertical line indicates the mean value of -0.49 for all basins.

## 4.3 Credibility of multisource datasets correction

### 4.3.1 Convergence between simulation and measurement system

As we stated before, the core objective of the PHPM-MDCF is to promote the convergence between the simulation and measurement systems (Fig. 3). In fact, this process can be divided into two parts. The first part, namely the measurement system approaching the simulation system, which is implemented by correction procedures, has gained confidence from the significant reduction in the inconsistency residuals (Fig. 6). On the other hand, to illustrate the convergence of the simulation system towards the measurement system, we present the changes in model simulation performance before and after correction of case basin 1013500, as depicted in Fig. 8. From the figure, we can clearly see that both the population solution sets (ranging from light to darker grey scatters) and the Pareto fronts (ranging from blue to red scatters) tend to the optimal point at the upper right corner after correction. More intuitively, Fig. S7 presents a comparison of measurements and simulations for each variable before and after correction. It is evident that the relationship between measurements and simulation is significantly strengthened after correction. These results suggest that the PHPM-MDCF has the ability to enhance the convergence between the simulation and measurement systems, supporting the credibility of the correction results to some extent.

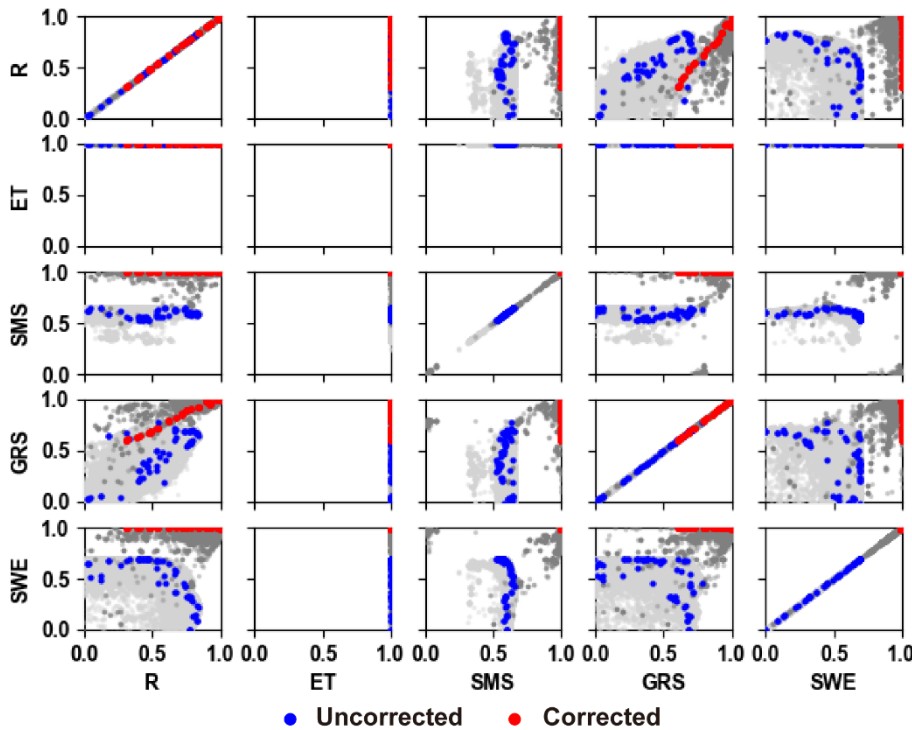

**Figure 8.** Comparison of multivariable simulation performance before and after correction at basin 1013500. Light grey and dark grey indicate population solution sets before and after correction, and blue and red indicate Pareto fronts before and after correction. Metrics evaluating SWE simulation performance have been normalized for consistency. The subplot in the second row, second column shows that the evaporation simulation maintains highly accurate at this basin, due to the alignment between the HBV algorithm and measurements.

### 4.3.2 Noise experiments

To further demonstrate the credibility of multisource datasets correction, we designed a series of noise experiments and applied them to the case basin 1013500, therefore examining whether the PHPM-MDCF can effectively handle the manual noises and produce robust correction results. These experiments are summarized in Table 2, where the first three experiments set different types of single-point noise at different positions of the same original datasets, and the last experiment adds an equal-length Gaussian white noise sequence to the runoff sequence. Eventually, two new noisy datasets were generated, as illustrated in Fig. S8 and S9. For clarity, we refer to them as NS1 (i.e., noise sequence) and NS2, and designate the noise-free datasets as OS (i.e., original sequence). The noise points are ordered from 1 to 4.

First, we examined the adaptation capability of the PHPM-MDCF to single-point extreme errors. The top row of Fig. 9 compares the differential form of the OS and NS1, highlighting the impact of the three noises. The first two noises introduce extremely unreasonable values in the runoff measurements, while the third noise significantly affects water balance by altering

all water budget variables, as evidenced in Fig. 9c-d. Through the application of the PHPM-MDCF for NS1 correction, we derived a new corrected sequence and compared it with the previous OS-based corrected sequence. In terms of runoff correction, as shown in Fig. 9c, whether extreme large or small noises (i.e., noise 1 and 2 with differences of three standard deviations), the correction process constrains them to reasonable runoff processes. This is achieved by the representation of physical hydrological processes underlying the correction strategy, which constrains the corrected values to avoid producing extreme outliers. Furthermore, water imbalance caused by combination of multivariable single-point noises can also be constrained to minimal levels through correction (Fig. 9d).

Another concern here is whether the correction of extreme noises in runoff will propagate to other variables, potentially leading to a series of unreasonable correction results, as questioned by Abolafia-Rosenzweig et al. (2020) regarding traditional methods. In Fig. S10, we specifically focus on the correction results around three single-point noises to address this question. The fact that simultaneous corrections of other variables during extreme runoff noises correction did not significantly differ from OS-based corrections further enhances our confidence in PHPM-MDCF. It suggests that the soft constraints based on physical hydrological processes will not lead to compensatory errors, as seen in traditional methods due to the rigid allocation of water budget residuals. From a theoretical perspective, the PHPM-MDCF assigns the weights of residual correction based on the distance between measurements and simulation for each variable. In the presence of a single extreme bias, the large distance between the measurement and simulation of the corresponding variable leads to a larger correction being applied to that variable, while the weights for other variables remain unaffected. However, in traditional methods, the correction weight for each variable remain constant over time, and the final residuals are constrained to zero. This leads to the propagation of extreme biases across different variables.

Subsequently, we assessed the robustness of correction results after incorporating Gaussian white noise into the original sequence. From the comparison between OS-based and NS2-based correction results (Fig. 10), it can be seen that the addition of Gaussian white noise slightly changed the correction in runoff, namely a minor decrease in the high-value range (with a slope less than 1). However, the overall evolution trend of runoff remains unchanged, as it is still constrained by the same hydrological physical processes. In such a basis, as excepted, the correction of other variables is minimally affected by Gaussian white noise in runoff.

In summary, the results yield from the above experiments indicate that both single-point noise and Gaussian white noise have minimal impact on the corrections. The final correction results are constrained by the hydrological model, with random errors in measurements not significantly altering the allocation of water budget residuals. The physical relationships among various water budget variables, as representation by the model, are also imposed onto the measurements through the correction process.

**Table 2.** Description of the noise experiments to examine the credibility of multisource datasets correction.

| ID | Description | Position of the noises | Noise sequence |
|---|---|---|---|
| Exp. 1 | A single positive-biased noise is added to R, with a magnitude of three standard deviations | Noise1: 1998-09-18 | |
| Exp. 2 | A single negative-biased noise is added to R, with a magnitude of three standard deviations | Noise2: 1999-04-26 | NS1 |
| Exp. 3 | A set of positive-biased noise at the same position are added to R, ET, SMS, GRS, and SWE, with a magnitudes of one standard deviation | Noise3: 2001-12-16 | |
| Exp. 4 | A series of zero-mean random Gaussian white noise is added to R, with a standard deviation of 20% relative to the original sequence | Noise4: the entire sequence | NS2 |

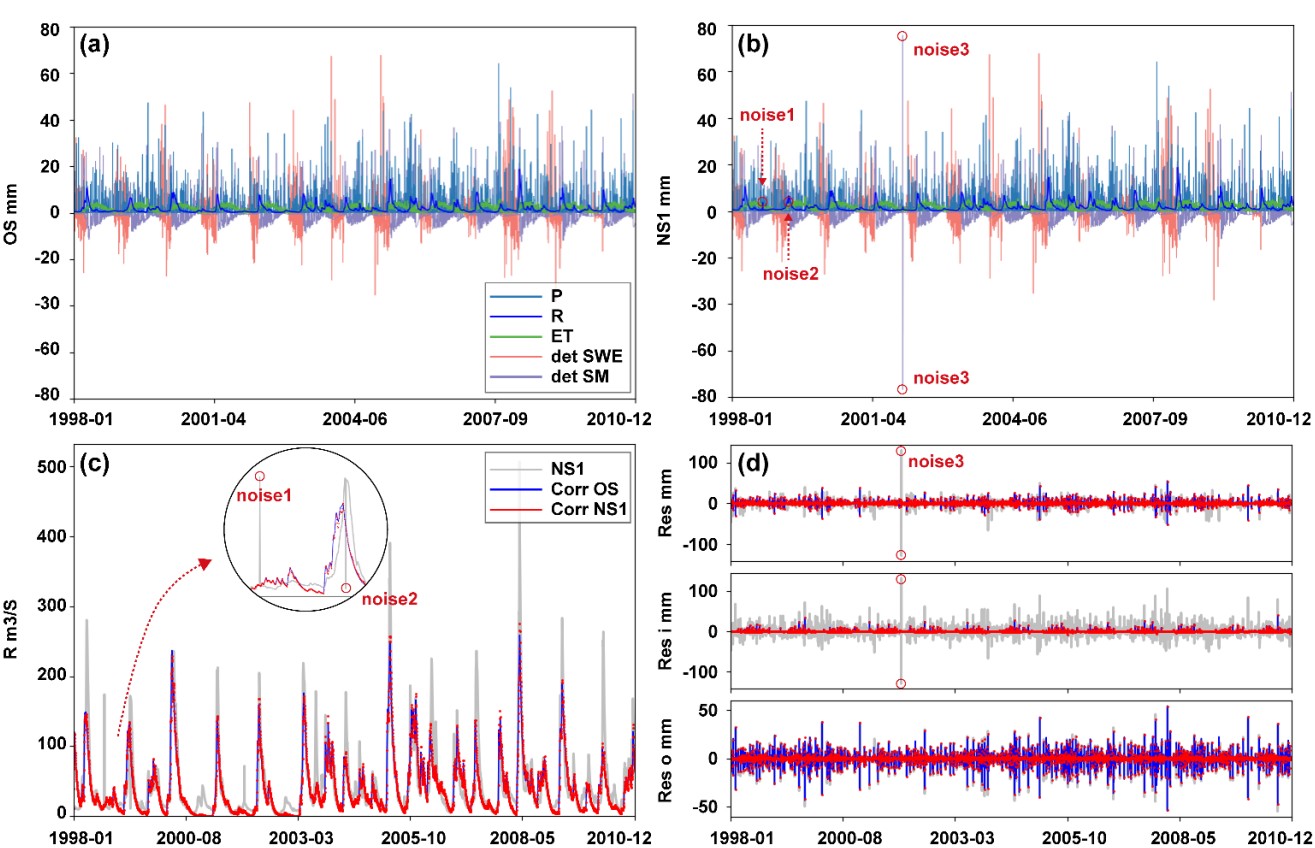

**Figure 9.** Correction results for multisource datasets corresponding to noise experiments 1-3. (a-b) Time series of OS and NS1 in form of differences. (c) Comparison among the runoff noise sequence (NS1), OS-based runoff corrected sequence (Corr OS), and NS1-based runoff corrected sequence (Corr NS1). (d) Comparison of water budget residuals generated by the three sequences at daily scale.

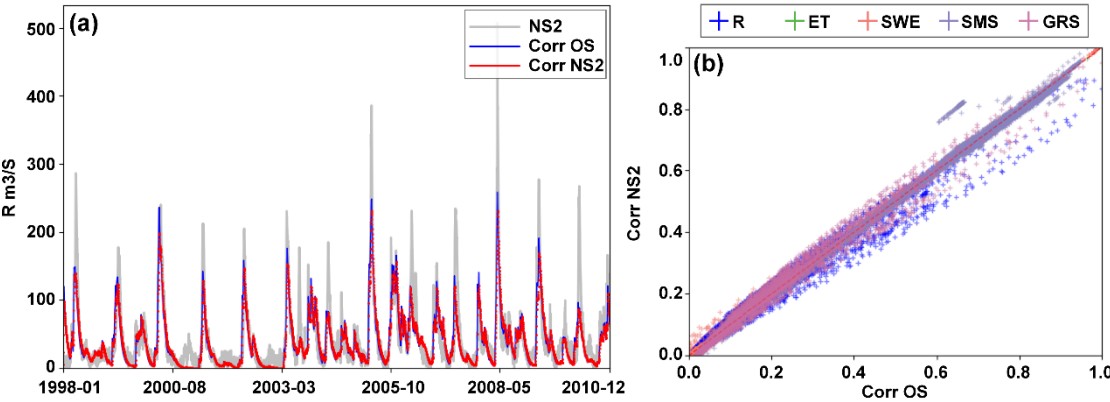

**Figure 10.** Correction results for multisource datasets corresponding to noise experiments 4. (a) Comparison among the runoff noise sequence (NS2), OS-based runoff corrected sequence (Corr OS), and NS2-based runoff corrected sequence (Corr NS2). (b) Comparison of multivariable between OS-based correction and NS2-based correction in terms of standardized values.

### 4.3.3 Comparison with existing correction methods

Previous analysis and experiments clarify the unique characteristics of the PHPM-MDCF, which impose closure constraints based on physical hydrological processes. This differs significantly from existing correction methods, such as PR and CEnKF (Luo et al., 2023). In this section, we conducted a comparison analysis with them to further evaluate the reliability of the PHPM-MDCF. To implement existing correction methods, support from multisource measurements for each water component is essential for calculating the residual allocation weights. Here, we obtained monthly datasets from Lehmann et al. (2022), which include 11 precipitation, 14 evaporation (ET), 11 runoff (R) and 2 terrestrial water storage (TWS) datasets (Table S3). The datasets previously utilized in this study were also included for data fusion and correction (Table 1). In general, these datasets were processed to a uniform monthly scale and a common period (2003-2010), and subsequently aggregated to the basin scale. Several representative basins (numbered 1539000, 1557500, and 3070500) were selected to illustrate the differences between the PHPM-MDCF and existing methods, based on the spatial coverage of multisource datasets.

Figure 11 presents a comparison of the monthly correction results from three methods (i.e., PR, CEnKF, and PHPM-MDCF) for three main water budget components at basin 1539000. Note that the measurements of precipitation are not compared here, as the PHPM-MDCF does not perform correction for this variable. It is clear from the figure that both the PHPM-MDCF and CEnKF method exhibit minimal correction of ET, whereas the PR method significantly expands the range of ET, particularly increasing seasonal peaks. This arises from the assumption of the PR method that relative errors are proportional to the relative magnitudes of each variable (Abhishek et al., 2022). However, in many cases, this assumption may not hold true.

In terms of the R and terrestrial water storage change (TWSC), the overall trends of the correction results from the three methods are generally consistent. However, the CEnKF appears to produce greater fluctuations in R and shows limited correction of TWSC (Fig. 11). This is linked to the computational mechanism underlying CEnKF, where the Kalman gain—or the error covariance between measurements and the ensemble mean of multisource datasets—determines the magnitude of the residuals corrected for each variable. Specifically, the measurements of R to be corrected is based on in-situ obervations,

while the multisource dataset includes model simulations and remote sensing values. Potential mismatches between the grids and basins may lead to significant discrepancies, resulting in an greater allocation of correction for R. On the contrary, measurements of TWSC are limited and primarily derived from GRACE, which results in relatively small error covariance and, consequently, smaller corrections. Furthermore, as previously noted, such method may generate unreasonable corrections due to propogation of extreme errors, such as the negative R values in Fig. 11b, which are more likely to occur in small basins.

PHPM-MDCF avoids these issues by considering physical process constraints, leading to more reasonable corrections. Additionally, it does not rely on multisource datasets and can perform correction on any model time step and for any model output variable. The TWSC derived from SWE and SM is consistent with GRACE TWSC, which also demonstrates the reliability of this framework in retrieving TWSC. The comparison results for the other two representative basins are shown in Fig. S11-12, leading to similar conclusions.

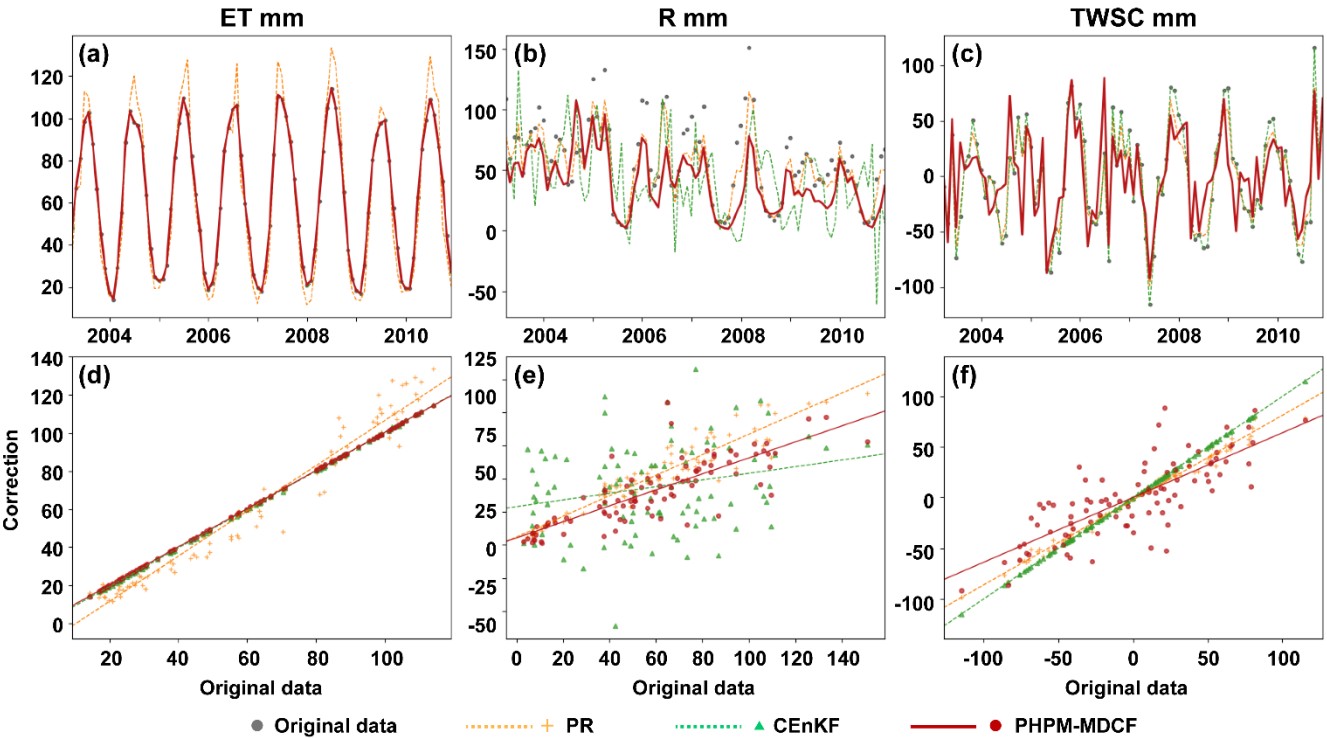

**Figure 11.** Comparison of monthly correction results between the PHPM-MDCF and existing methods (PR and CEnKF) at basin 1539000. (a-c) Time series of the original and corrected measurements of evaporation, runoff, and terrestrial water storage change. (d-f) Scatter plots and regression lines of the original and corrected measurements.

## 4.4 Potential influencing factors of water budget residuals

### 4.4.1 Factors influencing spatial distribution

In this section, we conducted a preliminary exploration of the potential factors influencing the formation and distribution of water budget residuals. As shown in Fig. 4, all three water budget residuals are subject to strong spatial organization, and these patterns are in agreement with previous studies. For example, Kauffeldt et al. (2013) found negative residuals (i.e., runoff coefficient > 1) along the western coastline of CONUS, while the eastern region showed notable positive residuals (i.e., P-R > ET). Other studies investigating water budget residuals with diverse dataset combinations have revealed similar spatial patterns (Zhang et al., 2016; Gordon et al., 2022). Therefore, we speculate that the spatial distribution of water budget closure is predominantly influenced by the characteristics of the basin.

Here we focus on the total water budget residuals (i.e., *Res*) and attempt to relate it with the hydro-meteorological conditions and the basin area. To bring out these relationships, from Fig. 12, three regression curves are obtained by correlating mean absolute residuals at different timescale with basin areas over 475 CAMELS basins. The negative gradients of the curves imply

a scale effect in the water budget non-closure phenomenon that as basin area increases, the water balance constructed from multisource datasets can be enhanced. Moreover, as expected, hydro-meteorological conditions within the basin play a crucial role in controlling the distribution of water budget residuals. The clear delineation between different levels of daily precipitation and runoff coefficient revealed in Fig. 12 strongly supports this reasoning, where multisource datasets yield larger water budget residuals in basins with high precipitation and runoff coefficients—large red spots are located in the upper portion of the figure. These results highlight the risks of using multisource datasets for hydrological inference in humid and small-scale basins—specifically, potential physical inconsistencies—and underscore the need to carefully test the water balance assumption.

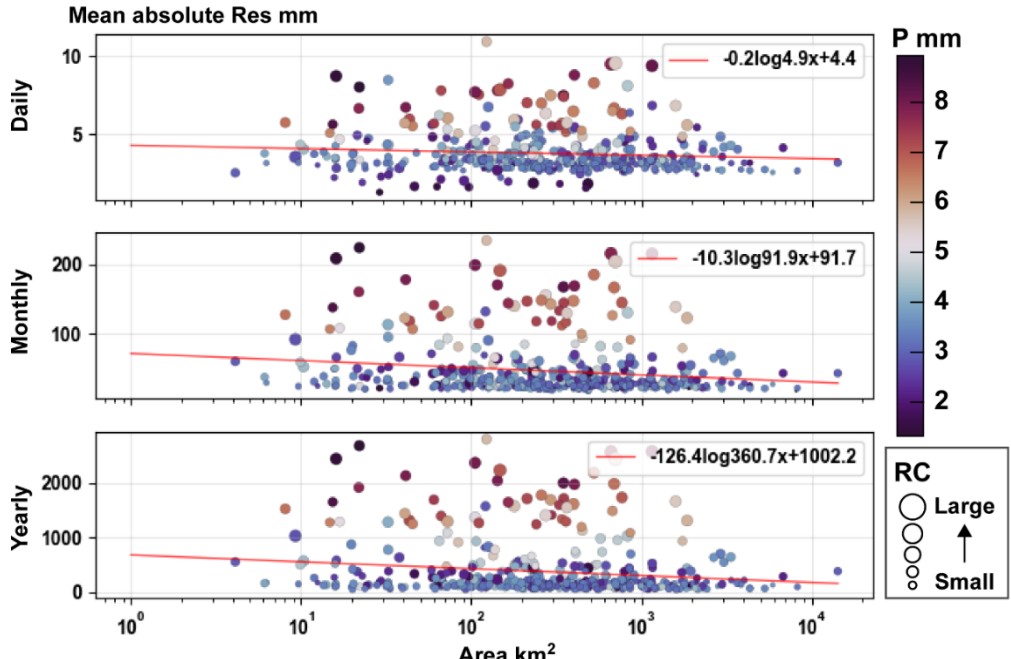

**Figure 12.** Relationship between the mean absolute of water budget residuals, basin area, long-term average daily precipitation, and runoff coefficient (RC) over 475 CAMELS basins with reliable simulations. The respective red lines represent the linear regression of residuals with basin area for each timescale.

### 4.4.2 Factors influencing temporal distribution

The pronounced seasonal pattern of non-closure residuals depicted in Fig. 5 is quite interesting. To gain more insight into the observed pattern, we compare it with the temporal factors reported in the literature. The first and foremost reported factor associated with the observed negative biases in *Res* during the cold season is the underestimation of precipitation (Newman et al., 2015). This systematic bias is related to phenomena such as snowfall, freezing rain, and non-convective precipitation that occur during the cold season, where measurements and simulations are prone to significant errors, including the well-

know undercatch phenomenon (Kauffeldt et al., 2013; Robinson and Clark, 2020). Another key factor influencing water budget non-closure is connected to the temperature and evaporation dynamics. Abolafia-Rosenzweig et al. (2020) evaluated the water budget residuals over 24 global basins and found that the likelihood of positive biases in the water balance increases with rising temperatures, which likely induced by the potential uncertainties in evaporation estimates. The research by Lv et al. (2017)

also support this perspective, indicating that the underestimation of evaporation is a primary contributor to the water budget non-closure. In summary, according to the literature, cold-season precipitation and warm-season evaporation seem to be the primary drivers of the temporal distribution of $Res$. To examine this reasoning, while obtaining the true values is impossible, we can provide evidence by comparing evaporation and precipitation, along with the corresponding residuals, between the cold and warm seasons.

Figure 13 depicts the relationship by separately comparing the ratios of evaporation and precipitation for the cold and warm seasons, with the corresponding water budget residuals. For the cold season, the scatter points can be split into two distinct regions along the vertical line where the ratio is 1. The scatter points in the left region indicate basins where cold-season precipitation is lower than in the warm season, leading to relatively smaller absolute residuals (clustered around zero residuals). In contrast, scatter points for basins with dominant cold-season precipitation are dispersed below the zero residual line, with

larger negative residuals becoming more prevalent as the proportion of cold-season precipitation increases. In other words, regions where cold-precipitation constitutes a larger proportion of the water budget residuals are more sensitive to the underestimates of precipitation, resulting in larger negative residuals. Furthermore, we observed similar trends in the warm season, where a higher proportion of warm-season evaporation is associated with larger positive residuals (the red dots exhibit an upward trend to the right). These results confirm the perspective of previous research, highlighting the potential uncertainties

in measurements of cold-season precipitation and warm-season evaporation.

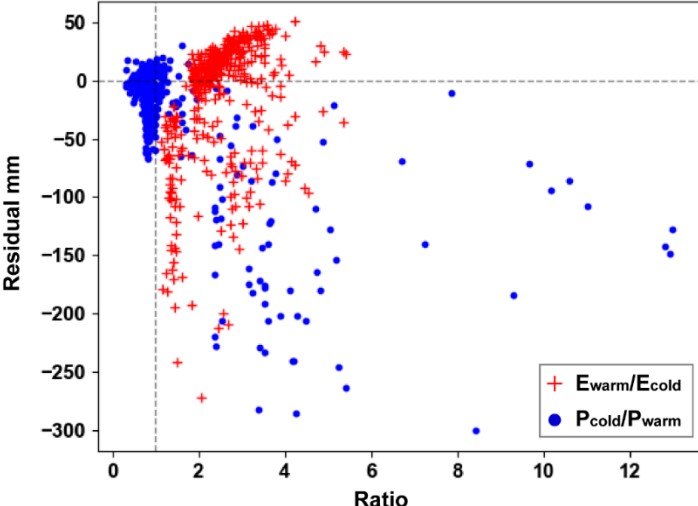

**Figure 13.** Relationship between the ratios of evaporation and precipitation for the cold and warm seasons separately and the corresponding water budget residuals. Note that blue represents residuals for the cold season, and red represents those for warm season. The seasonal division are consistent with Fig. 5. The unit of residuals is "mm".

### 4.4.3 Factors influencing the proportions of residuals components

Another interesting finding in Sect. 4.1 is that the magnitude of $Res_o$ is significantly smaller than that of $Res_i$. As a result, $Res$ is dominated by $Res_i$, leading to a highly consistent spatiotemporal distribution between them. However, the underlying question is what this implies and which factors drive the proportions of the residuals components.

$Res$ reflects the degree to which the measurements achieve water budget closure. In this study, we argue that two key conditions are necessary for using measurements to describe theoretical water balance. The first one is that measurements of different water components must be physically consistent. In practice, however, this condition is often challenging to meet due to inconsistencies and uncertainties in data production processes from different sources, which can result in non-zero $Res_i$ (Luo et al., 2020). The second crucial, yet frequently overlooked, condition is the completeness of the water budget equation. Building on the work of Gordon et al. (2022), we developed a more generalized water budget equation (Eq. (3)) and use $Res_o$ to account for the water imbalances caused by omitted water. From this perspective, $Res$ results from the interplay between $Res_i$ and $Res_o$, either through their accumulation or mutual cancellation. Therefore, the low proportion of $Res_o$ essentially suggests that our description of the water budget equation is comparatively comprehensive.

Consider that if our description of the water budget equation is incomplete and omits a significant water component, $Res_o$ would likely exert a greater influence on $Res$, resulting in a more pronounced discrepancy between $Res$ and $Res_i$. To examine this, we intentionally exclude the SWE component from the water budget equation to evaluate its impact on the decomposition of $Res$. This is a plausible scenario in practice, as it is likely that this component was not considered when reconstructing the TWSC. Figure 14 illustrates the comparison between $Res_o$ derived from the decomposition method excluding SWE (hereafter $Res_o^{NSWE}$), and its original values. It is evident that $Res_o^{NSWE}$ exhibits greater variability compare to the original values (i.e., with smaller minimum values and larger maximum values). The median differences indicate that the likelihood of increased omission residuals is higher after excluding SWE (Fig. 14b). Such differences reveal that omitting crucial SWE storage component results in a greater degree of water imbalance, and, as expected, this effect is more pronounce in high-latitude and high-elevation regions (Fig. 14d-f). Moreover, the spatiotemporal distribution of $Res_o$ has changed (Fig. S13-14). Notably, during the cold season (December to February), the proportion of $Res_o$ is much higher and exhibits a significant positive trend. These findings align with our definition of $Res_o$, which refers to the water imbalance caused by omitted water. It also supports the validity of our decomposition method to some extent, and highlights the importance of a comprehensive water budget equation in evaluating water balance.

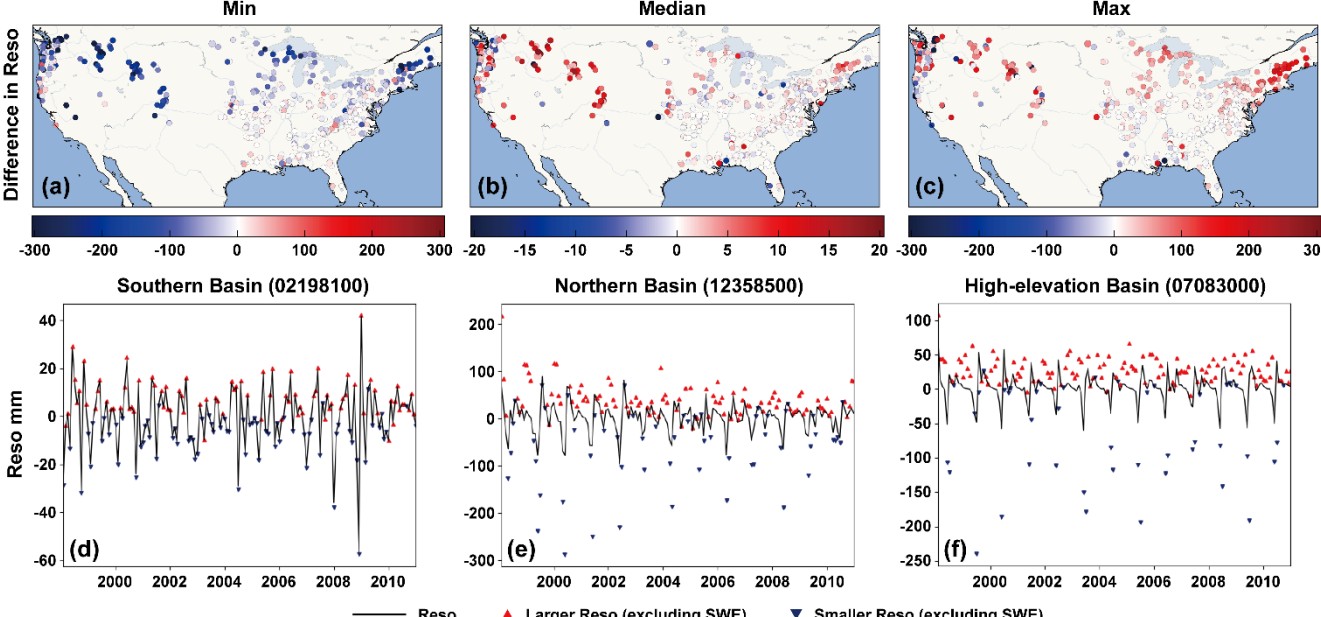

**Figure 14.** Comparison of $Res_o$ obtained from residual decomposition excluding SWE with the original values. (a-c) Spatial distribution of monthly mean $Res_o$ excluding SWE minus its original values. (d-f) Time series of $Res_o$ excluding SWE and its original values at the southern basin (02198100, 32.96°N), northern basin (12358500, 48.33°N), and high-elevation basin (07083000, elevation of 3.56 km) at monthly scale. The unit of residuals is "mm".

# 5 Discussion

## 5.1 What Lies Within the Realm of Belief

The foundation of modern experimental science is based on empiricism, emphasizing the repeatability of experiments, i.e., whether the results can perfectly reproduce observations. This idea has far-reaching implications across various fields, with a classic example being hydrologists always aiming for their model predictions to closely match observations. Importantly, the underlying assumption of this approach is that our observations are perfectly approximate reality and can be seen as true value. In most of small scale studies, such as those conducted in laboratory or field settings, this might hold true. However, as we shift our focus to larger spatial scales, obtaining observations directly often becomes challenging, thus necessitating reliance on indirect observations, which could potentially undermine this assumption. As a consequence, our confidence in the observations, or better referred to as measurements, may diminish, which is precisely the new challenge we face in the era of big data.

When we lack sufficient confidence in any single measurement, the utilization of multisource data fusion becomes a method to mitigate errors from all sources of measurements, thereby reducing uncertainty. Within the process of data fusion, the basic

step is to determinate the weights of all components. The ensemble mean method assumes an equal weight for all components, while the simple weighted method estimates weights based on the priori uncertainties, which are typically the differences between each component and the average of all measurements (Sahoo et al., 2011). In the widely used triple collocation (TC) method, weights can be determined by calculating errors (uncertainties) based on the similarity of the triplet inputs, without the need for "ground truth" (Stoffelen, 1998). Some other methods also determine uncertainty through manually assigned constants or error propagation calculations (Munier et al., 2014; Ansari et al., 2022). However, all of these methods face the same issue, the true value may be unattainable, and the determined error or uncertainty involves subjective factors. This presents a logical paradox: we resort to data fusion due to the absence of a true value, yet during the fusion process, we paradoxically assume the existence of this true value to estimate uncertainty. Essentially, we need to answer a fundamental question: what do we truly believe in?

The answer is what we have truly learned. A better approach is to leverage our existing knowledge about the physical world to enhance our confidence in measurements. In fact, this concept embodies to some extent a Bayesian philosophy and is reflected in many fields. Here, we present two modern examples to illustrate this idea. The first one is the atmospheric reanalysis, which has been one of the most significant topics in atmospheric science since the 19th century. This technique employs numerical models and assimilation techniques to integrate multiple types of historical measurements a unified modelling framework and assimilation scheme, thereby generating continuous and consistent estimates of climate states. In essence, its aim is to unify our knowledge system (i.e., numerical models) with the measurement system, thereby enhancing the credibility of the model output.

Another example is a research in the field of hydrology, where Liao and Barros (2022) proposed an Inverse Rainfall Correction (IRC) framework to improve Quantitative Precipitation Estimates (QPE) in headwater basins. Their fundamental concept is that errors propagate from precipitation to runoff, enabling the reversal of precipitation errors by calculating runoff simulation errors from distributed hydrological models and applying the travel time distribution for correction. In this example, existing knowledge is represented by the hydrological model, which is assumed to reflect the true physical processes and is then used to enhance the confidence in precipitation measurements.

The proposed correction framework (PHPM-MDCF) capitalizes on this concept by iteratively advancing the convergence between the knowledge system (i.e., hydrological model and water balance equation) and the measurement system, thus enhancing the credibility of the measurements. Although our current knowledge may not be entirely precise—for example, the depiction of hydrological processes in models may lack accuracy—it remains foundation upon which we can rely and strive to refine in the future. Furthermore, several underlying concepts in this framework, such as residuals decomposition and advancing water budget closure through correction, aligns with a recent study (Wang and Gupta, 2024). They introduced a novel hybrid model (i.e., Mass-Conserving-Perceptron) and discussed its potential application, including the bias correction

(lacking confidence for the measurements) and examination of non-observed interactions with the environment (corresponding to the omission errors). Coupling the PHPM-MDCF with hydrological models that provide stronger interpretability is a valuable and promising research effort, as it can offer insights into the physical attribution of water budget non-closure and enable more reasonable correction.

## 5.2 Limitations and Paths Forward

It is our opinion that some traditional hydrological inferences are based on a philosophy that involves some long-standing and problematic assumption arise from the unwarranted confidence in measurements. However, the fact that truth is almost impossible to be measured due to the complexity of real-world physical processes hampers the foundation of inferences, especially in large scale studies that employing multisource non-field data. The presented framework has advantages by integrating widely applicable water budget equation and reliable representation of hydrological process using a hydrological model, which significantly mitigates this issue and enhance our confidence to the corrected datasets. Although the efficiency and credibility of the PHPM-MDCF have been examined in the previous sections, there are several limitations and uncertainties worthy of further discussion.

### 5.2.1 Uncertainty of forcing data

Here, we return to the Hypothesis 2 posed at the beginning of the method section. As we acknowledge, the uncertainties arising from the forcing and model structure undeniably exist and were a limitation in this study. First, the uncertainty in the forcing may arise from two aspects, one is the inaccuracy of the datasets themselves, and the other is the uncertainty introduced by the scaling process (i.e., the conversion from grid scale to basin scale). To investigate the sensitivity of correction results to forcing data, we re-conducted multisource datasets correction using Daymet precipitation data at the same case basin (1013500) and compared it with the original correction (forcing by TRMM). The comparison of the two precipitation products is presented in Fig. S15, where Daymet precipitation is significantly lower. The top panels of Fig. 15 display slight differences between the two corrections; for instance, the Daymet correction shows larger SWE (with a slope greater than 1), while other variables are smaller. These differences can be entirely explained by variations in precipitation forcing. Nevertheless, the temporal patterns of all variables under the two corrections remain broadly consistent, with determination coefficients of all regression curves exceeding 0.70 (Fig. 15b). Theoretically, the consistency of correction stems from three aspects. Firstly, it is attributed to the adaptability of hydrological model to the input data, specifically the calibration compensation capability we described in the introduction (Wang et al., 2023). This enables the hydrological model to generate reasonable representation of hydrological process even with imprecise forcing. Secondly, as discussed in Sect. 4.3.2, the PHPM-MDCF serves as a soft constraint and utilizes the distance between measurements and simulations to allocate residuals correction, thereby mitigating the propagation of bias between variables. Thirdly, the uncertainty caused by the mismatch between the grids and basin boundaries is effectively alleviated through the unit conversion (i.e., from volume to depth units). These three features ensure that stability of the correction, rendering it less susceptible to interference from uncertainties in the forcing datasets.

Another evidence of the robustness of the PHPM-MDCF is provided by Fig. 15c-d, where corrected residuals tend to converge after several iterations, despite being forced by different precipitation datasets. The main influence of forcing data is manifested in the omission residuals. As expected, the omission residuals term is simply an approximation of the missing water fluxes or storages in the water budget equation, which can vary depending on the datasets chosen to characterize the equation. In Fig. 15e, the omission residuals driven by Daymet stabilize around 12.5mm, whereas those driven by TRMM stabilize around 6.5mm. Such discrepancy can be further highlighted in the comparison of the residuals time series (Fig. S16). Further investigation would be required to better understand the omission residuals from a physical perspective. For example, a distributed hydrological model with representation of subsurface later flow process will allow us to identify the magnitude of inter-basin interactions; a more detailed description of water budget equation in data-rich environments can help us examine the sources of omission errors. This is undoubtedly important, but not the focus here. In summary, the above results suggest that the correction is minimally sensitive to the choice of forcing, demonstrating the robustness of the correction results. This is achieved by maintaining similar inconsistency residuals—corresponding to a similar correction amount—as long as differences in precipitation do not result in substantial variations in the hydrological processes.

It is noted that the PHPM-MDCF has limitations in addressing inconsistency residuals in forcing. The reasons are twofold. On the one hand, this is due to our neglect of uncertainties in the forcing, which, as indicated by the above analysis, appears to have limited impact on the correction for other variables. On the other hand, this is because the PHPM-MDCF allocates residuals based on the distance between simulations and measurement, while the forcing cannot be simulated within the hydrological model. In this case, is there a potential to correct the inconsistency residuals in the forcing? Clues to this possibility are hidden in the above analysis. Systematic biases in precipitation products are directly reflected in the water budget equation, leading to different total input water volumes. Consequently, with the inconsistency residuals of other variables unchanged, maintaining the water balance would require an increase in omission residuals (Fig. 15e). Therefore, it can be inferred that, with other variables unchanged, TRMM demonstrates superior water budget closure compared to Daymet, which contains smaller inconsistency residuals. In other words, the difference in the two omission residuals reflects the discrepancy in inconsistency residuals contained within the two precipitation products. This portion of the omission residuals difference can be directly corrected in the precipitation. However, it is worth noting that not all omission residuals can be corrected in the precipitation, as it still contains residuals from some unknown omitted water content. Such correction must be relative and based on comparisons between different precipitation products, as true values and perfect water balance equation are unattainable. Another strategy is to couple an atmospheric model with this framework to generate simulated precipitation, allowing for the correction of precipitation products. In subsequent work, we will explore these approaches and try to extend the PHPM-MDCF based on these ideas.

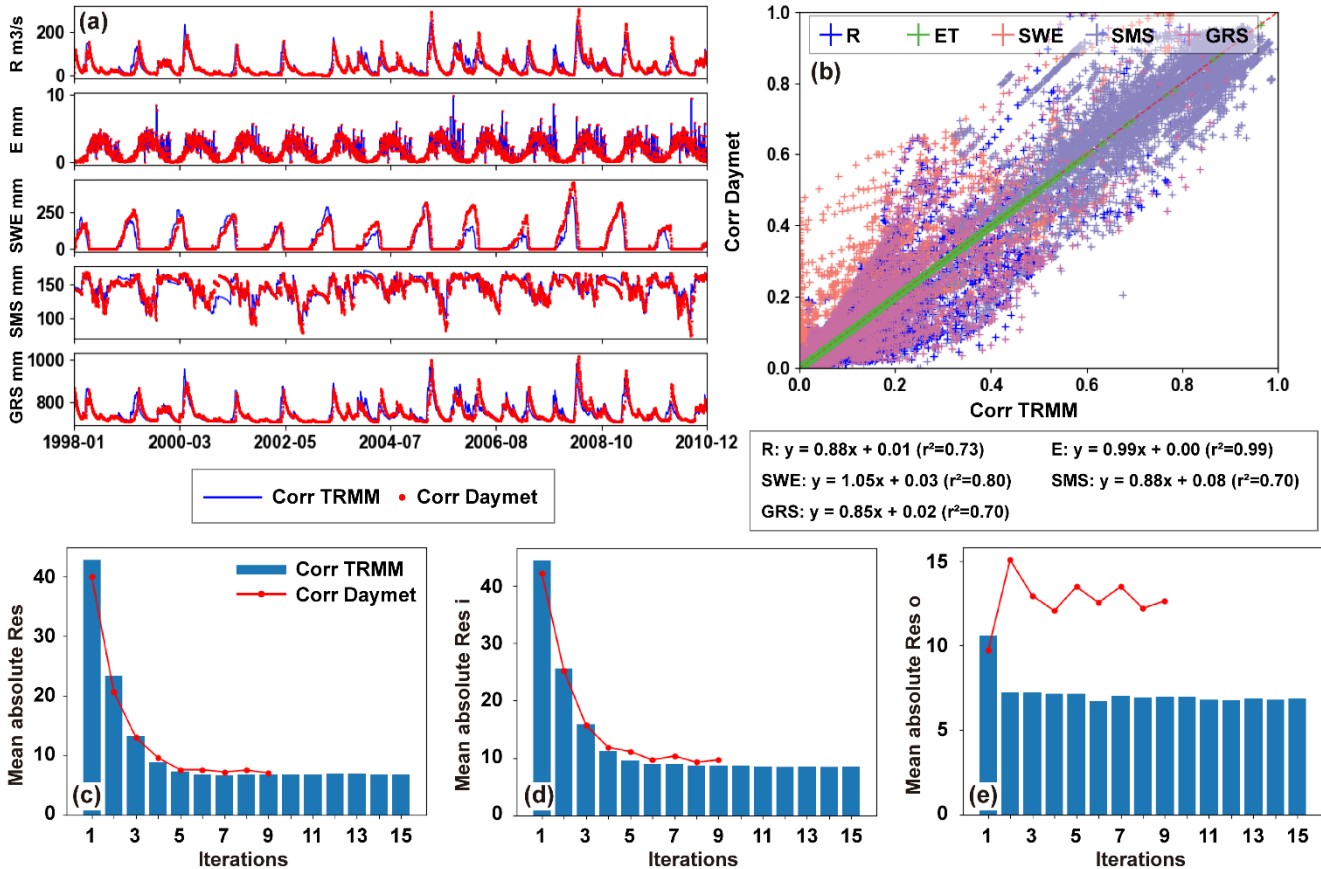

**Figure 15**. Comparison of correction results based on different forcing datasets (TRMM and Daymet) at basin 1013500. (a-b) Corrected time series of five water budget variables. (c-e) Variation of long-term mean absolute values of three residuals with correction iterations at the monthly scale. The unit of residuals is "mm".

### 5.2.2 Uncertainty of model structure

The characterization of physical hydrological processes through modelling constitutes the foundation of the correction framework. The internal model structure is the primary constraint for achieving water budget closure, and thus it is crucial for the final correction results. The selection of the lumped model (i.e., the HBV model) is intended to facilitate the application in large sample basins to derive more general conclusions, as has also been done in many previous large sample hydrology studies (Gupta et al., 2014). The reliability of model simulations has been confirmed by multi-objective evaluation. However, whether the spatial distribution of model performance is intrinsically related to the model structure is crucial to the robustness of the current work.

To address the question, we first compared the model performance with other studies that employed different models. As illustrated in Fig. C1, the model behaviour exhibits strong spatial organization, with unreliable simulations primarily concentrated in the central and western regions of CONUS. This spatial distribution of prediction skill broadly agrees with many previous studies. Brunner et al. (2021) classified this region as an intermittent regime and attribute the unsatisfactory simulation to the complex day-to-day variation of runoff. In their work, all four lumped models with different structures (i.e., SAC, HBV, VIC, mHM) supported the inference. In Yan et al. (2023), a more complex land surface model (i.e., CLM5) were utilized for evaluating the uncertainty of runoff prediction, they reported that the Southwest and Central U.S. showed the poorest prediction skill. A notable pioneering research is by Knoben et al. (2020), who evaluated runoff predictability in CAMELS basins using 36 hydrological models with different structures. After conducting a comprehensive analysis, they generated a multi-model runoff prediction performance map, which aligns closely with the results of this study. Therefore, we deduce that the spatial disparities in model performance, or predictability, predominantly depend on basin and climatic conditions rather than model structure. The consistency of the model performance with prior studies demonstrates that the HBV model is reliable in the context of this study.

To further substantiate the above inference, we categorized basins into four groups based on model performance in runoff and compared the inter-group differences in six types of basin and climatic characteristics (i.e., climate, hydrology, geology, topography, soil and vegetation). The four groups consist of: unreliable performance, reliable performance, below-average performance, and above-average performance. First, the two sample t-test at the 5% level was conducted to examine whether there are significant differences in each characteristics indicator between the unreliable and reliable groups. The indicators exhibit a statistically significant difference were then presented and compared in Fig. S17 and S18. For clarity, here we list indicators whose inter-group difference greater than 30% in terms of median cumulative probability: mean precipitation, mean potential evapotranspiration, aridity index (climate); proportion of silt (geology-soil); mean runoff, runoff coefficient, frequency of high-flow days (hydrology); and all vegetation indicators (vegetation). The significant inter-group differences in these indicators highlight critical basin and climatic characteristics pivotal to the successful modelling of the hydrology system, providing convincing evidence for our inference. In summary, basins with the following characteristics typically pose challenges to simulate: arid regions with low precipitation and high potential evaporation, resulting in a low runoff ratio and frequent alternation between zero flow and high flow. Vegetation in these basins tends to consist of lower vegetation types and lack forests. It is worth noting that, while we have validated the reliability of the HBV model in the current study, its simplistic physics and lumped design structure lead to significant limitations in simulating several processes such as snow and groundwater (Brunner et al., 2021). In other words, the HBV model may not be suitable for accurately representing the reality of these specific processes.

The distinctive perspective of this work lies in utilizing the physical processes described by hydrological model to constrain multisource datasets, thereby enhancing water budget closure among them. In particular, our next priority is to incorporating

more complex models to examine the PHPM-MDCF in different basins with specific hydro-meteorological conditions. For instance, distributed hydrological models and hybrid models (ML-HM) are valuable tools that can improve our understanding of water budget closure through more detailed physical processes representation (Liao and Barros, 2022; Wang and Gupta, 2024). By employing models that generate additional output variables, we can more comprehensively represent the water budget equation and extend the application of the PHPM-MDCF to more complex water budget systems. Additionally, multiple models can be utilized for "ensemble correction", which aids in quantifying uncertainty and providing more robust correction results.

## 6 Conclusions

Advanced measurement techniques open new opportunities for modern hydrological research. However, due to the lack of consistent data production protocols and evaluation standards, physical inconsistencies are prevalent in multisource datasets in the form of water budget residuals. Such inconsistencies undermine our confidence in data reliability and compromise the robustness of hydrological inferences rely on these datasets. In this study, we proposed a multisource datasets correction framework, the PHPM-MDCF, to achieve water budget closure through physical hydrological processes modelling. Build upon the decomposition of total water residuals and the iterative multi-objective calibration, the framework has the ability to reduce the inconsistency residuals among multisource datasets and promote convergence between the simulation and measurement systems. We demonstrated the spatiotemporal distribution of water budget residuals and the efficiency of the PHPM-MDCF across 475 COUNS basins selected by hydrological simulation reliability. Several experiments were conducted to verify the credibility of the framework, including the addition of manual noises and comparisons with existing correction methods. Furthermore, we explored potential factors influencing the spatiotemporal distribution and proportions of residuals. The major study findings are summarized as follows:

1. The results from water budget residuals decomposition indicate that inconsistency residuals dominate the total water budget residuals, showing highly consistent spatiotemporal distributions. In spatial terms, both demonstrate an east-west gradient and concentration of low values along the western coastline and eastern inland basins within CONUS. Temporally, they exhibit negative trends in the cold seasons and positive trends in the warm seasons. On the contrary, the omission residuals, which account for the water quantities omitted in the original water budget equation, have different drivers and thus exhibit distinct distributions compared to the former. This component constitutes a relatively small proportion of the total budget residuals.

2. The PHPM-MDCF demonstrates satisfactory correction efficiency, with an average reduction percentage of 49% in total water budget residuals across all 475 basins after correction. In certain basins, this reduction can exceed 80% (i.e., 84% in basin 1013500). The correction efficiency shows a latitudinal-dependent pattern, with greater absolute values in high latitude regions. The results from noise experiments validated the credibility of the correction framework. Both single-

point extreme noise and Gaussian white noise sequences exert a limited impact on final correction results. Corrections applied to extreme noises in one variable do not propagate to others, thereby avoiding the generation of unreasonable values. Its credibility was further substantiated through comparisons with existing methods.

3.  The water budget non-closure phenomenon exhibits noticeable scale effects and is closely related to hydro-meteorological conditions. This highlights the need for careful consideration of the water balance assumption when applying multisource datasets for hydrological inference in small and humid basins. Moreover, the underestimation of cold-season precipitation and warm-season evaporation could be directly associated with the negative and positive biases in water budget residuals for the corresponding seasons. As a foundation for evaluating water balance, a comprehensive water budget equation is undoubtedly crucial, as underscored by the analysis of residual proportions.

For the first time, this study presents a correction approach to achieve water budget closure based on the physical hydrological modelling. However, the Bayesian philosophy underlying the approach have been implicit in many previous methods, such as atmospheric reanalysis. The only thing we can rely on is our prior knowledge; therefore, continuously promoting convergence between knowledge and measurement systems is crucial for enhancing our confidence. An obvious extension of this research is the inclusion of more disciplines, both within the atmospheric science and broader earth sciences. This contributes to a better understanding in the era of big data of the distinctions and correlations between simulations, measurements, and reality.

**Appendix A: Implementation details of the HBV model**

Figure A1 illustrates the basic structure of the HBV model, encompassing three modules (i.e., snow routine, soil moisture routine and runoff routine) and three runoff components: quick runoff, interflow and baseflow. The cumulative sum of these components constitutes total runoff, which is routed through a triangular unit hydrograph (UH). At each model run step, the runoff at the outlet of the basin is determined. The HBV model is driven by daily precipitation (from TRMM), average temperature (from CAMELS) and potential evaporation (from GLEAM), enabling the simulation of various hydrological fluxes and state variables, including runoff, soil moisture storage, groundwater reservoir storage, evaporation and SWE. Table A1 lists the free parameters slated for calibration in the HBV model, providing their descriptions and respective ranges.

The period from 1998 to 2000 is looped five times for model spin-up and the subsequent 10-year period is used for model calibration. After each calibration, the optimal parameters set is selected from the Pareto fronts. Finally, these optimal parameters are applied to the entire 12-year period to yield the best simulation, thus facilitating the multisource datasets correction.

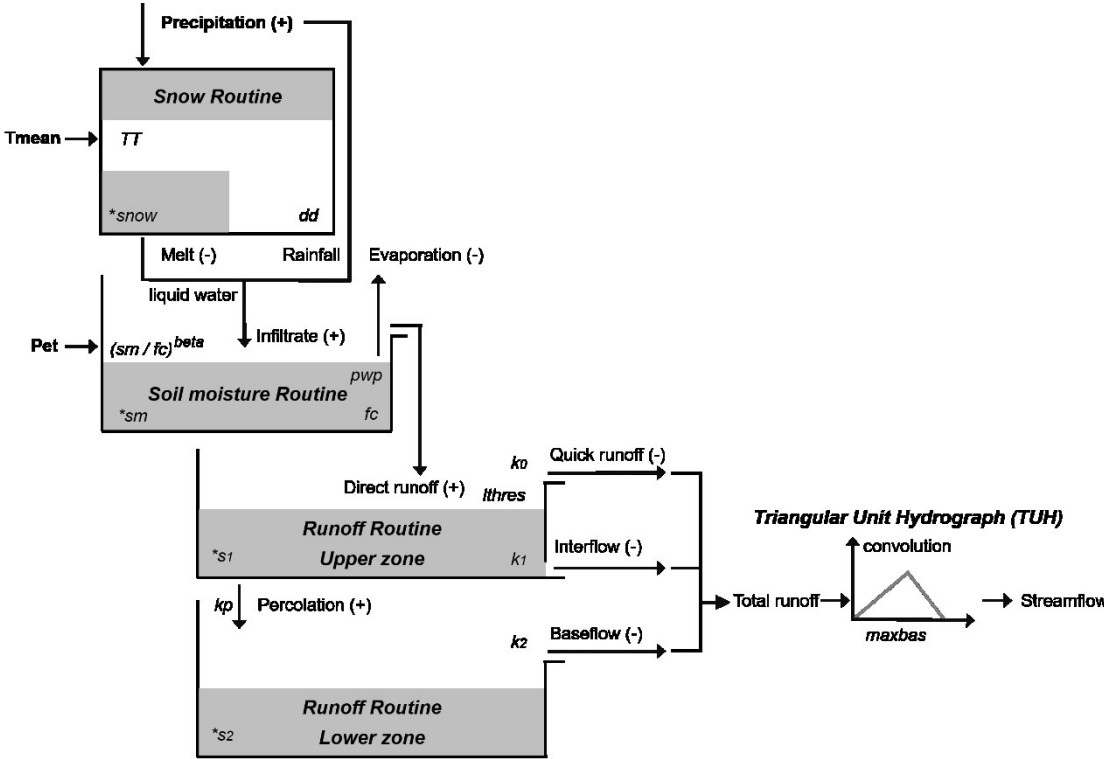

840

**Figure A1.** Schematic structure of the HBV model. The variables marked with asterisk (*) denote water storage, whereas those annotated with positive (+) and negative (-) signs represent the inputs and outputs of the storage.

**Table A1.** The description and ranges of free parameters in the HBV model for calibration.

| Parameter | Unit | Description | Min | Max |
|---|---|---|---|---|
| DD | [mm °C$^{-1}$ d$^{-1}$] | Degree-day factor | 1.0 | 10.0 |
| TT | [°C] | Threshold temperature for snowmelt initiation | -2.5 | 2.5 |
| Beta | [-] | Shape coefficient | 1.0 | 8.0 |
| FC | [mm] | Filed capacity | 10.0 | 600.0 |
| $K_0$ | [d$^{-1}$] | Recession coefficient of the quick runoff | 0.1 | 0.8 |
| $K_1$ | [d$^{-1}$] | Recession coefficient of the interflow | 0.01 | 0.5 |
| $K_2$ | [d$^{-1}$] | Recession coefficient of the baseflow | 0.001 | 0.15 |
| $K_p$ | [d$^{-1}$] | Recession coefficient of the percolation | 0.001 | 5.0 |
| PWP | [-] | Soil permanent wilting point as a fraction of FC | 0.2 | 1.0 |
| HL | [mm] | Threshold water level for near-surface flow | 10.0 | 200.0 |
| maxbas | [d] | Weighting parameter of triangular unit hydrograph | 1 | 10 |

## Appendix B: Evaluation metrics used for model calibration

The Kling-Gupta Efficiency (KGE) metric provides a comprehensive measure of the similarity between simulations and measurements by incorporating three components: correlation, the ratio of standard deviations, and the ratio of means. It has been demonstrated to exhibit superior performance in calibrating hydrological models (Knoben et al., 2020; Aerts et al., 2022). The Pearson correlation coefficient (r) quantifies the extent of shared information between simulations and measurements, characterized by its insensitivity to amplitude and mean values (Lorenz et al., 2014). Thus, it is suitable for evaluating variables that may exhibit mean differences between simulations and measurements, such as SMS and GRS. The Root Mean Square Error (RMSE) is a widely used evaluation metric in hydrological modelling. Despite it is not a normalized metric, its calculation does not involve division, making it particularly suitable for evaluating variables like SWE, which may be a sequence entirely consisting of zeros. Based on the simulated and measured values of the target variables, the three metrics can be calculated using the following formulas:

$$KGE = 1 - \sqrt{(r-1)^2 + (\frac{\sigma_{sim}}{\sigma_{obs}} - 1)^2 + (\frac{\mu_{sim}}{\mu_{obs}} - 1)^2}, \tag{B1}$$

$$r = \frac{\sum_{i=1}^{n}(V_{obs}^i - \overline{V_{obs}})(V_{sim}^i - \overline{V_{sim}})}{\sqrt{\sum_{i=1}^{n}(V_{obs}^i - \overline{V_{obs}})^2}\sqrt{\sum_{i=1}^{n}(V_{sim}^i - \overline{V_{sim}})^2}}, \tag{B2}$$

$$RMSE = \sqrt{\frac{1}{n}\sum_{i=1}^{n}(V_{sim}^i - V_{obs}^i)^2}, \tag{B3}$$

where $\sigma$ is the standard deviation and $\mu$ is the mean; $V^i$ is the target variable at time step $i$ and $n$ is the length of the sequence. The subscripts "sim" and "obs" denotes the simulation and measurements of the variable, respectively. The range and optimal values of the evaluation metrics are detained in Table B1.

**Table B1.** Description of evaluation metrics, including ranges and optimal values.

| Metrics | Full name | Variables to be evaluated | Range | Optimal value |
|---------|-----------|---------------------------|-------|---------------|
| KGE | Kling-Gupta Efficiency | Runoff, evaporation | $(-\infty, 1]$ | 1.0 |
| r | Pearson correlation coefficient | Soil moisture storage, groundwater reservoir storage | $[-1,1]$ | 1.0 |
| RMSE | Root Mean Square Error | Snow water equivalent | $[0, +\infty)$ | 0.0 |

## Appendix C: Simulation performance of the HBV model across CAMELS basins

In this Appendix we present the simulation performance of the HBV model on 653 CAMELS basins. As shown in Fig. C1, the performance of five target variables including runoff, evaporation, soil moisture storage, groundwater reservoir storage, and snow water equivalent, is described by three metrics (i.e., KGE, r, and RMSE). The gradient from white to deep blue indicates progressively better simulation performance. In contrast, red highlights basins of unreliable simulation, determined

by a KGE of less than -0.41 and r value failing the significance test at the 5% level. Table C1 summarizes the multivariable simulation performance of the HBV model across all basins.

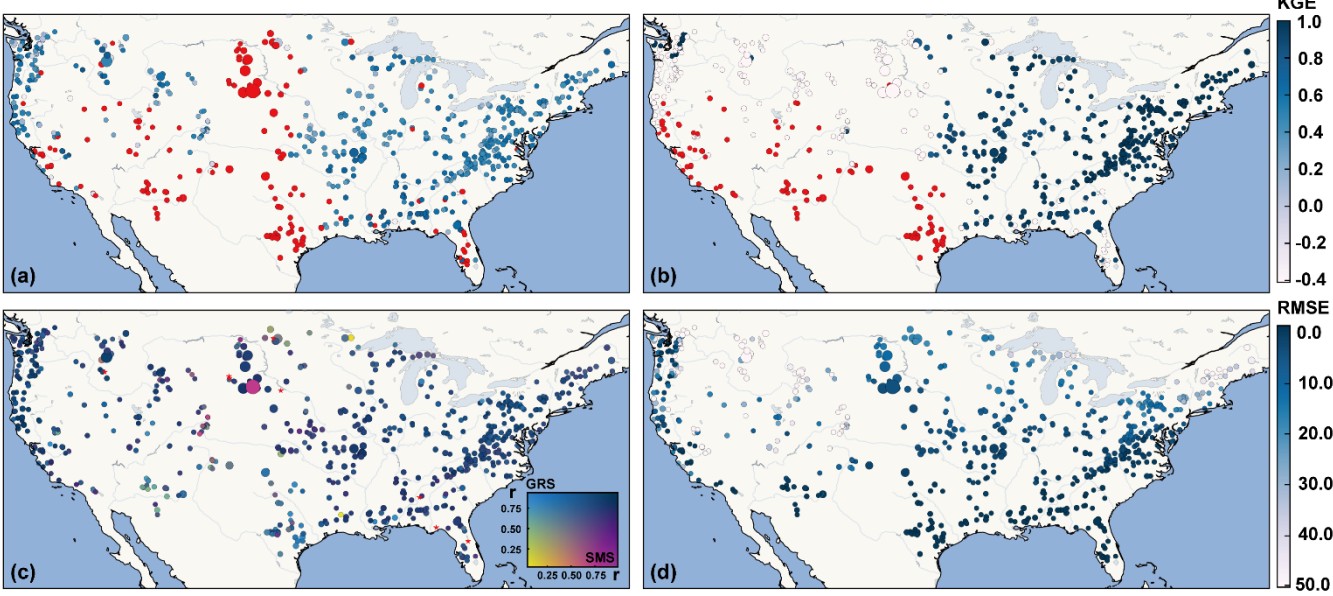

 **Figure C1.** The multi-objective simulation performances of the HBV model across the CAMELS basins. Results are based on (a) runoff, (b) evaporation, (c) soil moisture storage and groundwater reservoir storage, and (d) snow water equivalent. Red dots represent unreliable simulation performance, and the size of points is proportional to the basin area. The unit of RMSE is "mm".

**Table C1.** Performances of the HBV model in terms of five target variables across the CAMELS basins. The last row presents the number and proportion of basins where all target variables are reliably simulated. The unit of RMSE in the table is "mm".

| Variables | Median performance (KGE, r, RMSE) | Range (KGE, r, RMSE) | Reliable Simulations Count (Basins) | Reliable Proportion (%) |
|---|---|---|---|---|
| Runoff | 0.50 | -0.40~0.88 | 499 | 76.42% |
| ET | 0.94 | -0.40~0.99 | 548 | 83.92% |
| SMS | 0.80 | 0.07~0.95 | 645 | 98.77% |
| GRS | 0.72 | 0.02~0.95 | 653 | 100.00% |
| SWE | 5.97 | 0.00~353.34 | - | - |
| All variables | - | - | 475 | 72.74% |

## Data availability

All data used in this study is freely available through public open-source platforms. The TRMM 3B42V7 precipitation production is available at the NASA Goddard Earth Sciences Data and Information Services Center (GES DISC) website (https://disc.gsfc.nasa.gov/datasets/TRMM_3B42_Daily_7/summary, Huffman et al., 2016); the GLEAM evaporation and potential evaporation data from Martens et al. (2017), are available at https://www.gleam.eu/; the EAR5 Land data are available at https://cds.climate.copernicus.eu (Muñoz Sabater et al., 2021); the GlobSnow v3.0 SWE data can be downloaded from the official website: https://www.globsnow.info/swe/ (Luojus et al., 2021).

The basin characteristics and daily runoff records come from the Catchment Attributes and Meteorology for Large-sample Studies (CAMELS) dataset, which can be obtained from https://ncar.github.io/hydrology/datasets/CAMELS_attributes (Addor et al., 2017).

## Author contributions

XDZ: conceptualization, data curation, formal analysis, writing – original draft. DFL: conceptualization, supervision, writing – review and editing. SZH, HW, XMM: supervision and review.

## Competing interests

The authors declare that they have no conflict of interest.

## Acknowledgements

This work was performed as part of the PhD project of Xudong Zheng. The authors appreciate the constructive comments offered by three anonymous reviewers and the editor (Xing Yuan), who have helped improve this paper significantly during its preparation.

## Financial support

This study was financially supported by the National Key Research and Development Program of China (2022YFF1302200) and the National Natural Science Foundation of China (Grant Nos. 52279025 and 42071335).

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
