# Peer review of "Achieving water budget closure through physical hydrological processes modelling: insights from a large-sample study"

_Hydrology and Earth System Sciences, 2024_

## Author Comment (AC1)

**Response to Reviewer RC1**

**Title**: Achieving water budget closure through physical hydrological processes modelling: insights from a large-sample study
**Authors:** Xudong Zheng, Dengfeng Liu*, Shengzhi Huang*, Hao Wang, Xianmeng Meng
**Manuscript ID**: hess-2024-230

**Reply on RC1:**

Many thanks for taking the time and effort to review our paper. All comments from Reviewer RC1 are addressed below with point-by-point responses.

For better readability, replies will start with "**R/**", following the original comments that start with "**C/**" and are shown in **bold**. The revisions to be added into the revised manuscript is highlighted in red. The important parts are highlighted in blue. The quoted content is displayed in *italics*.

**Point-to-point response:**

**C/ This paper emphasizes the issue of decreasing data confidence at the watershed scale in the era of big data, caused by the non-closure of water budget from multiple data sources. In their analysis, the total water budget residuals were quantitatively decomposed into two components, inconsistency and omission residuals, to account for different drivers of water budget non-closure phenomenon. This is an interesting addition, as previous studies have typically given little or only qualitative consideration to the water imbalance caused by omissions in the original water balance equation.**

**Attempting to close the water balance is valuable, both hydrological inference under climate change and hydrological modeling require data that satisfy the basic assumption of water balance. The PHPM-MDCF proposed in this work employ hydrological model to constrain multisource datasets, which is reasonable because hydrological models are well-known for their water balance capabilities. The correction also seems to be effective, which comes from the validation with results from large sample basins.**

**R/** Thanks for your positive feedback and recognition of our work. Your comments are valuable for revising and improving our paper. Below, we provide detailed responses to each of your concerns. The corresponding revisions will be incorporated into the subsequent revised manuscript.

**C/ However, there are still some concerns that need to be explained in the response or addressed in the manuscript. The authors have observed the typical seasonal pattern of non-closure phenomena but lack corresponding explanations. In addition, although the authors decomposed the closure residuals into two parts, it seems that only the inconsistency residuals were corrected. What is the rationale behind this approach? Why were the omission residuals not corrected?**

**R/** Your points are very insightful. Adding explanations about the seasonal characteristics of the non-closure phenomenon will indeed strengthen our argument. In addition, as you mentioned, our framework primarily addresses the Resi (inconsistency residuals), without considering the correction of Reso (omission residuals). This is because we consider it as unaccounted-for water in the original water budget equation, which should be explained by other water components. We provide a more detailed clarification in our response to the second major concern and include further revisions to the manuscript. By addressing these two questions, we anticipate that our manuscript will be significantly improved.

**C/ In summary, this paper is innovative and aligns with the interests of potential readers of the HESS. After careful consideration and revision, this work has the potential to make a significant contribution to this field. As they described, the underlying Bayesian philosophy is an approach for aligning our understanding of natural processes with real-world observations.**

**R/** Thank you again for acknowledging our work and perspectives.

**Major concerns**

**C/ Sect. 4.1, the patterns of the Res are of interest to me. The authors identified typical spatial distributions and compared them with previous studies in Sect. 4.4, explaining these patterns through hydro-meteorological conditions and watershed area. From a physical perspective, this explanation is consistent with common sense and is sufficient for me. However, the temporal patterns of the Res are also of interest (Fig. 5). The authors should provide further explanation in this regard or compare them with previous studies, as this could offer valuable insights into the causes of the non-closure of water balance.**

**R/** Thanks for your suggestion. As we mentioned earlier, we agree with your suggestion to include an explanation of the temporal distribution of Res. This will be addressed in two ways: (1) Comparing the observed seasonal patterns in Res (residuals) with previous studies, and (2) providing an analysis from a physical causation perspective.

Indeed, the temporal distribution of Res shown in Fig. 5 is quite striking. Specifically, as we mentioned in our manuscript, there is a positive bias in Res during the warm season and a negative bias during the cold season. By comparing with previous literature, we found similar temporal distributions and potential influencing factor—namely, the potential underestimation of warm-season evaporation and cold-season precipitation (Kauffeldt et al., 2013; Newman et al., 2015; Lv et al., 2017; Abolafia-Rosenzweig et al., 2020; Robinson and Clark, 2020).

From a physical perspective, the underestimation is related to phenomena such as snowfall, freezing rain, and non-convective precipitation that occur during the cold season, as well as the calculation of evaporation during the warm season.

A further analysis was conducted to examine this by comparing the ratios of evaporation and precipitation for cold and warm seasons separately, along the corresponding Res. Scatter plot shows that basins dominated by cold-season precipitation are more likely to exhibit larger negative Res during cold-season, while basins with higher warm-season evaporation tend to have larger positive Res during warm season.

In both cases, the Res are more sensitive to underestimation of precipitation and evaporation, which is consistent with findings from previous research.

Although it is impossible to obtain true values to evaluate the measurements, these results still highlight potential uncertainties in cold-season precipitation and warm-season evaporation measurements, which could severely impact the assumption of water balance.

The analysis process and corresponding figure to be added into the revised manuscript are given below:

"The pronounced seasonal pattern of non-closure residuals depicted in Fig. 5 is quite interesting. To gain more insight into the observed pattern, we compare it with the temporal factors reported in the literature. The first and foremost reported factor associated with the observed negative biases in Res during the cold season is the underestimation of precipitation (Newman et al., 2015). This systematic bias is related to phenomena such as snowfall, freezing rain, and non-convective precipitation that occur during the cold season, where measurements and simulations are prone to significant errors, including the well-know undercatch phenomenon (Kauffeldt et al., 2013; Robinson and Clark, 2020). Another key factor influencing water budget non-closure is connected to the temperature and evaporation dynamics. Abolafia-Rosenzweig et al. (2020) evaluated the water budget residuals over 24 global basins and found that the likelihood of positive biases in the water balance increases with rising temperatures, which likely induced by the potential uncertainties in evaporation estimates. The research by Lv et al. (2017) also support this perspective, indicating that the underestimation of evaporation is a primary contributor to the water budget non-closure. In summary, according to the literature, cold-season precipitation and warm-season evaporation seem to be the primary drivers of the temporal distribution of Res. To examine this, although it is impossible to obtain the true values, we can provide some evidence by comparing evaporation and precipitation, along with the corresponding residuals, between cold and warm seasons.

Figure 12 depicts this relationship by comparing the ratios of evaporation and precipitation for the cold and warm seasons separately, with the corresponding water budget residuals. For the cold season, the scatter points can be split into two distinct regions along the vertical line where the ratio is 1. The scatter points in the left region indicate basins where cold-season precipitation is lower than in the warm season, leading to relatively smaller absolute residuals (clustered around zero residuals). In contrast, scatter points for basins with dominant cold-season precipitation are dispersed below the zero residual line, with lager negative residuals becoming more prevalent as the proportion of cold-season precipitation increases. In other words, regions where cold-precipitation constitutes a larger proportion of the water budget residuals are more sensitive to the underestimates of precipitation, resulting in larger negative residuals. Furthermore, we observed similar trends in the warm season, where a higher proportion of warm-season evaporation is associated with larger positive residuals. These results confirm the perspective of previous research, highlighting the potential uncertainties in measurements of cold-season precipitation and warm-season evaporation."

[Figure]

**Figure 12.** Relationship between the ratios of evaporation and precipitation for the cold and warm seasons separately and the corresponding water budget residuals. Note that blue represents residuals for the cold season, and red represents those for warm season. The seasonal division are consistent with Fig.5. The unit of residuals is "mm".

**C/ From Fig. 6, it appears that the Res and Resi have been effectively corrected, but the Reso have not changed significantly. Is this merely a specific case for this basin or a general situation? If it is a general situation, dose this imply that PHPM-MDCF only corrects for Resi and does not account for Reso? I believe that further explanation of this treatment could improve the transparency of the methods used in the paper.**

**R/** Yes, as you mentioned, our framework only corrects for Resi and does not account for the correction of Reso. This is a general situation for all basins. Essentially, such treatment is guided by the underlying logic of the correction process, as revealed by the residuals decomposition in Eq. 3. Reso is separated from the total water budget residuals to account for water components not considered in the original equation, such as inter-basin exchange.

From a causal perspective, this portion of residuals is less associated with physical inconsistency, as confirmed by the spatiotemporal distribution difference (Fig. 4-5) discussed in Sect. 4.1. Therefore, the framework focused on constraining residuals using physically consistent hydrological model cannot correct this part of residuals. This also explains why Resi decreases significantly after correction in Fig. 6, while Reso remains unchanged.

In addition, the discussion in Sect. 4.2 also highlighted this issue:

*"However, despite recalibrating the model with corrected datasets, $Res_o$ driven by the omission in water budget equation exhibited no substantial changes before and after correction (e.g., the monthly mean absolute values maintain around 6.5 mm, see Fig 6f). This phenomenon occurs because we only corrected the inconsistency residuals with reference to the simulation system, while the omission accounting for addition water terms should not be corrected in the existing datasets."*

In our opinion, using measurements to describe the theoretical water balance requires two key conditions: (1) physically consistent measurements, and (2) comprehensive description of the water budget equation. However, this is challenging to achieve in practice, whether due to inadequate understanding or limitations in measurement techniques, resulting in residuals corresponding to Resi and Reso.

The framework proposed in this work can, to some extent, enhance physical consistency between measurements through the model, resulting in reduced Resi. However, achieving a more comprehensive description (i.e., reducing Reso) may involve more issues, such as scale effects, more detailed data (both surface and subsurface), and a deeper understanding of the watershed. Addressing these questions is beyond the scope of this study. We look forward to more detailed future research addressing these issues, as mentioned in our discussion:

"*Further investigation would be required to better understand the omission residuals from a physical perspective. For example, a distributed hydrological model with representation of subsurface later flow process will allow us to identify the magnitude of inter-basin interactions; a more detailed description of water budget equation in data-rich environments can help us examine the sources of omission errors. This is undoubtedly important, but not the focus here.*"

Thank you again for your reminder. We recognize the difficulties caused by insufficient explanations and will further emphasize this issue in the revised manuscript.

**C/ Although the author has clearly articulated the main scientific problem of the paper, there are still areas that could be further improved, which I have listed in the specific issues.**

**R/** Tank you for your thorough and detailed review. We have addressed each point and provided responses below.

**Specific issues**

**C/ Line 22-25: According to the results, it seems that humid/wet basins are also prone to larger closure residuals, which needs to be emphasized here.**

**R/** According to your suggestion, we will revise the phrasing to:

"This emphasizes the importance of carefully evaluating the water balance assumption when employing multisource datasets for hydrological inference in small and humid basins."

**C/ Line 36-46: I believe this section should place greater emphasis on the issues of scale mismatch and difficulty in obtaining reference data.**

**R/** Thanks for your comment. We will revise the manuscript to strengthen the issue of scale mismatches and the challenges associated with obtaining site data. The following statement will be added:

"The issue of scale mismatches and the availability of site data in certain regions also pose challenges for data evaluation."

**C/ Line 58-60: It is recommended to cite the review by Beven (2002).**

**R**/ Thanks for your suggestion, we will include this reference to the following sentence:

"Such inconsistency poses an obstacle to robust hydrological inferences (Beven, 2002)."

**C/ Line 83-84: It is recommended to add references to support the argument.**

**R**/ We found supporting evidence in the literature Luo et al. (2023) and will include this reference to substantiate our argument.

Luo et al. (2023): *"therefore, the results confirm that increasing the water budget closure accuracy of budget-component data sets reduces the accuracy of individual budget-component products."*

"In the context of applying such closure constraint, it becomes evident that the precision of certain individual components may notably deteriorate, particularly when uncertainties are challenging to quantify (Luo et al., 2023)."

**C/ Line 119: "Res" does not appear to be in italics.**

**R**/ Thank you for your reminder. This formatting issue will be corrected throughout the revised manuscript.

**C/ Line 126-127: It is recommended to change it to: "(a) How can the total water budget residuals be quantitatively decomposed into inconsistency and omission residuals based on Eq. (3)?"**

**R/** Thanks for your careful review. We will revise the sentence according to your suggestions:

"(a) How can the total water budget residuals be quantitatively decomposed into inconsistency and omission residuals based on Eq. (3)?"

**C/ Table1: The "period" should be "Original Period".**

**R/** Thanks, we will correct this.

**C/ Figure5: The figure caption seems to contain an error. There are no other subfigures.**

**R/** The caption of this figure does indeed contain errors due to update to the figure, and we will revise it accordingly (i.e., remove redundant subplots sequence numbers):

"**Figure 5.** Temporal distribution of monthly water budget residuals ($Res$), inconsistency residuals ($Res_i$), and omission residuals ($Res_o$) across 475 CAMELS basins with reliable simulations. Boxplot-like diagrams describe variability across catchments, and outliers represent the 10th and 90th percentiles. The unit of residuals is "mm"."

**C/ Line 332-334: The argument here doesn't seem to correspond with the figure. Could it be that the figure has been updated?**

**R/** Thank you for pointing out this error. We will revise the statement while updating the figure caption (remove redundant subplots sequence numbers):

"On the contrary, $Res_o$ tends to be mainly positive except from September to November; its extent of variability is also significantly smaller than that of the other two residuals. In regard to magnitude, $Res_i$ is greater than $Res_o$, whether considering positive or negative bias."

**C/ Line 418: add "which are" before "implemented".**

**R/** Thanks, we will correct it to:

"This is achieved by the representation of physical hydrological processes underlying the correction strategy, which are implemented in the hydrological model, such as runoff generation and routing."

**C/ Line424-426: Change the sentence to "The fact that simultaneous corrections of other variables during extreme runoff noise corrections did not significantly differ from OS-based corrections further enhances our confidence in PHPM-MDCF."**

**R/** Thanks for your comment. We will revise the statement according to your suggestion.

"The fact that simultaneous corrections of other variables during extreme runoff noises correction did not significantly differ from OS-based corrections further enhances our confidence in PHPM-MDCF"

**C/ Line 417: It is necessary to further emphasize the issue of the non-closure phenomenon in humid regions.**

**R/** Thank you for your suggestion, we will revise the entire manuscript to emphasize the issue of non-closure phenomenon in humid regions. Below are several examples of the revisions will be made:

Abstract: "This emphasizes the importance of carefully evaluating the water balance assumption when employing multisource datasets for hydrological inference in small and humid basins."

Sect. 4.4: "These results highlight the risks of using multisource datasets for hydrological inference in humid and small-scale basins—specifically, potential physical inconsistencies—and underscore the need to carefully test the water balance closure assumption."

Conclusion: "This highlights the need for careful consideration of the water balance assumption when applying multisource datasets for hydrological inference in small and humid basins."

**C/ Figure 12: There seems to be a mistake with the R2 values.**

**R/** Thank you for pointing this mistake. We will correct this mistake in the revised manuscript. The updated figure with corrected R2 values is shown below:

[Figure]

**Figure 13.** Comparison of correction results based on different forcing datasets (TRMM and Daymet) at basin 1013500. (a-b) Corrected time series of five water budget variables. (c-e) Variation of long-term mean absolute values of three residuals with correction iterations at the monthly scale. The unit of residuals is "mm".

**C/ Line 639: Humid regions is a better expression.**

**R/** Thanks for your suggestion. We will make revisions throughout the entire manuscript. Here is an example:

"This highlights the need for careful consideration of the water balance assumption when applying multisource datasets for hydrological inference in small and humid basins."

**Reference**

Abolafia-Rosenzweig, R., Pan, M., Zeng, J., and Livneh, B.: Remotely sensed ensembles of the terrestrial water budget over major global river basins: An assessment of three closure techniques, Remote Sensing of Environment, 252, 10.1016/j.rse.2020.112191, 2020.

Kauffeldt, A., Halldin, S., Rodhe, A., Xu, C. Y., and Westerberg, I. K.: Disinformative data in large-scale hydrological modelling, Hydrology and Earth System Sciences, 17, 2845-2857, 2013.

Luo, Z., H. Li, S. Zhang, L. Wang, S. Wang, and L. Wang (2023), A Novel Two-Step Method for Enforcing Water Budget Closure and an Intercomparison of Budget Closure Correction Methods Based on Satellite Hydrological Products, Water Resources Research, 59.

Lv, M., Ma, Z., Yuan, X., Lv, M., Li, M., and Zheng, Z.: Water budget closure based on GRACE measurements and reconstructed evapotranspiration using GLDAS and water use data for two large densely-populated mid-latitude basins, Journal of Hydrology, 547, 10.1016/j.jhydrol.2017.02.027, 2017.

Newman, A. J., Clark, M. P., Sampson, K., Wood, A., Hay, L. E., Bock, A., Viger, R. J., Blodgett, D., Brekke, L., Arnold, J. R., Hopson, T., and Duan, Q.: Development of a large-sample watershed-scale hydrometeorological data set for the contiguous USA: data set characteristics and assessment of regional variability in hydrologic model performance, Hydrology and Earth System Sciences, 19, 209-223, 10.5194/hess-19-209-2015, 2015.

Robinson, E. and Clark, D.: Using Gravity Recovery and Climate Experiment data to derive corrections to precipitation data sets and improve modelled snow mass at high latitudes, Hydrology and Earth System Sciences, 24, 1763-1779, 10.5194/hess-24-1763-2020, 2020.

---

## Author Comment (AC2)

**Response to Reviewer RC2**

**Title**: Achieving water budget closure through physical hydrological processes modelling: insights from a large-sample study
**Authors:** Xudong Zheng, Dengfeng Liu*, Shengzhi Huang*, Hao Wang, Xianmeng Meng
**Manuscript ID**: hess-2024-230

**Reply on RC2:**

Thank you very much for dedicating your time and effort to reviewing our paper. All comments from Reviewer RC2 are addressed below with point-by-point responses.

For better readability, replies will start with "**R/**", following the original comments that start with "**C/**" and are shown in **bold**. The revisions to be added into the revised manuscript is highlighted in red. The important parts are highlighted in blue. The quoted content is displayed in *italics*.

**Point-to-point response:**

**C/ The paper presents an interesting concept, and its organization and writing are well done. However, I have some differing views regarding the underlying assumptions and principles of the proposed method. My main comments are as follows:**

**R/** First and foremost, we sincerely appreciate your interest in the concept shared in our paper, as well as your kind recognition of our writing and organization. We hold your constructive comments in high regard and believe it will be instrumental in enhancing the quality of our paper. These comments will be addressed point by point below, and revisions will be made in the manuscript to the best of our ability.

**Major Comments:**

**C/ (1) I do not agree with the two underlying assumptions of the PHPM-MDCF method, nor with the significance of using Equation 4 to calculate omission errors. My main reasons are as follows:**

**Firstly, the errors in hydrological models are non-negligible and represent the sum of both omission errors and data errors, rather than omission errors alone. The paper assumes that hydrological models have no data errors (inconsistency errors) and only omission errors, which is evidently unreasonable. This assumption is particularly problematic because hydrological models are typically validated against observed runoff, often neglecting the validation of ET (Evapotranspiration) and TWSC (Terrestrial Water Storage Change) simulation accuracy. As a result, using Equation 4 to calculate omission errors is not justified. Due to the complexity of hydrological models and the impact of errors in driving variables, the water imbalance caused by errors in the hydrological model may be substantial. Even if the inputs to the hydrological model are observational data and the model itself is developed based on the principle of water budget, the primary contributor to water imbalance errors between input and output might still be data errors.**

**Secondly, the total residual is calculated using multiple sources of data, and omission errors are calculated using data that drive the hydrological model as per Equation 4. The difference between these is then used to calculate data inconsistency errors. However, this approach might introduce uncertainties due to data inconsistency.**

**R/** Thank you for your comment. We acknowledge that employing hydrological models to constrain measurements and thereby enhance water budget closure among them is an ambitious idea, as it has not been previously presented in the literature. We also recognize that accepting this idea is challenging. However, this idea is not proposed arbitrarily; rather, it is developed progressively along a specific logical path.

First, the errors in hydrological model that we describe as ignorable refer to inconsistencies occurring within the input, output, and state, rather than those between measurements. This distinction is important to emphasize. In other words, each variable in Eq. (4) originates from the model itself, and from this perspective, these variables are independent of measurements. Such consistency in hydrological model has been described in numerous studies. For example, DeChant and Moradkhani, (2014) provided reduced structural equations for general distributed hydrological models from a state-space view:

$$s_{i,t} = f(x_{i,t}, s_{i,t-1}, \theta_i), \tag{R1}$$

where $f()$ represents the model structure, $x_{i,t}$ is the forcing of the $i$th grid at time $t$. $\theta_i$ is the parameter of the $i$th grid. In this equation, a quantitative balance is maintained between the input/forcing and output/state variables. In the general hydrological models, whether distributed or lumped, water balance serves as a fundamental governing equation to constrain the model, which is a well-established practice (Beven., 2001). The above constitutes the logical basis for our assumption that the hydrological model satisfies water balance, ensuring physical consistency. This also aligns with our definition of inconsistency residuals, which refer to non-closure arising from physical inconsistency.

However, given our current understanding of the water cycle, Eq. (4) may still be prone to omission residuals. It can be challenging to be aware of all water components, certain omissive components result in omission residuals. This portion of the residuals can be identified through variables derived from the hydrological model, as these variables are consistent with water balance.

In extreme cases, if all components are considered in water budget equation, the omission residual can be reduced to zero. At this point, no water imbalance exists within the simulation system (i.e., Eq. (4)), and any remaining residuals in the measurement system would be the potential inconsistency residual.

Return to your question, the "data errors" you refer to are more likely the differences between simulated and measured values (e.g., simulated versus gauged runoff). This pertains to model performance, specifically whether the model can accurately represent hydrological process. This does not conflict with the water balance feature of the model itself. It is important to emphasize once again that all variables used in Eq. (4) are derived from the model, not from measurements.

I hope the above response provides some clarity on the issues related to water balance in the hydrological model and the potential neglect of inconsistency residuals in Eq. (4). In addition, we would like to further address the question of the relationship between measurements and simulations in this method. We believe that clarifying this point may help address your concerns.

In the PHPM-MDCF method, measurements are used not only calculate the total residuals (i.e., Eq. (5)), but also to constrain the model through a multi-objective calibration process (i.e., tuning parameters). As you emphasized, using only observed runoff to validate the model is insufficient. In this work, we considered five different variables—streamflow, ET, SMS (soil moisture storage), GRS (groundwater reservoir storage), and SWE—to validate the performance of the model. After model performance evaluation, we selected 475 basins with reliable simulation for all variables for subsequent analysis. The first paragraph of Sect. 4.1 and Appendix C provide detailed information. We present the main information here:

*"To ensure the robustness of the results, as mentioned previously, it is essential that hydrological model reliably represent hydrological processes. With reference to previous studies (Clark et al., 2021), we have adopted KGE≥-0.41 and r statistically significant at the 5% level as criteria for guaranteeing reliable simulations. The multi-objective simulation performances of the HBV model are detailed in Appendix C. In general, the majority of basins (475, accounting for 72.24% of the total basins) achieved reliable simulations across all variables."*

[Figure]

**Figure C1.** The multi-objective simulation performances of the HBV model across the CAMELS basins. Results are based on (a) runoff, (b) evaporation, (c) soil moisture storage and groundwater reservoir storage, and (d) snow water equivalent. Red dots represent unreliable simulation performance, and the size of points is proportional to the basin area. The unit of RMSE is "mm".

In general, this helps ensure simulation accuracy to some extent and reduces the uncertainty in the residual decomposition. Furthermore, the multi-objection calibration process is repeatedly applied during multisource datasets correction to ensure that, after each iteration of data correction, the model can produce reliable simulations corresponding to the dataset.

Based on the response to this concern, we recognize the importance of further emphasizing the water balance assumption in hydrological model used in this method, particularly with respect to Eq. (4). Therefore, we will add the following statements to the manuscript (Sect. 3.1):

"It is crucial to clarify that all variables in Eq. (4) are derived from the model itself, rather than from measurement, and can therefore be considered physically consistent."

**C/ (2) The validation of results should include a comparison between the PHPM-MDCF method and existing methods. The paper repeatedly emphasizes the inadequacy of current methods in distributing residuals, yet no comparison with existing methods is provided in the results to verify the accuracy of the PHPM-MDCF method. The goal of closing the water budget is to reduce residuals while improving the accuracy of water cycle variables. Therefore, the credibility of the model should not be judged solely by the reduction of residuals (Figure 6). A comparison with existing methods would be more convincing. I strongly recommend supplementing the results with a comparison against existing correction methods, particularly CKF, PR, and MCL methods. For instance, the accuracy of the datasets after calibration using these methods, including P (Precipitation), ET (Evapotranspiration), Q (Runoff), and TWSC (Terrestrial Water Storage Change).**

**R/** Your point is very logical and intuitive. The introduction of any new method inevitably involves comparison with existing methods, which was also one of our initial objectives. However, after a thorough process of reflection and analysis, we have found that a direct comparison with existing methods is either infeasible or not meaningful for the following reasons:

(a) Difference in underlying logic.

The PHPM-MDCF exhibits a fundamental difference from existing methods, particularly in its interpretation of the realism. In existing methods, such as CKF, PR, or MCL, data correction relies on an assumed "true value" as reference. This true value might be the ensemble mean of multiple products or a set of gauged observations considered more credible. In other words, they assume that this true value can represent reality. However, this assumption is often challenged by issues such as scale mismatches and systematic biases in products. Although this approach is a common practice, but in our opinion, this notion of realism maybe untenable.

As an alternative, the realism of the PHPM-MDCF is reflected in our understanding of physical processes. Throughout the data correction process, the physical hydrological processes represented by the hydrological model play a central role. They act as constraints, iteratively correcting measurements into a physically consistent system. Although the reality represented by the hydrological model remains an abstraction, the iterative coupling of information in measurement through parameter calibration enhances the confidence in this representation. This approach embodies the underlying Bayesian philosophy. In addition, the pre-selection of basins with reliable simulation reduces the uncertainty associated with this method.

Due to the differences in these notions of realism, comparing these methods appears to be of limited value. The former typically aims to correct to an assumed "true value", while the PHPM-MDCF focuses on correcting measurements to the physically hydrological processes represented by the model.

(b) Lack of real true values for reference.

As noted, the differences in realism among the methods make direct comparison challenging. Thus, the question arises whether an objective true value can be obtained as a benchmark for comparing the accuracy of different correction method, as mentioned in the comment. The answer is no. As we discussed in the induction:

*"the fact remains that the 'true value' is perpetually unattainable, rendering any form of reference*

*data uncertain"*

Since water budget non-closure study typically focus on datasets from different sources that estimate across varying scales, even when using field observations as the reference, there are challenges with scale mismatches. Additionally, even without scale mismatch issues, acquiring such data across extensive spatial and temporal scales remains a significant challenge.

(c) Different understanding of the relationships between measurements of different variables.

Another reason we cannot directly compare existing methods with PHPM-MDCF is the difference in their understanding of the relationships between observations of different variables.

In the data fusion based correction methods (e.g. CKF, PR, MCL), the physical connections between different variables seem to be overlooked. Or more cautiously, these relationships are not explicitly utilized as constraints for data correction. In such cases, although residuals can be constrained to zero, the correction process might disrupt the physical connections between variables, leading to unreasonable adjustments. In contrast, the PHPM-MDCF leverages these relationships, as represented by hydrological model, to constrain the measurements.

This difference ultimately reflects in the correction results. As noted by Luo et al. (2023), correction may lead to a decrease in the accuracy of individual variables:

*"therefore, the results confirm that increasing the water budget closure accuracy of budget-component data sets reduces the accuracy of individual budget-component products."*

For the above reason, we think the direct comparison between our method and existing methods is not meaningful, as their correction direction are fundamentally different.

Although we cannot conduct a direct comparison, but a theoretical indirect analysis is possible. The noise experiment in Sect. 4.3.2 can provide such an indirect analysis.

When extreme single-point noise is present in streamflow measurement (NS1 and NS2), it is expected that, to ensure water balance closure, existing correction methods will impose constraints across all variable by referencing "true values". Typically, streamflow measurements are considered to have the least uncertainty, leading to the smallest correction. As a result, extreme bias in streamflow can propagate to other variables by correction process, such as ET and TWSC. This is also the reason why the correction process, as previously discussed, can lead to a reduction in the accuracy of individual variables.

Figures 9 and S9 indicate that the PHPM-MDCF can effectively reduce residuals without causing such bias to propagate across different variables, thereby avoiding the aforementioned issues. This indirect analysis also provides some explanation for the differences between PHPM-MDCF and existing methods, and, to some extent, supports its reliability.

Responding to your comment has stimulated further reflection on our part. This is highly valuable, and we will make the following revisions according these reflection:

(a) We will further emphasize the issues of scale mismatch and the availability of site data in the induction:

"The issue of scale mismatches and the availability of site data in certain regions also pose challenges

for data evaluation."

(b) We will include the reference by Luo et al. (2023) to strengthen the expression of our viewpoints:

"In the context of applying such closure constraint, it becomes evident that the precision of certain individual components may notably deteriorate, particularly when uncertainties are challenging to quantify (Luo et al., 2023)."

**C/ (3) The description of the reference datasets is unclear. It is necessary to specify which observational system datasets were used for P (Precipitation), ET (Evapotranspiration), Q (Runoff), and TWSC (Terrestrial Water Storage Change), and why these datasets can be considered observational data. I recommend clarifying this in the text.**

**R/** Thank you for your suggestion. We will revise Table 1 in accordance with your suggestions and provide the explanation for the selection of these datasets for each variable. Here is the revised version:

"Specifically, daily precipitation estimation derived from the Tropical Rainfall Measuring Mission (TRMM 3B42V7) is used in this study. The well-known international NASA project aims to comprehensively estimate all forms of precipitation, including rain, drizzle, snow, graupel, and hail, through the integration of satellite data and ground-based rain gauge measurements (Huffman et al., 2016). The accuracy of TRMM dataset has validated by many studies through comparisons with observation data and other reanalysis datasets (Kittel et al., 2018; Villarini et al., 2009). For evaporation, we utilized the third version of Global Land Evaporation Amsterdam Model (GLEAM v3) product (https://www.gleam.eu/), which employs a set of algorithms to separately estimate the different components of land evaporation (Miralles et al., 2011). Several studies have demonstrated that this product aligns well with flux measurements and multisource product ensemble (Munier et al., 2014; Robinson and Clark, 2020). And, as mentioned above, the runoff measurements on a basin scale are provided by the CAMELS dataset, which is derived from site observations."

**Table 1.** Overview of the products for constructing water balance equation used in this study.

| Variable | Product | Original Resolution | | Original Period | Reference |
| | | Spatial | Temporal | | |
| --- | --- | --- | --- | --- | --- |
| Precipitation | TRMM 3B42V7 | 0.25 °×0.25 ° | Daily | 1998-2019 | *Huffman et al. (2016)* |
| Evaporation | GLEAM v3.8a | 0.25 °×0.25 ° | Daily | 1980-2022 | *Martens et al. (2017)* |
| Soil moisture layer 1/2/3/4 | EAR5 Land | 0.1 °×0.1 ° | Hourly | 1950-present | *Muñoz Sabater et al. (2021)* |
| Snow water equivalent | GlobSnow v3.0 | 25km×25km | Daily | 1979-2018 | *Luojus et al. (2021)* |
| Streamflow | CAMELS-USGS | Basin scale | Daily | 1980-2010 | *Newman et al. (2015)* |

**C/ (4) Only a single product was selected for each water cycle variable. I believe that selecting multiple products is crucial for validating the proposed PHPM-MDCF method. This is because different datasets have different sources of error, leading to varying inconsistency residuals depending on the data combination. If the proposed method can be used to identify inconsistency residual error, using multiple data combinations would better verify the reliability of the proposed**

**method in this study.**

**R/** Thank you for your comment. We acknowledge that a common practice in previous water budget assessments is to use a range of products for each water components, evaluating the availability of different product combinations to closure the water budget. For example, Lorenz et al. (2014) compared 180 combinations of datasets for P, ET, TWS, and Q to access the degree of atmospheric-land water balance achieved. Lehmann et al. (2022) investigated the budget closure at catchment scales using 11 P, 14 ET, and 11 Q datasets together with GRACE.

However, almost all similar studies have reached the same conclusion that no single combination can close the water budget well across all regions (Lv et al., 2017). This implies that while introducing multiple products for ranking may be meaningful for specific regions, it holds limited significance for the correction framework of this study, which focuses on broader spatial scales (large sample basins). As Petch et al. (2023) handled in their optimization-based correction method, a single product was used for each water budget component, and they emphasize:

*"In this study, we use only a single data product for each component, which we account for in our uncertainty calculations. We aimed to use Earth observation data where possible and sought global gridded products to ensure the uniformity of the uncertainties across all basins."*

*"Overall, the specific datasets chosen were not critical, as our primary goal was to evaluate our new optimisation methodology and its ability to bring independent products into consistency."*

In addition, different products process varying spatiotemporal scales and have regional applicability, incorporating additional product may introduce further uncertainty.

A possible realization in the current study is to use different precipitation datasets (i.e., TRMM and Daymet datasets) to force the hydrological model and conduct correction, which has been implemented in Sect. 5.2.1. The results indicated that the correction is not sensitive to the choice of precipitation data.

*"In summary, the above results suggest that the correction is minimally sensitive to the choice of forcing, demonstrating the robustness of the correction results."*

For the reasons mentioned above, we think that introducing additional products in the current study may not be necessary. However, we look forward to applying more models and datasets in future research to further extend the framework.

**C/ (5) In Step 2 at line 250, please explain why is it reasonable to allocate residuals based on the difference between simulated values and reference values? It is worth noting that the simulated ET (Evapotranspiration) and TWSC (Terrestrial Water Storage Change) by the hydrological model may not have been validated for accuracy and may contain significant uncertainties. If their errors are used to allocate residuals, substantial uncertainties could lead to unreasonable allocation of residuals to ET and TWSC. The formula for residual allocation needs to be supplemented. Additionally, if Step 3 determines that the residual allocation is unreasonable, can simply halving the residual solve the issue? The underlying principles need to be clarified, or an example should be provided.**

**R/** Thank you for your careful review. For clarity, we have reorganized the questions in this comment and will analyze them individually.

**(a) Why allocate residuals based on the distance between measurements and simulations?**

As we discussed earlier, in this study, the simulations from the hydrological model are considered a physically consistent system that satisfies the water balance (See the reply to major concern (1)). Therefore, the Eq. (4) based on the simulations inevitably leads to $Res_i$ being 0. In other words, when all measurements are corrected to equal the simulations, the $Res_i$ in the measurements are corrected to 0. This determines the correction direction for measurements of each variable.

However, directly correcting the measurements to equal the simulation at once can also introduce uncertainty, as the simulation system is not precise (i.e., model parameters). Therefore, we considered an iterative approach for correction.

From the perspective of hydrological processes, the simulations reflect an ideal system that is physically consistent and strongly physically interrelated. On the contrary, the measurements reflect a system that variables are relatively loosely connected and physically inconsistent. To facilitate the convergence of the measurement system towards the ideal simulation system, it is important to determine the relative magnitude of the corrections for each water component.

The different water components cannot be corrected to the same extent, as their physical connections must be taken into account. For example, consider a region with high evaporation and low streamflow. Typically, it is reasonable to apply more correction to evaporation. However, if measurement of streamflow exhibits extreme high values, it would be more reasonable to apply more correction to streamflow. This is because our understanding of hydrological process suggests that the likelihood of such extreme high streamflow in this region is very low. Such understanding is reflected in the hydrological process, that is, in the simulations. Given this, we allocate the correction of $Res_i$ based on the distance between measurements and simulations. In other words, the greater the distance between the measurement and the expected values, the more correction we will apply. This idea is illustrated in Fig. 3.

[Figure]

**Figure 3.** Illustration of the correction process advancing convergence between the simulation and measurement systems.

To better assist readers in understanding this idea, we will revise the statement in Step2 to:

"Step 2: Correction for the inconsistency residuals. Allocate inconsistency residuals based on the magnitude of differences (i.e., the distance between simulation and measurement systems) between simulated and measured values for each variable in Eq. (5) and (6). This is because this difference indicates the correction direction and magnitude for each variable, thereby facilitating the convergence of the measurement system towards the simulation system. Here, an initial correction rate of 0.5 is set to gradually correct the multisource datasets, thereby avoiding potential uncertainties that arise from excessive correction. Formally, the allocation of inconsistency residuals can be described by the following equation.

$$M_c^v = M_o^v - Res_i \times \frac{d_v}{d_{all}}, \tag{7}$$

where $M_c^v$ is the measurements after correction of variable $v$, and $M_o^v$ is the original measurements; $d_v$ is the difference between simulation and measurement of variable $v$, and $d_{all}$ represents the aggregate of differences for all variables."

**(b) Were the simulations of ET and TWSC validated?**

Yes, we validated the simulation results across five variables (i.e., streamflow, ET, SMS, GRS, and SWE) to ensure reliable simulations, where the SMS and GRS are used to represent TWS. We have provided a detailed explanation in our response to Concern (1) above. Through model performance evaluation, we have ensured that all basins undergoing multisource dataset correction exhibit reliable simulation. Additionally, the simulation performance has significantly improved after correction, as evidenced by the changes in the Pareto front shown in Fig. 8.

[Figure]

**Figure 8.** Comparison of multivariable simulation performance before and after correction at basin 1013500. Light grey and dark grey indicate population solution sets before and after correction, and blue and red indicate Pareto fronts before and after correction. Metrics evaluating SWE simulation performance have been normalized for consistency. The subplot in the second row, second column shows that the evaporation simulation maintains highly accurate at this basin, due to the alignment between the HBV algorithm and measurements.

**(c) Supplement the residual allocation formula.**

Thank you for pointing out this. According tor your suggestion, we will add the corresponding formula as shown blow.

"Formally, the allocation of inconsistency residuals can be described by the following equation.

$$M_c^v = M_o^v - Res_i \times \frac{d_v}{d_{all}}, \tag{7}$$

where $M_c^v$ is the measurements after correction of variable $v$, and $M_o^v$ is the original measurements; $d_v$ is the difference between simulation and measurement of variable $v$, and $d_{all}$ represents the aggregate of differences for all variables."

**(d) If Step 3 determines that the residual allocation is unreasonable, can simply halving the residual solve the issue? What is the principle behind this?**

In Step 3, a judgment will be made to determine whether the previous correction was reasonable based on whether the model can provide a reliable simulation. A misunderstanding that needs to be clarified here is that if the simulation proves unreliable, we will discard the previous correction, return to Step 2, halve the correction rate rather than directly halving $Res$, and then proceed with the correction again. Naturally, after this correction, the judgment in Step 3 will be re-evaluated until the correction or inconsistency residual falls below a pre-set threshold.

In other words, this iterative process involves continual trial and error, with each error prompting us to approach the next correction more cautiously. The underlying consideration is that the convergence of the measurement system and the simulation system is a mutual process. Measurements approach the simulated system through correction, while the simulation system, through re-calibration after each correction, aligns more closely with the measurement system. As described in the process shown in Fig. 3 above. Excessive correction may lead to the measurement system going out of bounds, preventing further convergence of the two systems. Specifically, this manifests as producing unreliable simulations, and further model calibration will not enable the two system to converge.

We have noted that our expression might lead to misunderstandings; therefore, we will revise the phrasing in Step 3 to:

"Step 3: Calibration and evaluation of the model. Recalibrate and evaluate the hydrological model using the datasets corrected in the previous step to assess the reliability of this correction. If the recalibrated model yields unreliable simulations, consider this correction excessive, halve the correction rate, and repeat Step 2. Otherwise, maintain the correction rate and proceed with the next iteration of correction. The consideration behind this step is that excessive correction may lead to the measurement system going out of bounds, preventing further convergence of the two systems. In other words, the iterative process involves continual trial and error, with each error prompting us to approach the next correction more cautiously."

**C/ (6) Please clearly state the scope and spatiotemporal scale of this study. Most studies investigate water budget closure at the monthly scale rather than the daily scale. Aside from data availability, I believe this is mainly due to larger data errors and the lag effect of hydrological processes at the daily scale. If this study focuses on water budget closure at the daily scale, how were these issues addressed?**

**R/** Your perspective is very insightful. As you commented, the scale of the water budget study is crucial. The water budget non-closure phenomenon exhibits different behaviors at varying spatial and temporal scales. It is widely recognized that achieving water budget closure is much easier at relatively larger spatial and temporal scales.

On the one hand, at lager temporal scales, the TWSC exert a smaller influence on water budget closure. In relatively long time periods, TWSC can be assumed to negligible, making precipitation approximately equal to the sum of streamflow and evaporation. This is a common assumption in water budget assessment studies when TWSC measurements are unavailable. For example, Weligamage et al. (2023) suggested a 10-year period during which changes in water storage were considered negligible. Other several studies suggested that TWSC can be disregarded at the annual scale (Cooper et al., 2011; Kauffeldt et al., 2013; Hoeltgebaum et al., 2023). On the other hand, at larger spatial scales, inter-basin water exchanges can be considered negligible (Lv et al., 2017). Therefore, in most previous studies, it has been more feasible to conduct water budget studies at larger spatial and temporal scales. Additionally, another important reason for the choice of a monthly scale in much of the prior research is the reliance on GRACE TWSC measurements, which are only available at this temporal resolution.

In this study, TWSC is represented by a combination of observed soil moisture storage (SMS), groundwater reservoir storage (SMS), and snow water equivalent (SWE), avoiding the resolution constraints of GRACE TWSC, thus can be conducted at a daily scale. This is detailed in Sect. 2.2, where the main information is as follows:

*"Assuming that TWSC can be retrieved through a combination of different water storages, we obtained the four-layer soil moisture from ERA5 Land and Snow Water Equivalent (SWE) from GlobSnow to estimate overall TWSC. This approach has been implemented in the investigation of Hoeltgebaum and Dias (2023), yield a high consistency between estimated TWSC and GRACE observation (i.e., correlation coefficient exceeding 0.71). Another consideration in this method is that the decomposed TWSC products (i.e., soil moisture and SWE) can correspond to the results simulated by hydrological model, thereby allowing us to correct water budget residuals, as discussed later."*

*"Overall, all datasets were resampled to a daily time step, and then aggregated over basins through simple averaging to perform analysis of water budget closure on a basin scale."*

Although the primary temporal scale of this study is daily, we also performed statistical analyses at monthly and annual scales. For example, Figure 4-5 aggregate the residuals to the monthly scale to illustrate their spatiotemporal distribution. Figure 6 displays the correction results at daily, monthly and annual scales. This was done for both of visualization purposes and facilitating potential comparisons with previous studies.

Through a comparison of water budget at different timescales, we observed distinct behaviors of residuals across these scales. Specifically, at smaller scale (daily), residuals show greater variability but smaller magnitudes. As aggregation occurs at lager scales (monthly and annual), the magnitude increase while the variability decreases, demonstrating a filtering behavior. The primary mechanism behind such behavior is the positive and negative offset and accumulation of residuals and biases in different water components. Figure 6 provides an example to illustrate this:

[Figure]

**Figure 6**. Correction results of water budget residuals for multisource datasets at basin 1013500. (a-c) Time series of water budget residuals ($Res$), inconsistency residuals ($Res_i$), and omission residuals ($Res_o$) at daily, monthly and yearly scales, grey line represents residuals during the correction process. (d-f) Variation of long-term mean absolute values of three residuals with correction iterations at the monthly scale. The unit of residuals is "mm".

According tor your comment, we will further emphasize the temporal scale used in this study by adding the following statements in Sect. 3.1 and 3.2:

"Therefore, residuals are calculated at daily scale and subsequently aggregated to the monthly and annual scales for further analysis."

"Notably, the correction is performed at the daily scale, aligning with the model step."

**C/ (7) At line 320, it is necessary to explain the reasons behind the spatial distribution of Res. How does the difference in spatial patterns indicate that inconsistency residuals and omission residuals are driven by different factors? Please provide a detailed explanation. The most likely reason for Resi and Res having the same spatial pattern is that the former was calculated based on the latter. Their difference from Reso is due to the different error sources used in calculating Reso and Res, which does not necessarily demonstrate the reliability of the method for separating inconsistency residuals from omission residuals. Additionally, the residual values in Figure 4 differ significantly from those reported in previous studies. What is the reason for this discrepancy?**

**R/** Thank you for your comment. For clarity, we reorganized the questions in the comment into two separate points and address each one individually.

(a) **What are the reasons behind the spatial distribution of Res? Does its distribution show significant differences compared to previous studies? If so, what are the reasons for these differences?**

This is a good question. Indeed, as we discussed in our manuscript, the spatial distribution of $Res$ in Fig. 4 exhibits very pronounced clustering characteristics.

*"Res and Resí both present an east-west gradient for three statistical measures (i.e., min, median, max), with low values occur along the western coastline and high values primarily concentrated in eastern inland basins. The exception is a cluster of low median values located in the central CONUS"*

From a geo-statistical perspective, the spatial heterogeneity of *Res* likely involves multiple direct and indirect influences from basin characteristics. Clarifying these potential influencing factors is crucial for understanding the formation of *Res*. Therefore, we conducted an exploratory analysis in Sect. 4.4 and found that *Res* is closely related to basin area and hydro-meteorological conditions. Specifically, we found that achieving water budget closure with multisource datasets is more challenging in larger and humid basins (characterized by high precipitation and runoff coefficient). Figure 11 provide the corresponding evidence.

[Figure]

**Figure 11.** Relationship between the mean absolute of water budget residuals, basin area, long-term average daily precipitation, and runoff coefficient (RC) over 475 CAMELS basins with reliable simulations. The respective red lines represent the linear regression of residuals with basin area for each timescale.

Additionally, the comparison of the spatial distribution of *Res* with previous studies is also presented in Sect. 4.4. The results indicate that the pattern of *Res* identified in this study is consistent with previous research:

*"As shown in Fig. 4, all three water budget residuals are subject to strong spatial organization, and these patterns are in agreement with previous studies. For example, Kauffeldt et al. (2013) found negative residuals (i.e., runoff coefficient > 1) along the western coastline of CONUS, while the eastern region showed notable positive residuals (i.e., P-R > ET). Other studies investigating water budget residuals with diverse dataset combinations have similarly revealed similar spatial patterns (Zhang et al., 2016; Gordon et al., 2022)."*

We noticed a loose connection between Sect 4.1 and Sect 4.4; thus we will add the following statement in the former section to strengthen the linkage between the two sections:

"The potential factors affecting the spatiotemporal distribution of *Res* will be further investigated in

Furthermore, we will divide Sect. 4.4 into three subsections to ensure a clear structure. The titles of the three subsections are:

"4.4.1 Factors influencing spatial distribution"
"4.4.2 Factors influencing temporal distribution"
"4.4.3 Factors influencing the proportions of residuals components"

(b) **Why are the differences between the spatial patterns of Resi and Reso driven by different factors? What is the theoretical basis for residual decomposition? How can the reliability of this decomposition be demonstrated?**

In previous studies, $Res$ (water budget residuals) have typically been used as a whole to measure the degree to which the measurements achieve water budget closure. The cause of $Res$ is often simply attributed to inconsistencies in the processing of different products (refer to the review provided by Lv et al., 2017). Few studies have thoroughly discussed the causes of $Res$ formulation.

An exception is the study by Gordon et al., (2022), where they qualitatively decomposed $Res$ into data inconsistency error ($e$) and groundwater exchange ($G$) not accounted for in the water budget equation (see Eq. (2)). We extended Eq. (2) to incorporate additional source of potential water omission, and further attempted a quantitative decomposition of $Res$ into $Res_i$ and $Res_o$ to elucidate the distinct factors contributing to the observed water budget non-closure.

In our opinion, using measurements to describe the theoretical water balance requires two key conditions: (1) physically consistent measurements, and (2) comprehensive description of the water budget equation. Correspondingly, the causes of water budget non-closure ($|Res| > 0$) can be attributed to two factors: (1) physical inconsistency in the measurements ($Res_i$), potentially arising from discrepancies in data production process mentioned in previous studies; and the incomplete description of the water budget equation ($Res_o$).

Indeed, as you noted, the decomposition of $Res$ is fundamentally based on the following sample equation, which capture the essence of our decomposition method:

$$Res_i = Res - Res_o \qquad (R2)$$

However, the similar spatiotemporal distribution of $Res_i$ and $Res$ cannot be simply attributed the calculation. Essentially, this similar pattern is attributed to the relative small proportion of $Res_o$, suggesting that our description of the water budget equation is comparatively comprehensive.

Consider that if our description of the water budget equation were incomplete and omitted a significant water component, $Res_o$ would likely exert a greater influence on $Res$, resulting in a more pronounced discrepancy between $Res$ and $Res_i$.

To examine this, we intentionally exclude the SWE component from the water budget equation to access its impact on the decomposition of $Res$. This is a plausible scenario in practice, as it is likely that this component was not considered when reconstructing the TWSC. The results indicate that the proportion

of $Res_o$ obtained from residuals decomposition after excluding SWE increases significantly, with this effect being more pronounced in high-latitude regions, high elevations, and during the cold season (see the revisions and figure below). This is consistent with physical principles, as the impact of omitting SWE on water balance is greater under these situations. These findings align with our definition of $Res_o$ which refers to the water imbalance caused by omitted water. It also, to some extent, supports the validity of our decomposition method, and highlights the importance of a comprehensive water budget equation.

Based on the response to this issue, in order to further demonstrate the reliability of the residual decomposition, we will add a new subsection in Sect. 4.4 to explain the potential factors for the proportion of $Res$ components.

"4.4.3 Factors influencing the proportions of residuals components

Another interesting finding in Sect. 4.1 is that the magnitude of $Res_o$ is significantly smaller than that of $Res_i$. As a result, $Res$ is dominated by $Res_i$, leading to a highly consistent spatiotemporal distribution between them. However, the underlying question is what this implies and what factors drive the proportions of residuals components.

$Res$ reflects the degree to which the measurements achieve water budget closure. In this study, we argue that two key conditions are necessary for using measurements to describe theoretical water balance. The first one is that measurements of different water components must be physically consistent. In practice, however, this condition is often challenging to meet due to inconsistencies and uncertainties in data production processes from different sources, which can result in non-zero $Res_i$ (Luo et al., 2020). The second crucial, yet frequently overlooked, condition is the completeness of the water budget equation. Building on the work of Gordon et al. (2022), we developed a more generalized water budget equation (Eq. (3)) and use $Res_o$ to account for the water imbalances caused by omitted water. From this perspective, $Res$ results from the interplay between $Res_i$ and $Res_o$, either through their accumulation or mutual cancellation. Therefore, the low proportion of $Res_o$ essentially suggests that our description of the water budget equation is comparatively comprehensive.

Consider that if our description of the water budget equation were incomplete and omitted a significant water component, $Res_o$ would likely exert a greater influence on $Res$, resulting in a more pronounced discrepancy between $Res$ and $Res_i$. To examine this, we intentionally exclude the SWE component from the water budget equation to evaluate its impact on the decomposition of $Res$. This is a plausible scenario in practice, as it is likely that this component was not considered when reconstructing the TWSC. Figure 13 illustrates the comparison between $Res_o$ derived from the decomposition method excluding SWE (hereafter $Res_o^{NSWE}$), and its original values. It is evident that $Res_o^{NSWE}$ exhibits greater variability compare to the original values (i.e., with smaller minimum values and larger maximum values). The median differences indicate that the likelihood of increased omission residuals is higher after excluding SWE (Fig. 13b). Such differences indicate that omitting crucial SWE storage component results in a greater degree of water imbalance, and, as expected, this effect is more pronounce in high-latitude and high-elevation regions (Fig. 13d-f). Moreover, the spatiotemporal distribution of $Res_o$ has changed (Fig. S11-12). Notably, during the cold season (December to February), the proportion of $Res_o$ is much higher and exhibits s significant positive trend. These findings align with our definition of $Res_o$ which refers to the water imbalance caused by omitted water. It also, to some extent, supports the validity of our decomposition method, and highlights the importance of a comprehensive water budget equation."

[Figure]

**Figure 13.** Comparison of $Res_o$ obtained from residuals decomposition excluding SWE with the original values. (a-c) Spatial distribution of monthly mean $Res_o$ excluding SWE minus its original values. (d-f) Time series of $Res_o$ excluding SWE and its original values at the southern basin (02198100, 32.96 °N), northern basin (12358500, 48.33 °N), and high-elevation basin (07083000, elevation of 3.56 km) at monthly scale. The unit of residuals is "mm"

[Figure]

**Figure S11.** Same as Fig. 4, but for residuals decomposition excluding SWE

[Figure]

**Figure S12.** Same as Fig. 5, but for residuals decomposition excluding SWE

**C/ (8) In the multi-source dataset correction framework for achieving water budget closure, what is the rationale for setting the initial correction rate to 0.5? Why is the correction rate halved when the model produces unreliable simulations? Is there a potential proportional relationship between the adjustment of the correction rate and the magnitude of bias in unreliable simulations that could allow for more efficient correction rate adjustments? Additionally, what is the basis for setting the conditions for iteration and termination of the correction process as "the inconsistency residuals decreases to 10% of its initial value or the correction rate falls below 4%"?**

**R/** This is a very insightful comment. What you mentioned are precisely three key issues we encountered during the implementation process. Just in our response to the fourth question in Major Concern (5), the iterative process involves continuous trial and error to prevent over-correction and ensure that measurement remain within the appropriate range.

The first issue is determining the initial correction rate ($r_0$). At the beginning, to ensure a high correction speed, we set the initial correction rate to 1 and 0.7. However, for most basins, this often resulted in measurements exceeding a reasonable range after the first iteration of the correction, leading to unreliable simulations and unreasonable corrected measurements. Through experimentation, we found that 0.5 is a suitable initial correction rate, as it ensures that the first iteration of the correction is effective in most cases.

The second key issue is determining the decay rate of correction rate ($\Delta r$) following the occurrence of unreliable simulations. The generation of unreliable simulations suggests that the current correction is excessive. Effectively reducing the correction magnitude and re-correcting may further facilitate the convergence of measurement system with the simulation system. Linear decay is a conventional approach, which aligns with our perception. For example, reducing the correction rate by 0.1 or 0.2 each time. However, testing has shown that such linear decay results in excessively long correction times, making the application of the PHPM-MDCF across a wide range of basins (i.e., 475 basins) difficult. On the other hand, exponential decay can cause the correction rate to quickly fall into a small value range, thereby

reducing the correction efficiency. Given the above, we chose a multiplicative decay approach, where the correction rate is halved each time for re-correction. The results indicate that this approach is effective, as shown in the iterative process depicted in Figures 6 and S3-6. For illustration, we provide a case here:

[Figure]

**Figure R1.** The decline of $Res$ with the number of correction iterations for basin 1013500. The unit of residuals is "mm".

The final issue is determining when to terminate the correction, as this criterion significantly affects the final correction efficiency. Here we consider two points.

(a) The first is that the correction has achieved satisfactory results, with the final $Res$ being relatively small ($Res_t$). This threshold must be appropriately set; it cannot be too large, as this would indicate insufficient correction, nor too small, since the PHPM-MDCF, as a soft constraint, has limited correction capacity. An excessively small final $Res$ threshold could result in an infinite number of correction iteration. Based on comparative experiments, we believe that reducing it to 10% of the initial value is appropriate. As shown in Fig. R1, $Res$ stabilizes and no longer changes once it decreases to around 10% of the initial value (from 40 to 4 mm).

(b) The second point is that the correction rate should not be too small, as this would imply excessively low calibration efficiency. This is closely related to the initial correction rate and decay rate (here, 0.5 and halving, respectively). A threshold of 4% means that the correction will cease once the correction rate, decayed four times from 0.5 to 0.03125, is reached. This threshold setting is relatively subjective, but it has proven to be reasonable based on testing results.

Notably, although the parameters for the three issues mentioned above are set subjectively, the choice follow a certain logic and have passed a series of tests. At least, cautiously speaking, they are suitable for the current study area, as shown in Fig. 7. Further adjustments are possible, but they have minimal impact on the current results (based on some testing).

We will add the following statement in Sect. 3.2 to further emphasize the issues mentioned above.

"In addition, the parameters settings in the PHPM-MDCF (i.e., initial correction rate, decay rate of the correction rate, correction termination threshold) are appropriate for the current study area (Table S2). When applying this framework to different regions, additional adjustments and testing may be required."

**Table S2.** Summary of the parameters settings in the PHPM-MDCF.

| Parameters | Reference value | Reference range | Description |
|---|---|---|---|
| $r_0$ | 0.5 | 0.3~0.6 | Initial correction rate. |
| Decay approach | Multiplicative | Linear, exponential, and multiplicative decay | The method of reduction in correction rate following an unreliable simulation. |
| $\Delta r$ | 50% | 30%~70% | Decay rate of the correction rate. |
| $Res_t$ | 10% | 5%~20% | Correction termination threshold for inconsistency residuals. |
| $r_t$ | 4% | 1%~10% | Correction termination threshold for correction rate. |

**Minor Comments:**

**C/ (1) Please provide additional explanation on how Section 4.3.1 demonstrates the reliability of the PHPM-MDCF method.**

**R/** Thank you for your suggestion. We will add scatter plots comparing measurements and simulation before and after correction to further illustrate the convergence of the measurement and simulation systems, thereby demonstrating the reliability of the PMPH-MDCF method. The following revisions will be added to Section 4.3.1.

"More intuitively, Fig. S7 presents a comparison of measurements and simulations for each variable before and after correction. It is evident that the relationship between measurements and simulation is significantly strengthened after correction. This suggests that the PHPM-MDCF has the ability to enhance the convergence between the simulation and measurement systems, supporting the credibility of the correction results to some extent."

[Figure]

**Figure S7.** Scatter plots comparing measurements and simulation before and after correction at basin 1013500.

**C/ (2) The paper does not validate the accuracy of the Reso, Resi, and Res separation method in the results.**

**R/** Thank you for your comment. We have addressed this issue in detail in our response to the second question of Major Concern (7) and will include a new subsection to demonstrate the reliability of the residuals decomposition method. Please review the response above.

**C/ (3) At line 310, can KGE ≥ −0.41 really indicate that the hydrological model accurately represents the observed hydrological system?**

**R/** Thanks for your comment. The Kling-Gupta Efficiency (KGE) metric, introduced by Gupta et al. (2009), provides a method for achieving a balanced improvement of simulated mean, variability, and correlation (see Eq. B1). Many studies have demonstrated the effectiveness of KGE, which is currently a popular metric in hydrological modelling (Knoben et al., 2020; Clark et al., 2021). The KGE is bound by $(-\infty, 1]$ with 1 being the ideal value. For such a metric, it is challenging to give a benchmark value to determine whether the simulation is reliable. Thus, to ensure caution, we opt to reference previous literature for guidance. For instance, Aerts et al. (2022) use the -0.41 of KGE as the benchmark to evaluate the performance of wflow_sbm in simulating streamflow:

*"Ideal model performance has a KGE score of 1 and a KGE score of −0.41 is equal to taking the mean flow as a benchmark."*

Bruno et al. (2002) noted that a KGE of -0.41 serves as the threshold for no skill:
*"(KGE ∈ (- ∞, 1], optimal value = 1, no-skill threshold over mean flow as predictor = -0.41)."*

The notable example is Knoben et al. (2019), who, by comparing the NSE and KGE metrics, established a KGE value of -0.41 as the threshold for evaluating whether model simulations outperform the mean flow:

*"Here we show that using the mean flow as a predictor does not result in KGE = 0, but instead KGE =1- √2≈-0.41. Thus, KGE values greater than −0.41 indicate that a model improves upon the mean flow benchmark – even if the model's KGE value is negative."*

Based on the aforementioned literature, we used a KGE value greater than -0.41 as the threshold for reliable simulations. Although this threshold may still be somewhat subjective, evaluating simulation reliability across five variables (i.e., streamflow, ET, SMS, GRS, SWE) simultaneously can help mitigate this uncertainty.

For better address the question, we will include the above references in the manuscript.

"With reference to previous studies (Knoben et al. 2019; Clark et al., 2021; Aerts et al., 2022), we have adopted $KGE \geq -0.41$ and $r$ statistically significant at the 5% level as criteria for guaranteeing reliable simulations."

**C/ (4) In Figure 5, Reso is closer to 0. Can we attribute this to the principle of water budget in the development of the hydrological model, rather than merely to omission errors? Since Resi = Res - Reso, and Reso is relatively small, it is evident that the values and spatial patterns of Resi and Res are more similar. What does this imply?**

**R/** Thank you for your comment. Our response to the second question of Major Concern (7) provides some clarification on this issue. Specifically, $Res_o$ approaching zero indicates that our description of the water budget equation is relatively comprehensive and cannot be simply attributed to the water balance features of the hydrological model.

When the SWE component is omitted without changing the model, $Res_o$ increases significantly, with this effect being more pronounced at high elevations, high latitudes, and during the cold season (Fig. 13).

The equation ($Res_i = Res - Res_o$) is indeed the essence of our decomposition method, but it is not the sole reason for the similarity between $Res_i$ and $Res$. The fundamental reason lies in the completeness of the water budget equation description, which results in a smaller contribution of $Res_o$ to the formation of $Res$.

**C/ (5) Please explain from a theoretical standpoint why the PHPM-MDCF method has such advantages over previous methods: "It suggests that the soft constraints based on physical hydrological processes will not lead to compensatory errors, as seen in traditional methods due to the rigid allocation of water budget residuals.".**

**R/** Thank you for your suggestion. We will add the following statement to theoretically demonstrate the advantages of the PHPM-MDCF.

"From a theoretical perspective, the PHPM-MDCF assigns the weights of residual correction based on the distance between measurements and simulation for each variable. In the presence of a single extreme bias, the large distance between the measurement and simulation of the corresponding variable leads to a larger correction being applied to that variable, while the weights for other variables remain unaffected. However, in traditional methods, the correction weight for each variable remain constant over time, and the final residuals are constrained to zero. This leads to the propagation of extreme biases across different variables."

**C/ (6) I do not find this statement reasonable: "When the hydrological model calibrated against multiple variables measured by the multisource datasets and achieves reliable performance, we consider the simulation system approaching the measurement system.".**

**R/** Thank you for your comment. We will revise this inappropriate statement to:

"When the hydrological model calibrated against multiple variables measured by the multisource datasets and achieves reliable performance, we consider the water budget represented by the simulation and measurement systems to be comparable."

**C/ (7) At line 255, please clarify the data sources for the observed values of P, ET, Q, and TWSC used in this study. Without this information, it is difficult to judge whether the deviation between the simulation system and the measurement system is calculated reasonably.**

**R/** Thank you for pointing out the unclear aspects of our manuscript. According to your suggestion, we reiterated the data sources (see our response to Major Concern (3)) and will further emphasize them in this section as follows:

"In the subsequent application of the PHPM-MDCF, the measurements are derived from the data provided in Sect. 2.2."

**C/ (8) I personally feel that the discussion in Section 5.1 would be more effective if it were more closely aligned with the scope of this study.**

**R/** Thank you for your suggestion. According to your suggestion, we will enhance Sect. 5.1 with more arguments relevant to this study and reduce unnecessary statements. The revisions will be made are as follows:

Remove this sentence from the penultimate paragraph: "Although our current knowledge may not be entirely precise—for example, the depiction of hydrological processes in hydrological models may lack accuracy—it remains foundation upon which we can rely and strive to refine in the future."

The last paragraph will be revised to: "The proposed correction framework (PHPM-MDCF) capitalizes on this concept by iteratively advancing the convergence between the knowledge system (i.e., hydrological model and water balance equation) and the measurement system, thus enhancing the credibility of the measurements. Although our current knowledge may not be entirely precise—for example, the depiction of hydrological processes in hydrological models may lack accuracy—it remains foundation upon which we can rely and strive to refine in the future. Furthermore, several underlying concepts in this framework, such as residuals decomposition and advancing water budget closure through correction, aligns with a recent study (Wang and Gupta, 2024). They introduced a novel hybrid model (i.e., Mass-Conserving-Perceptron) and discussed its potential application, including the bias correction (lacking confidence for the measurements) and examination of non-observed interactions with the environment (corresponding to the omission errors). Therefore, coupling the PHPM-MDCF with hydrological models that provide stronger interpretability is a valuable and promising research effort, as it can offer insights into the physical attribution of water budget non-closure and enable more reasonable correction."

**C/ (9) The limitations discussed in Section 5.2 are not explained from a theoretical perspective. I hope that some convincing explanations can be supplemented from this standpoint.**

**R/** Thanks for your suggestion. We will add the following statement to Sect. 5.2 to further explain the theoretical basis of the adaptability to forcing datasets of the framework.

"Theoretically, the consistency of correction stems from two aspects. Firstly, it is attributed to the adaptability of hydrological model to the input data, specifically the calibration compensation capability

we described in the introduction (Wang et al., 2023). This enables the hydrological model to generate reasonable representation of hydrological process even with imprecise forcing. Secondly, as discussed in Sect. 4.3.2, the PHPM-MDCF serves as a soft constraint and utilizes the distance between measurements and simulations to allocate residuals correction, thereby mitigating the propagation of bias between variables. These two features ensure that stability of the correction, rendering it less susceptible to interference from uncertainties in the forcing datasets."

**C/ (10) The structure of the article lacks a keywords section. Please add keywords.**

**R/** Thank you for your careful review. According to the current HESS official template and guidelines, the keywords section is not a required option. Please the following URLs:

https://www.hydrology-and-earth-system-sciences.net/submission.html#templates
https://www.hydrology-and-earth-system-sciences.net/submission.html#manuscriptcomposition

**C/ (11) Please add references related to the water budget equation.**

**R/** Thank you for pointing out the omissions in our manuscript. We will add the relevant reference (Lehmann et al., 2022) for Eq. 1 as:

"For a closed basin, the water budget can be mathematically expressed as (Lehmann et al., 2022),

$$\frac{dTWS}{dt} = P - ET - R, \tag{1}$$

where $\frac{dTWS}{dt}$ is change in terrestrial water storage, P is precipitation, ET is evaporation, R is streamflow at the outlet."

**C/ (12) The text states "as illustrated in Fig. 3" but the caption provided is "Figure 3". The authors should ensure that all figure captions are consistent with the text descriptions. Please carefully check the rest of the article for similar errors and make the necessary corrections.**

**R/** Thank you for your careful review. For the abbreviation format, we referred to the official guidelines provided by HESS. Please see the following URL and explanation:

https://www.hydrology-and-earth-system-sciences.net/submission.html#figurestables

*"Figure composition: …*
*…*
*The abbreviation 'Fig.' should be used when it appears in running text and should be followed by a number unless it comes at the beginning of a sentence, e.g.: "The results are depicted in Fig. 5. Figure 9 reveals that."*

[revised manuscript text omitted]

---

## Author Comment (AC3)

**Response to Reviewer CC1**

**Title**: Achieving water budget closure through physical hydrological processes modelling: insights from a large-sample study
**Authors:** Xudong Zheng, Dengfeng Liu*, Shengzhi Huang*, Hao Wang, Xianmeng Meng
**Manuscript ID**: hess-2024-230

**Reply on CC1:**

Thank you very much for your interest in our paper and for taking time and effort to review it. All comments from Reviewer CC1 are addressed below with point-by-point responses.

For better readability, replies will start with "**R/**", following the original comments that start with "**C/**" and are shown in **bold**. The revisions to be added into the revised manuscript is highlighted in red. The important parts are highlighted in blue. The quoted content is displayed in *italics*.

**Point-to-point response:**

**C/ In the context of fast development of measurement techniques, it is our mission to develop methods to leverage the advantages of the measured variables and thus promote the hydrological simulation. This study is a valuable try, which proposed a multisource datasets correction framework, the PHPM-MDCF, to achieve water budget closure with calibration of various variables. This experiment was carried out in 475 COUNS basins, showing great potential to reduce the inconsistency residuals.**

**R/** We appreciate your recognition of the importance of our work. We hope that this paper can contribute to the data foundations in various fields, such as earth system science and hydrology, within the context of big data. Your comments are very valuable in enhancing the quality of our manuscript. Below, we will provide point-by-point responses to these comments and make the corresponding revisions in the manuscript.

**Major concerns:**

**C/ (1) There are PTRMM in both Eq. (5) and (6), then how do we reduce the inconsistency residuals brought by P in the water budget?**

**R/** This is a crucial point, but cannot be solved within the current framework. As we have assumed that "*(2) the uncertainties associated with the model forcing and structure can be considered negligible during the modelling process*" in the methods section. In fact, the PHPM-MDCF employs the distance between simulations and measurements to allocate residuals corrections among variables. As a forcing or boundary condition, precipitation cannot be corrected within this framework, or in other words, it cannot be simulated.

Nevertheless, we consider that the uncertainty, or residuals, in precipitation has a minimal impact on the correction of other variables measurements. Some evidences are provided in Sect. 5.2.1, where a comparison of correction results under different precipitation forcing (i.e., TRMM and Daymet) reveals that the correction shows minimal sensitivity to the precipitation forcing.

*"In summary, the above results suggest that the correction is minimally sensitive to the choice of forcing, demonstrating the robustness of the correction results."*

Theoretically, such behavior stems from the adaptability of hydrological mode to the input data, specifically the calibration compensation capability we described in the introduction (Wang et al., 2023). This enables the model to generate reasonable representation of hydrological process even with imprecise forcing.

However, can the current results offer any guidance or insights for precipitation correction? The answer is affirmative. It is the comparison of corrections with different precipitation presented in Sect. 5.2.1 that highlights the impacts associated with varying precipitation inputs. Starting from this point, we can discern some potential clues.

It is evident that different precipitation products do not impact the correction of inconsistency residuals (Fig. 14c-d) but do results in varying omission residuals (Fig. 14e). On the one hand, discrepancies in precipitation products are compensated by model calibration, result in similar representation of hydrological process and thus similar inconsistency residuals corrections.

[Figure]

**Figure 14**. Comparison of correction results based on different forcing datasets (TRMM and Daymet) at basin 1013500. (a-b) Corrected time series of five water budget variables. (c-e) Variation of long-term mean absolute values of three residuals with correction iterations at the monthly scale. The unit of residuals is "mm".

On the other hand, the precipitation products exhibit a systematic bias. In particular, Daymet reports significantly lower precipitation in this basin compared to TRMM (see Fig. S13). Such bias will manifest in the water budget equation, leading to different total input water volumes. Consequently, with the inconsistency residuals of other variables unchanged, maintaining the water balance would require an increase in $Res_o$ (Fig. 14e). Note that the $Res_o$ presented in Fig. 14e represents the mean of absolute values.

Fig. S13 will be added to our manuscript along with the corresponding explanation (Sect. 5.2.1).

"The comparison of the two precipitation products is presented in Fig. S13, where Daymet precipitation is significantly lower."

[Figure]

**Figure S1.** Comparison of TRMM and Daymet precipitation products.

Therefore, it can be inferred that, with other variables unchanged, TRMM precipitation demonstrates superior water budget closure compared to Daymet precipitation, which contains larger inconsistency residuals. This difference in inconsistency residuals is directly reflected in the variations in omission residuals after correction (Fig. 14e). In other words, this portion of the omission residuals (i.e., the difference between the two omission residuals after correction) can be directly corrected in the precipitation.

Note that not all omission residuals can be corrected in the precipitation data, as it still contains residuals from some unknown omitted water content. In other words, such correction must be relative and based on comparisons using multiple precipitation products, as the true values and perfect water balance equation are unattainable. Only through comparisons can the discrepancies in $Res_o$ arising from precipitation inconsistencies be identified.

To focus the attention of this paper on the PHPM-MDCF framework, we have not conducted actual experiments here. Instead, we will introduce this idea in the discussion section.

"It is noted that the PHPM-MDCF has limitations in addressing inconsistency residuals in forcing. The reasons are twofold. On the one hand, this is due to our neglect of uncertainties in the forcing, which, as indicated by the above analysis, appears to have limited impact on the correction for other variables. On the other hand, this is because the PHPM-MDCF allocates residuals based on the distance between simulations and measurement, while the forcing cannot be simulated within the hydrological model. In

this case, is there a potential to correct the inconsistency residuals in the forcing? Clues to this possibility are hidden in the above analysis. Systematic biases in precipitation products are directly reflected in the water budget equation, leading to different total input water volumes. Consequently, with the inconsistency residuals of other variables unchanged, maintaining the water balance would require an increase in omission residuals (Fig. 14e). Therefore, it can be inferred that, with other variables unchanged, TRMM demonstrates superior water budget closure compared to Daymet, which contains larger inconsistency residuals. In other words, the differences in omission residuals reflect the discrepancies in precipitation inconsistency residuals. This portion of the omission residuals difference can be directly corrected in the precipitation. However, it is worth noting that not all omission residuals can be corrected in the precipitation, as it still contains residuals from some unknown omitted water content. Such correction must be relative and based on comparisons using multiple precipitation products, as the true values and perfect water balance equation are unattainable. We will explore the approach in future work and extend the PHPM-MDCF based on this idea."

**C/ (2) This paper focuses on the terrestrial water balance (Eq. (1)). However, whether this framework is applicable to broader water balances, such as atmospheric water balance or local water balance, or if any adjustments are needed?**

**R/** The ideas in your comment are very interesting. Although this paper primarily focuses on terrestrial water cycle systems, exploring broader water balance applications is highly valuable for extending the scope of this research.

Through a review of the literature, we found several water balance equations designed for other systems. For example:

- The steady-state hydrological budget equation of the proglacial zone (Cooper et al., 2011):

$$W_{PZ} = W_P + W_R - W_E - W_{SSS} - W_{SR} \pm \Delta W_S, \tag{R1}$$

  where $W_{PZ}$ is the net proglacial water flux, $W_P$ is the precipitation water flux, $W_R$ is the channel recharge water flux, $W_E$ is the evaporation water flux, $W_{SSS}$ is the sub-surface seepage water flux, $W_{SR}$ is the surface runoff water flux, and $\Delta W_S$ is the change in water storage.

- The atmospheric water vapor budget with a focus on the oceans (Penning et al., 2021):

$$\frac{\Delta W}{\Delta t} = E - P - \nabla \cdot (vq), \tag{R2}$$

  where $W$ being the total column water vapor and $\nabla \cdot (vq)$ the moisture flux divergence.

- The coupled atmospheric–terrestrial water balance equation (Lorenz et al., 2014):

$$\frac{dW}{dt} + \nabla \cdot Q = ET_a - P, \tag{R2}$$

  where $W$ denotes the total column water content in the atmosphere and $\nabla \cdot Q$ is the net balance of moisture flux (i.e., moisture flux divergence).

Regardless of the water balance system under consideration, the key to applying the PHPM-MDCF is whether the utilized model can represent the components of the water budget equation. The core principle of the PHPM-MDCF is to characterize the physical relationships among water budget components

through the model, thereby imposing closure constraints on the measurements. As we noted in the last paragraph of Sect. 4.3.2:

*"The physical relationships among various water budget variables, as representation by the model, are also imposed onto the measurements through the correction process. This constitutes the core principle of PHPM-MDCF."*

In other words, the application of the PHPM-MDCF to more complex systems to conduct correction can be achieved by replacing the hydrological model (HBV) in the framework with other more suitable models that can output more variables, such as physically distributed models (VIC model; Liang et al., 1994), coupled models (WRF-TOPMODEL; Rogelis and Werner, 2018), or even deep learning models (MCP; Wang and Gupta., 2024).

We will add a statement at the end of Sect. 5.2.2 to emphasize this issue.

"By employing models that generate additional output variables, we can more comprehensively represent the water budget equation and extend the application of the PHPM-MDCF to more complex water budget systems."

**C/ (3) Uncertainty plays a crucial role, and this study qualitatively address the uncertainty associated with the model structure. A pertinent question is whether this uncertainty can be quantified. While we know that validating this uncertainty through multiple models may be both challenging and unnecessary within the scope of the current work, it would be valuable if the authors could suggest potential avenues for future research and development.**

**R/** Thank you for your insightful comment. We would like to address this question from the perspective of Bayesian philosophy. In practical Bayesianism, all models are inherently flawed, yet each model can be assigned a level of confidence that indicates the degree to which we trust it (Hoang, 2020). Only by considering more than one theory and model can we more effectively approach the truth. This is also the core idea of the Beven's Alternative Blueprint (Beven, 2002). As they stated:

*"Why should there be any expectation of a single 'real' description when the direct observation of the responses of the most important part of hydrological systems is quite beyond our current capabilities and will be until there is a dramatic improvement in the available geophysical techniques?"*

*"The fact that there may be no unique answer does not mean that the approach is not science or scientific. Indeed, such an approach has then the additional advantage that we will work more naturally with the many potential worlds of future (and therefore unknown) boundary condition scenarios and the uncertain predictions that should ensue (e.g. Cameron et al., 2001)."*

We strongly align with this scientific perspective. The "uncertainty" should be regarded as varying descriptions of the assumed "truth" and the associated confidence levels. Relying on a single theory alone is insufficient.

Due to limitations in data and resources, this study employs only one model for measurements correction, and we acknowledge that this introduces uncertainty. In future work, an effective method for quantifying

uncertainty is to use an "ensemble" approach. Specifically, employing multiple models (theories) to describe the same hydrological process enables the range of ensemble corrections to be used for quantifying uncertainty. This is very similar to the ensemble forecasting (Nicolle et al., 2014). Additionally, confidence can be assigned to each correction based on the simulation accuracy of each model, resulting in a unique weighted correction outcome.

The description of the possible approach will be added into the Sect. 5.2.2 as follows:

"Additionally, multiple models can be employed for 'ensemble correction', which aids in quantifying uncertainty and providing more robust correction results."

**Minor comments:**

**C/ (1) Please check carefully of the text, to avoid grammatic errors, e.g. km2 in Line 183.**

**R/** Thank you for your thorough review. We have conducted a comprehensive check and will make the revision. Below is an example of the revision made.

"However, the assumption is fragile when applied to small basin, leading to significant uncertainty in estimating TWSC for basins with areas less than 63,000 $km^2$ (Lehmann et al., 2022)."

**C/ (2) Line 75-76: The semantics are repetitive; it is recommended to delete "to ensure data consistency".**

**R/** Thanks for your comment. We will delete the redundant expression. The revised sentence is as follows.

"Other approaches, such as post-Processing Filtering technique (PF) and bias correction method (Munier et al., 2014; Weligamage et al., 2023), can also be helpful in closing water budget."

**C/ (3) Line 80: "residuals" is more precise than "bias".**

**R/** Yes, thank you for pointing that out. The revised sentence is as follows.

"However, the closure constraints imposed by the above methods (hereafter referred to as traditional methods) have been questioned, with Abolafia-Rosenzweig et al. (2020) arguing about the potential incorrect assignment of residuals."

**C/ (4) Line 131-134: It seems that these sentences should be changed to the past tense.**

**R/** Thank you for your comment. We will revise the sentence as follows.

"Furthermore, we developed a multisource datasets correction framework based on decomposition of water budget residuals and multi-objective calibration within hydrological modeling. The presented

framework, providing the capability to enhance the water budget closure and hydrological connections among multisource datasets, was applied to a large-sample basins dataset across CONUS."

**C/ (5) Line 165: "One of the main aims" might be more appropriate.**

R/ Thank you for your suggestion. We will revise the manuscript according to your suggestion.

"One of the main aims of this study is to investigate the decomposition of water budget residuals and correction to datasets, rather than comparing the differences and rankings of closure residuals across different dataset combinations."

**C/ (6) Line 167: This sentence should be in the past tense.**

R/ Thanks for your comment. We will make the revisions as follows.

"In line with this objective, referring to the work of Petch et al. (2023), we strategically selected single product for each water component to construct water budget equation, thereby laying the foundation for further research."

**C/ (7) Fig. 3: It is recommended to add further explanations in the caption of Fig. 3.**

R/ Thank you for your valuable suggestion. We will add further explanations to the caption of Fig.3 as follows.

"Figure 3. Illustration of the correction process advancing convergence between the simulation and measurement systems. The measurement system is corrected to approach the simulation system, while the simulation system is refined via parameter calibration to better approximate the measurement system. As a result, the distance between the two systems is reduced, leading to better physical consistency in the measurement system."

**C/ (8) Line 458: I suggest emphasizing the spatial distribution of water balance closure.**

R/ Thank you for your suggestion. Based on your advice, we will revise the sentence to:

"Therefore, we speculate that the spatial distribution of water budget closure is predominantly influenced by the characteristics of the basin."

**C/ (9) Line 619: A "." is missing before the "The major".**

R/ Thank you for pointing out this oversight. We will add a period between the sentences.

**Reference**

Beven, K.: Towards an Alternative Blueprint for a Physically Based Digitally Simulated Hydrologic Response Modeling System, Hydrological Processes - HYDROL PROCESS, 16, 189-206, 10.1002/hyp.343, 2002.

Cooper, R., Hodgkins, R., Wadham, J., and Tranter, M.: The hydrology of the proglacial zone of a high-Arctic glacier (Finsterwalderbreen, Svalbard): Sub-surface water fluxes and complete water budget, Journal of Hydrology, 406, 88-96, 10.1016/j.jhydrol.2011.06.008, 2011.

Hoang, L.: La formule du savoir: Une philosophie unifiée du savoir fondée sur le théorème de Bayes, 10.1051/978-2-7598-2261-4, 2020.

Liang, X., Lettenmaier, D. P., Wood, E., and Burges, S.: A simple hydrologically based model of land-surface water and energy fluxes for general-circulation models, J. Geophys. Res., 99, 14415-14428, 10.1029/94JD00483, 1994.

Lorenz, C., Kunstmann, H., Devaraju, B., Tourian, M., Sneeuw, N., and Riegger, J.: Large-Scale Runoff from Landmasses: A Global Assessment of the Closure of the Hydrological and Atmospheric Water Balances, Journal of Hydrometeorology, 15, 10.1175/JHM-D-13-0157.1, 2014.

Nicolle, P., Pushpalatha, R., Perrin, C., Francois, D., Thiéry, D., Mathevet, T., Le Lay, M., Besson, F., Soubeyroux, J.-M., Viel, C., Rousset, F., Andréassian, V., Maugis, P., Augeard, B., and Morice, E.: Benchmarking hydrological models for low-flow simulation and forecasting on French catchments, Hydrology and Earth System Sciences, 18, 2829-2857, 10.5194/hess-18-2829-2014, 2014.

Penning de Vries, M., Fennig, K., Schröder, M., Trent, T., Bakan, S., Roberts, J., and Robertson, F.: Intercomparison of freshwater fluxes over ocean and investigations into water budget closure, Hydrology and Earth System Sciences, 25, 121-146, 10.5194/hess-25-121-2021, 2021.

Rogelis, C. and Werner, M.: Streamflow forecasts from WRF precipitation for flood early warning in mountain tropical areas, Hydrology and Earth System Sciences, 22, 853-870, 10.5194/hess-22-853-2018, 2018.

Wang, J., Zhuo, L., Han, D., Liu, Y., and Rico-Ramirez, M.: Hydrological Model Adaptability to Rainfall Inputs of Varied Quality, Water Resources Research, 59, 10.1029/2022WR032484, 2023.

Wang, Y. H. and Gupta, H.: A Mass-Conserving-Perceptron for Machine-Learning-Based Modeling of Geoscientific Systems, Water Resources Research, 60, 10.1029/2023WR036461, 2024.

---

## Author Comment (AC5)

**Response to Referee #2**

**Title**: Achieving water budget closure through physical hydrological processes modelling: insights from a large-sample study
**Authors:** Xudong Zheng, Dengfeng Liu*, Shengzhi Huang*, Hao Wang, Xianmeng Meng
**Manuscript ID**: hess-2024-230

**Reply on RC3:**

First and foremost, I would like to express my sincere gratitude for your prompt reply and for the time and effort you have so generously devoted to reviewing our paper. We also greatly appreciate your recognition of the value of our work, as well as the opportunity you have given us to make revisions.

As you rightly pointed out, in our previous response, we primarily provided explanations for your concerns and conducted a few minor experiments, such as our response to Major Concern (7), Question b. This seems to have addressed your concerns to some extent, but we understand that it is insufficient. The lack of comparison with existing methods was mainly due to the two challenges: (1) first, finding data with the same time, spatial range, and appropriate resolution; (2) second, the time required to implement the existing methods.

After further interactive discussion with you, we recognized that this comparison is essential. Therefore, we sought to gather multiple sources of data (including site observations, remote sensing, and simulations) as much as possible, and implemented several existing correction methods (i.e., PR and CKF) to compare their correction with our results. This comparison was conducted in several representative basins (following your suggestion), which provides evidence for the reliability of our framework. We hope this experiment will address your concerns, and we also appreciate your valuable suggestions. The details of the comparison are provided in the point-by-point responses below.

Thank you again for your reply. We are also very pleased to engage in the academic discussion with you, which is highly meaningful. Below, we will provide a point-by-point reply to your comments.

**Note:**
For better readability, replies will start with "**R/**", following the original comments that start with "**C/**" and are shown in **bold**. The revisions to be added into the revised manuscript is highlighted in red. The important parts are highlighted in blue. The quoted content is displayed in *italics*.

**Point-to-point response:**

**C/ The author's approach of studying water balance closure from the perspective of physical mechanisms does indeed have academic value.**

**R/** Thank you very much for recognizing the value of our work; this is a great encouragement for us.

**C/ However, the core issues I raised have not been fully addressed. The author mainly provided some explanations without offering experimental evidence to demonstrate the reliability of the proposed method.**

**R/** We sincerely apologize for having avoided addressing your concerns in our previous response. In this response, we have adopted your suggestions, given them careful consideration, and made every effort to conduct related experiments within the limited time available. Detailed experimental results are provided below, presented as a new subsection that will be added to the manuscript.

**C/ I maintain that a comparison with existing methods is necessary to validate the accuracy and reliability of the proposed approach. The purpose of achieving water balance closure has two main components: improving data consistency and accuracy. Regarding data consistency, the author's method does not fully achieve water budget closure (I agree with the principle behind the author's approach). Therefore, if the method's performance cannot be verified in terms of data accuracy, its overall effectiveness and reliability remain questionable. I recommend that the author select some representative basins with measurements of budget components for validation.**

**R/** We are very pleased that you recognize the principles behind our methods, and we greatly appreciate the valuable suggestions you have provided. As you mentioned, further validating the calibration results through comparisons with existing methods can emphasize the reliability of our proposed approach. The approach of selecting representative basins for validation is also feasible, therefore we proceed with experiment in this regard. In this experiment, potential issues may include inconsistencies in temporal and spatial scales, as well as mismatches between grids and basins. Detailed results are provided in the responses below, presented as a new subsection that will be added to the manuscript.

**C/ As for the author's claim that a comparison with existing methods is not appropriate, I disagree. Some current methods estimate the distribution weights of water imbalance based on fused values (some methods are not such as PR and MCL), rather than using the fused values as exact reference points. I recommend validating the proposed method by comparing it with existing methods based on in-situ measurements of budget components (in regions with in-situ measurements, such as P and Q). Additionally, considering multiple datasets for each hydrological variable would be beneficial for validating the proposed method. The author argues that errors in hydrological model simulations only represent physical inconsistency errors, while datasets capture comprehensive errors. If multiple datasets consistently identify omission errors, this would demonstrate the reliability of the method. I recommend that the author select some representative basins for validation.**

**R/** We acknowledge your perspective. Considering multi-source data for each hydrological component, along with comparisons of the corrected results from existing methods, will effectively demonstrate the reliability of our approach.

Therefore, we collected multisource datasets from in-situ observations, remote sensing retrievals, and model simulations. This includes 11 precipitation, 14 evaporation, 11 streamflow and 2 terrestrial water storage datasets (see Table S3). We have implemented two existing correction methods: the PR and CEnKF methods (Luo et al., 2023). A new subsection will be added to the manuscript to clarify the

comparison between the PHPM-MDCF and existing methods (see below). In general, the comparison results from several representative basins indicate that the PHPM-MDCF can produce reliable correction results, reflected in several aspects: (1) a consistent over trend with existing method; (2) the absence of unreasonable corrections in streamflow; (3) the correction was also applied to TWSC (compared to CEnKF); and (4) a good consistency between the retrieved TWSC (from SM and SWE change) and GRAEC TWSC.

This comparison indeed further demonstrates the reliability of PHPM-MDCF, with detailed results presented below (in red). Due to time constraints, we have conducted experiments to the best of our ability. Therefore, it is worth mentioning that this comparison still includes potential uncertainty from scale and spatial mismatch issues.

Regardless, the PHPM-MDCF retains advantages in generating high-resolution corrections (daily), as it does not rely on multi-source datasets for the every variable but rather utilizes physical processes characterized by hydrological models as constraints. Theoretically, we can perform this correction at any model time step and for any model output variable.

"4.3.3 Comparison with existing correction methods

Previous analysis and experiments clarify the unique characteristics of the PHPM-MDCF, which impose closure constraints based on hydrological physical processes. This differs significantly from existing correction methods, such as PR and CEnKF (Luo et al., 2023). In this section, we conducted a comparison analysis with them to further evaluate the reliability of the PHPM-MDCF. To implement existing correction methods, support from multisource measurements for each water component is essential for calculating the residual allocation weights. Here, we obtained monthly datasets from Lehmann et al. (2022), which include 11 precipitation, 14 evaporation (ET), 11 streamflow (R) and 2 terrestrial water storage (TWS) datasets (Table S3). The datasets previously utilized in this study were also included for data fusion and correction (Table 1). In general, these datasets were processed to a uniform monthly scale and a common period (2003-2010), and subsequently aggregated to the basin scale. Several representative basins (numbered 1539000, 1557500, and 3070500) were selected to illustrate the differences between the PHPM-MDCF and existing methods, based on the spatial coverage of multisource datasets.

Figure 11 presents a comparison of the monthly correction results from three methods (i.e., PR, CEnKF, and PHPM-MDCF) for three main water budget components at basin 1539000. Note that the measurements of precipitation are not compared here, as the PHPM-MDCF does not perform correction for this variables. It is clear from the figure that both the PHPM-MDCF and CEnKF method exhibit minimal correction of ET, whereas the PR method significantly expands the range of ET, particularly increasing seasonal peaks. This arises from the assumption of the PR method that relative errors are proportional to the relative magnitudes of each variable (Abhishek et al., 2022). But in many cases, this assumption may not hold true.

In terms of the R and terrestrial water storage change (TWSC), the overall trends of the correction results from the three methods are generally consistent. However, the CEnKF appears to produce greater fluctuations in R (Fig. 11 b and e) and shows limited correction of TWSC. This is linked to the computational mechanism underlying CEnKF, where the Kalman gain—or the error covariance between measurements and the ensemble mean of multisource datasets—determines the magnitude of the residuals for each variable. The measurements of R to be corrected is based on in-situ obervations, while the

multisource dataset includes model simulations and remote sensing values. Potential mismatches between the grids and basins may lead to significant discrepancies, resulting in an greater allocation of correction for R. On the contraty, measurements of TWSC are limited and primarilty deriving from GRACE, which results in relatively small error covariance and, consequentlt, smaller corrections. Furthermore, as previously noted, such method may generate unreasonable corrections due to propogation of extreme errors, such as the negative R values in Fig. 11b, which are more likely to occur in small basins. PHPM-MDCF avoids these issues by considering physical process constraints, leading to reasonable corrections. Additionally, it dose not rely on multisource datasets and can perform correction on a daily scale. The TWSC derived from SWE and SM is consistent with GRACE TWSC, which also demonstrates the reliability of this framework. The comparison results for the other 2 representative basins are shown in Fig. S11-12, leading to similar conclusions."

[Figure]

**Figure 11.** Comparison of monthly correction results between the PHPM-MDCF and existing methods (PR and CEnKF) at basin 1539000.

(a-c) Time series of the original and corrected measurements of evaporation, streamflow, and terrestrial water storage change. (d-f) Scatter plots and regression lines of the original and corrected measurements.

**Table S3.** Summary of datasets from Lehmann et al. (2022).

| Variable | Product | Original Resolution | | Original Period |
| --- | --- | --- | --- | --- |
| | | Spatial | Temporal | |
| Precipitation | CPC | 0.5 °×0.5 ° | Monthly | 2002-2017 |
| | CRU | 0.5 °×0.5 ° | Monthly | 1901-2019 |
| | ERA5 Land | 0.1 °×0.1 ° | Monthly | 1981-2020 |
| | PGF | 1.0 °×1.0 ° | Monthly | 1948-2014 |
| | GPCC | 0.5 °×0.5 ° | Monthly | 1891-2016 |
| | GPCP | 2.5 °×2.5 ° | Monthly | 1979-2020 |
| | GPM | 0.1 °×0.1 ° | Monthly | 2000-2020 |
| | JRA55 | 0.5 °×0.5 ° | Monthly | 1959-2020 |
| | MERRA2 | 0.5 °×0.625 ° | Monthly | 1980-2020 |
| | MSWEP | 0.5 °×0.5 ° | Monthly | 1979-2020 |
| | TRMM | 0.25 °×0.25 ° | Monthly | 1998-present |
| Evaporation | ERA5 Land | 0.1 °×0.1 ° | Monthly | 1981-2020 |
| | FLUXCOM | 0.5 °×0.5 ° | Monthly | 2001-2015 |
| | GLDAS22 CLSM | 0.25 °×0.25 ° | Daily | 2003-2020 |
| | GLDAS20 CLSM/NOAH/VIC | 1.0 °×1.0 ° | Monthly | 1979-2014 |
| | GLDAS21 NOAH/CLSM/VIC | 1.0 °×1.0 ° | Monthly | 2000-2020 |
| | GLEAM | 0.25 °×0.25 ° | Monthly | 1980-2018 |
| | JRA55 | 0.5 °×0.5 ° | Monthly | 1959-2020 |
| | MERRA2 | 0.5 °×0.625 ° | Monthly | 1980-2020 |
| | MOD16 | 0.5 °×0.5 ° | Monthly | 2000-2014 |
| | SEBBop | 0.5 °×0.5 ° | Monthly | 2003-2020 |
| Streamflow | ERA5 Land | 0.1 °×0.1 ° | Monthly | 1981-2020 |
| | GLDAS22 clsm | 0.25 °×0.25 ° | Daily | 2003-2020 |
| | GLDAS20 CLSM/NOAH/VIC | 1.0 °×1.0 ° | Monthly | 1979-2014 |
| | GLDAS21 CLSM/NOAH/VIC | 1.0 °×1.0 ° | Monthly | 2000-2020 |
| | GRUN | 0.5 °×0.5 ° | Monthly | 1902-2014 |
| | JRA55 | 0.5 °×0.5 ° | Monthly | 1959-2020 |
| | MERRA5 | 0.5 °×0.625 ° | Monthly | 1980-2020 |
| Terrestrial water storage | GRACE JPL mascons | 0.5 °×0.5 ° | Monthly | 2002-present |
| | GRACE CSR mascons | 0.5 °×0.5 ° | Monthly | 2002-present |

[Figure]

**Figure S11.** Same as Fig. 11, but for basin 1557500.

[Figure]

**Figure S12.** Same as Fig. 11, but for basin 3070500.

**C/ Finally, the observational data referenced by the author is not in-situ measurements, and attention should be given to the terminology used.**

**R/** Thank you for pointing this out, we will emphasize the scope of this term's usage in this paper. The following is the content we will add to the data description section.

"Notably, the term 'measurements' referred in this work are derived from multisource datasets and do not

specifically refer to in-situ measurements."

---

## Author Response (AR1)

**Response to comments on Manuscript hess-2024-230**

**Title**: Achieving water budget closure through physical hydrological processes modelling: insights from a large-sample study

**Authors:** Xudong Zheng, Dengfeng Liu*, Shengzhi Huang*, Hao Wang, Xianmeng Meng

**Manuscript ID**: hess-2024-230

Dear Editor and Reviewers,

Please find enclosed our responses to the manuscript assessment entitled "Achieving water budget closure through physical hydrological processes modelling: insights from a large-sample study".

We would like to express our sincere gratitude to the Editor, the two anonymous reviewers, and the community reviewer for their invaluable support and constructive suggestions, as well as for the opportunity afforded to us to revise our work. We have given full attention to all comments and suggestions and made all revisions accordingly. It has resulted in an improved manuscript that fully addresses all concerns.

Concerning the revision of the manuscript, the following major changes have been made:

(1) We have added Sect. 4.3.3, which further validates the reliability of the proposed framework by comparing the correction results of existing methods with those of PHPM-MDCF. This is accomplished through the collection of multisource products, including 11 precipitation, 14 evaporation, 11 runoff, and 2 terrestrial water storage datasets.

(2) Section 4.4 has been reorganized into three subsections to investigate the potential factors influencing the spatial distribution, temporal distribution, and proportions of water budget residuals. The results further demonstrate the rationale of the decomposition method by validating the physical meaning of omission residuals.

(3) We have expanded the description of the data and methods section, emphasizing the reliability of the multisource datasets used in this study while also incorporating formulas, explanations, and parameter descriptions to enhance the comprehensibility of the PHPM-MDCF framework.

We believe the paper quality has significantly improved through this review process. We are also happy to address any comments that may further strengthen the paper quality. We are uploading our point-by-point response to the comments, an updated manuscript with red highlighting indicating changes, and a clean updated manuscript without highlights.

In the point-by-point responses below, the original comments are displayed in **bold**. **EC-n/**, **RC1-n/**, **RC2-n/**, **RC3-n/**, and **CC1-n/** correspond to the Editor, Referee 1, Referee 2, Referee 3 and Community Comments, respectively. The corresponding responses begin with **R/** and the revisions is highlighted in red, while important sections are marked in blue. The quoted content is displayed in *italics*.

We thank you for your consideration,

Dengfeng Liu
Email: liudf@xaut.edu.cn

**Responses to Editor:**

**EC-1/ Both reviewers agree with the novelty of the concept, but they also raised major concerns. Please address their comments carefully, and upload the revised manuscript. Should you agree or disagree with comments, please provide a point-by-point response. The response and revised manuscript will be sent to reviewers for the second round of assessment.**

**R/** We sincerely appreciate your timely handing of our manuscript and the opportunity to revise it. All comments from reviewers have been addressed point-by-point below. The updated manuscript, which includes a tracked version with changes highlighted in red and a clean version without highlights, will be submitted alongside this response file.

We believe that our manuscript has significantly improved through this review process, and we are open to addressing any comments that may further enhance the quality of our paper. If there are any questions or suggestions, please feel free to contact us.

**Responses to RC1:**

Many thanks for taking the time and effort to review our paper. All comments from RC1 are addressed below with point-by-point responses.

**RC1-1/ This paper emphasizes the issue of decreasing data confidence at the watershed scale in the era of big data, caused by the non-closure of water budget from multiple data sources. In their analysis, the total water budget residuals were quantitatively decomposed into two components, inconsistency and omission residuals, to account for different drivers of water budget non-closure phenomenon. This is an interesting addition, as previous studies have typically given little or only qualitative consideration to the water imbalance caused by omissions in the original water balance equation.**

**Attempting to close the water balance is valuable, both hydrological inference under climate change and hydrological modeling require data that satisfy the basic assumption of water balance. The PHPM-MDCF proposed in this work employ hydrological model to constrain multisource datasets, which is reasonable because hydrological models are well-known for their water balance capabilities. The correction also seems to be effective, which comes from the validation with results from large sample basins.**

R/ Thanks for your positive feedback and recognition of our work. Your comments are valuable for revising and improving our paper. Below, we provide detailed responses to each of your concerns. The corresponding revisions are attached after the responses and have been incorporated into the revised manuscript.

**RC1-2/ However, there are still some concerns that need to be explained in the response or addressed in the manuscript. The authors have observed the typical seasonal pattern of non-closure phenomena but lack corresponding explanations. In addition, although the authors decomposed the closure residuals into two parts, it seems that only the inconsistency residuals were corrected. What is the rationale behind this approach? Why were the omission residuals not corrected?**

R/ Your points are very insightful. Adding explanations about the seasonal characteristics of the non-closure phenomenon will indeed strengthen our argument. In addition, as you mentioned, our framework primarily addresses the $Res_i$ (inconsistency residuals), without considering the correction of $Res_o$ (omission residuals). This is because we consider it as unaccounted-for water in the original water budget equation, which should be explained by other water components.

Addressing the two questions has provided us with excellent insights to further strengthen the arguments in our manuscript and significantly improved its quality. To ensure clarity in the structure of our responses, we provide more detailed responses and revisions in ***RC1-4*** and ***RC1-5***. Please find them below.

**RC1-3/ In summary, this paper is innovative and aligns with the interests of potential readers of the HESS. After careful consideration and revision, this work has the potential to make a significant contribution to this field. As they described, the underlying Bayesian philosophy is an approach for aligning our understanding of natural processes with real-world observations.**

**R/** Thank you again for acknowledging our work and perspectives.

**Major concerns**

**RC1-4/ Sect. 4.1, the patterns of the Res are of interest to me. The authors identified typical spatial distributions and compared them with previous studies in Sect. 4.4, explaining these patterns through hydro-meteorological conditions and watershed area. From a physical perspective, this explanation is consistent with common sense and is sufficient for me. However, the temporal patterns of the Res are also of interest (Fig. 5). The authors should provide further explanation in this regard or compare them with previous studies, as this could offer valuable insights into the causes of the non-closure of water balance.**

**R/** Thanks for your suggestion. As we mentioned earlier, we agree with your suggestion to include further explanation of the temporal distribution of $Res$. This is addressed in two ways: (1) comparing the observed seasonal patterns in $Res$ with previous studies, and (2) providing an analysis from a physical causation perspective. A new subsection (Sect. 4.4.2, Line 565-595 in tracked version) has been added to the manuscript to clarify this issue. The revised content is attached at the end of this response.

Indeed, the temporal distribution of $Res$ shown in Fig. 5 is quite striking. Specifically, as we mentioned in our manuscript, there is a positive bias in $Res$ during the warm season and a negative bias during the cold season. By comparing with previous literature, we found similar temporal distributions and potential influencing factor—namely, the potential underestimation of warm-season evaporation and cold-season precipitation (Kauffeldt et al., 2013; Newman et al., 2015; Lv et al., 2017; Abolafia-Rosenzweig et al., 2020; Robinson and Clark, 2020).

From a physical perspective, the underestimation is related to phenomena such as snowfall, freezing rain, and non-convective precipitation that occur during the cold season, as well as the calculation of evaporation during the warm season.

A further analysis was conducted to examine this by comparing the ratios of evaporation and precipitation for cold and warm seasons separately, along the corresponding $Res$. Scatter plot shows that basins dominated by cold-season precipitation are more likely to exhibit larger negative $Res$ during cold-season, while basins with higher warm-season evaporation tend to have larger positive $Res$ during warm season (Fig. 13). In both cases, $Res$ is more sensitive to underestimation of precipitation and evaporation, which is consistent with findings from previous research.

Although it is impossible to obtain true values to evaluate the measurements, these results still highlight potential uncertainties in cold-season precipitation and warm-season evaporation measurements, which could severely impact the assumption of water balance.

The analysis process and corresponding figure added into the revised manuscript are given below:

**4.4.2 Factors influencing temporal distribution**

The pronounced seasonal pattern of non-closure residuals depicted in Fig. 5 is quite interesting. To gain more insight into the observed pattern, we compare it with the temporal factors reported in the literature. The first and foremost reported factor associated with the observed negative biases in Res during the cold season is the underestimation of precipitation (Newman et al., 2015). This systematic bias is related to

phenomena such as snowfall, freezing rain, and non-convective precipitation that occur during the cold season, where measurements and simulations are prone to significant errors, including the well-know undercatch phenomenon (Kauffeldt et al., 2013; Robinson and Clark, 2020). Another key factor influencing water budget non-closure is connected to the temperature and evaporation dynamics. Abolafia-Rosenzweig et al. (2020) evaluated the water budget residuals over 24 global basins and found that the likelihood of positive biases in the water balance increases with rising temperatures, which likely induced by the potential uncertainties in evaporation estimates. The research by Lv et al. (2017) also support this perspective, indicating that the underestimation of evaporation is a primary contributor to the water budget non-closure. In summary, according to the literature, cold-season precipitation and warm-season evaporation seem to be the primary drivers of the temporal distribution of Res. To examine this reasoning, while obtaining the true values is impossible, we can provide evidence by comparing evaporation and precipitation, along with the corresponding residuals, between the cold and warm seasons.

Figure 13 depicts the relationship by separately comparing the ratios of evaporation and precipitation for the cold and warm seasons, with the corresponding water budget residuals. For the cold season, the scatter points can be split into two distinct regions along the vertical line where the ratio is 1. The scatter points in the left region indicate basins where cold-season precipitation is lower than in the warm season, leading to relatively smaller absolute residuals (clustered around zero residuals). In contrast, scatter points for basins with dominant cold-season precipitation are dispersed below the zero residual line, with larger negative residuals becoming more prevalent as the proportion of cold-season precipitation increases. In other words, regions where cold-precipitation constitutes a larger proportion of the water budget residuals are more sensitive to the underestimates of precipitation, resulting in larger negative residuals. Furthermore, we observed similar trends in the warm season, where a higher proportion of warm-season evaporation is associated with larger positive residuals. These results confirm the perspective of previous research, highlighting the potential uncertainties in measurements of cold-season precipitation and warm-season evaporation.

[Figure]

**Figure 13.** Relationship between the ratios of evaporation and precipitation for the cold and warm seasons separately and the corresponding water budget residuals. Note that blue represents residuals for the cold season, and red represents those for warm season. The seasonal division are consistent with Fig.5. The unit of residuals is "mm".

**RC1-5/ From Fig. 6, it appears that the Res and Resi have been effectively corrected, but the Reso have not changed significantly. Is this merely a specific case for this basin or a general situation? If it is a general situation, dose this imply that PHPM-MDCF only corrects for Resi and does not account for Reso? I believe that further explanation of this treatment could improve the transparency of the methods used in the paper.**

**R/** Yes, as you mentioned, our framework only corrects for $Res_i$ and does not account for the correction of $Res_o$. This is a general situation for all basins. Essentially, such treatment is guided by the underlying logic of the correction process, as revealed by the residuals decomposition in Eq. 3. $Res_o$ is separated from the total water budget residuals to account for water components not considered in the original equation, such as inter-basin exchange.

From a causal perspective, this portion of residuals is less associated with physical inconsistency, as confirmed by the spatiotemporal distribution difference (Fig. 4-5) discussed in Sect. 4.1. Therefore, the framework focused on constraining residuals using physically consistent hydrological model cannot correct this part of residuals. This also explains why $Res_i$ decreases significantly after correction in Fig. 6, while $Res_o$ remains unchanged.

In addition, the discussion in Sect. 4.2 also highlighted this issue:

*However, despite recalibrating the model with corrected datasets, $Res_o$ driven by the omission in water budget equation exhibited no substantial changes before and after correction (e.g., the monthly mean absolute values maintain around 6.5 mm, see Fig 6f). This phenomenon occurs because we only corrected the inconsistency residuals with reference to the simulation system, while the omission accounting for addition water terms should not be corrected in the existing datasets.*

In our opinion, using measurements to describe the theoretical water balance requires two key conditions: (1) physically consistent measurements, and (2) comprehensive description of the water budget equation. However, this is challenging to achieve in practice, whether due to inadequate understanding or limitations in measurement techniques, resulting in residuals corresponding to $Res_i$ and $Res_o$.

The framework proposed in this work can, to some extent, enhance physical consistency between measurements through the model, resulting in reduced $Res_i$. However, achieving a more comprehensive description (i.e., reducing $Res_o$) may involve more issues, such as scale effects, more detailed data (both surface and subsurface), and a deeper understanding of the watershed. Addressing these questions is beyond the scope of this study. We look forward to more detailed future research addressing these issues, as mentioned in our discussion:

*Further investigation would be required to better understand the omission residuals from a physical perspective. For example, a distributed hydrological model with representation of subsurface later flow process will allow us to identify the magnitude of inter-basin interactions; a more detailed description of water budget equation in data-rich environments can help us examine the sources of omission errors. This is undoubtedly important, but not the focus here.*

To further validate the rationale of the physical meaning of $Res_o$ in this study, we intentionally exclude the SWE component from the water budget equation to access its impact on the decomposition of $Res$. This is a plausible scenario in practice, as it is likely that this component was not considered when

reconstructing the TWSC. The results and analysis process of the experiment have been organized into a new subsection (Sect. 4.4.3, Line 596-629 in tracked version) added to the manuscript, which is attached at the end of this response.

The results indicate that the proportion of $Res_o$ obtained from residuals decomposition after excluding SWE increases significantly, with this effect being more pronounced in high-latitude regions, high elevations, and during the cold season (Fig. 14). This is consistent with physical principles, as the impact of omitting SWE on water balance is greater under these situations. These findings align with our definition of $Res_o$ which refers to the water imbalance caused by omitted water. It also, to some extent, supports the validity of our decomposition method, and highlights the importance of a comprehensive water budget equation.

The revised content is as follows:

**4.4.3 Factors influencing the proportions of residuals components**

Another interesting finding in Sect. 4.1 is that the magnitude of $Res_o$ is significantly smaller than that of $Res_i$. As a result, $Res$ is dominated by $Res_i$, leading to a highly consistent spatiotemporal distribution between them. However, the underlying question is what this implies and which factors drive the proportions of the residuals components.

$Res$ reflects the degree to which the measurements achieve water budget closure. In this study, we argue that two key conditions are necessary for using measurements to describe theoretical water balance. The first one is that measurements of different water components must be physically consistent. In practice, however, this condition is often challenging to meet due to inconsistencies and uncertainties in data production processes from different sources, which can result in non-zero $Res_i$ (Luo et al., 2020). The second crucial, yet frequently overlooked, condition is the completeness of the water budget equation. Building on the work of Gordon et al. (2022), we developed a more generalized water budget equation (Eq. (3)) and use $Res_o$ to account for the water imbalances caused by omitted water. From this perspective, $Res$ results from the interplay between $Res_i$ and $Res_o$, either through their accumulation or mutual cancellation. Therefore, the low proportion of $Res_o$ essentially suggests that our description of the water budget equation is comparatively comprehensive.

Consider that if our description of the water budget equation is incomplete and omits a significant water component, $Res_o$ would likely exert a greater influence on $Res$, resulting in a more pronounced discrepancy between $Res$ and $Res_i$. To examine this, we intentionally exclude the SWE component from the water budget equation to evaluate its impact on the decomposition of $Res$. This is a plausible scenario in practice, as it is likely that this component was not considered when reconstructing the TWSC. Figure 14 illustrates the comparison between $Res_o$ derived from the decomposition method excluding SWE (hereafter $Res_o^{NSWE}$), and its original values. It is evident that $Res_o^{NSWE}$ exhibits greater variability compare to the original values (i.e., with smaller minimum values and larger maximum values). The median differences indicate that the likelihood of increased omission residuals is higher after excluding SWE (Fig. 14b). Such differences reveal that omitting crucial SWE storage component results in a greater degree of water imbalance, and, as expected, this effect is more pronounce in high-latitude and high-elevation regions (Fig. 14d-f). Moreover, the spatiotemporal distribution of $Res_o$ has changed (Fig. S13-14). Notably, during the cold season (December to February), the proportion of $Res_o$ is much higher and exhibits a significant positive trend. These findings align with our definition of $Res_o$, which refers to the water imbalance caused by omitted water. It also supports the validity of our decomposition method to

some extent, and highlights the importance of a comprehensive water budget equation in evaluating water balance.

[Figure]

**Figure 14.** Comparison of $Res_O$ obtained from residuals decomposition excluding SWE with the original values. (a-c) Spatial distribution of monthly mean $Res_O$ excluding SWE minus its original values. (d-f) Time series of $Res_O$ excluding SWE and its original values at the southern basin (02198100, 32.96°N), northern basin (12358500, 48.33°N), and high-elevation basin (07083000, elevation of 3.56 km) at monthly scale. The unit of residuals is "mm".

**RC1-6/ Although the author has clearly articulated the main scientific problem of the paper, there are still areas that could be further improved, which I have listed in the specific issues.**

**R/** Thank you for your thorough and detailed review. We have addressed each point and provided responses below.

**Specific issues**

**RC1-7/ Line 22-25: According to the results, it seems that humid/wet basins are also prone to larger closure residuals, which needs to be emphasized here.**

**R/** According to your suggestion, we revised the phrasing to (Line 25-26 in tracked version):

This emphasizes the importance of carefully evaluating the water balance assumption when employing multisource datasets for hydrological inference in small and humid basins.

**RC1-8/ Line 36-46: I believe this section should place greater emphasis on the issues of scale mismatch and difficulty in obtaining reference data.**

**R/** Thanks for your comment. We have revised the manuscript to strengthen the issue of scale mismatches and the challenges associated with obtaining site data.

The following statement has been added (Line 44-45 in tracked version):

The issue of scale mismatches and the availability of site data in certain regions also pose challenges for data evaluation.

**RC1-9/ Line 58-60: It is recommended to cite the review by Beven (2002).**

**R/** Thanks for your suggestion, we have included this reference to the Line 62 in tracked version:

Such inconsistency poses an obstacle to robust hydrological inferences (Beven, 2002).

**RC1-10/ Line 83-84: It is recommended to add references to support the argument.**

**R/** We found supporting evidence in the literature Luo et al. (2023) and have included this reference to substantiate our argument (Line 87 in tracked version).

Luo et al. (2023): *therefore, the results confirm that increasing the water budget closure accuracy of budget-component data sets reduces the accuracy of individual budget-component products.*

In the context of applying such closure constraint, it becomes evident that the precision of certain individual components may notably deteriorate, particularly when uncertainties are challenging to quantify (Luo et al., 2023).

**RC1-11/ Line 119: "Res" does not appear to be in italics.**

**R/** Thank you for your reminder. This formatting issue has been corrected throughout the revised manuscript (Line 110, 112, 113, 122 in tracked version).

**RC1-12/ Line 126-127: It is recommended to change it to: "(a) How can the total water budget residuals be quantitatively decomposed into inconsistency and omission residuals based on Eq. (3)?"**

**R/** Thanks for your careful review. We have revised the sentence according to your suggestions (Line 129-130 in tracked version):

(a) How can the total water budget residuals be quantitatively decomposed into inconsistency and omission residuals based on Eq. (3)?

**RC1-13/ Table1: The "period" should be "Original Period".**

**R/** Thanks, we have corrected this (Line 213 in tracked version).

**RC1-14/ Figure5: The figure caption seems to contain an error. There are no other subfigures.**

**R/** The caption of this figure did indeed contain errors due to update to the figure, and we have revised it accordingly (i.e., remove redundant subplots sequence numbers, Line 374-377 in tracked version):

**Figure 5.** Temporal distribution of monthly water budget residuals ($Res$), inconsistency residuals ($Res_i$), and omission residuals ($Res_o$) across 475 CAMELS basins with reliable simulations. Boxplot-like diagrams describe variability across catchments, and outliers represent the 10th and 90th percentiles. The unit of residuals is "mm".

**RC1-15/ Line 332-334: The argument here doesn't seem to correspond with the figure. Could it be that the figure has been updated?**

**R/** Thank you for pointing out this error. We have revised the statement while updating the figure caption (remove redundant subplots sequence numbers, Line 365-366 in tracked version):

On the contrary, $Res_o$ tends to be mainly positive except from September to November; its extent of variability is also significantly smaller than that of the other two residuals. In regard to magnitude, $Res_i$ is much greater than $Res_o$, whether considering positive or negative bias.

**RC1-16/ Line 418: add "which are" before "implemented".**

**R/** Thanks, we have corrected it to (Line 456-458 in tracked version):

This is achieved by the representation of physical hydrological processes underlying the correction strategy, which constrains the corrected values to avoid producing extreme outliers.

**RC1-17/ Line424-426: Change the sentence to "The fact that simultaneous corrections of other variables during extreme runoff noise corrections did not significantly differ from OS-based corrections further enhances our confidence in PHPM-MDCF."**

**R/** Thanks for your comment. We have revised the statement according to your suggestion (Line 464-466 in tracked version).

The fact that simultaneous corrections of other variables during extreme runoff noises correction did not significantly differ from OS-based corrections further enhances our confidence in PHPM-MDCF.

**RC1-18/ Line 417: It is necessary to further emphasize the issue of the non-closure phenomenon in humid regions.**

**R/** Thank you for your suggestion, we have revised the entire manuscript to emphasize the issue of non-closure phenomenon in humid regions. Below are several examples of the revisions we have made:

Line 25-26 in tracked version: This emphasizes the importance of carefully evaluating the water balance

assumption when employing multisource datasets for hydrological inference in small and humid basins.

Line 555-557 in tracked version: These results highlight the risks of using multisource datasets for hydrological inference in humid and small-scale basins—specifically, potential physical inconsistencies—and underscore the need to carefully test the water balance assumption.

Line 823 in tracked version: This highlights the need for careful consideration of the water balance assumption when applying multisource datasets for hydrological inference in small and humid basins.

**RC1-19/ Figure 12: There seems to be a mistake with the R2 values.**

**R/** Thank you for pointing this mistake. We have corrected this mistake in the revised manuscript (Line 740 in tracked version). The updated figure with corrected R2 values is shown below:

[Figure]

**Figure 15.** Comparison of correction results based on different forcing datasets (TRMM and Daymet) at basin 1013500. (a-b) Corrected time series of five water budget variables. (c-e) Variation of long-term mean absolute values of three residuals with correction iterations at the monthly scale. The unit of residuals is "mm".

**RC1-20/ Line 639: Humid regions is a better expression.**

**R/** Thanks for your suggestion. We have made revisions throughout the entire manuscript. Here is an example:

Line 823 in tracked version: This highlights the need for careful consideration of the water balance assumption when applying multisource datasets for hydrological inference in small and humid basins.

**Thank you once again for your suggestions and help; they significantly improved the quality of our paper.**

**Responses to RC2:**

Thank you very much for dedicating your time and effort to reviewing our paper. All comments from RC2 are addressed below with point-by-point responses.

**RC2-1/ The paper presents an interesting concept, and its organization and writing are well done. However, I have some differing views regarding the underlying assumptions and principles of the proposed method. My main comments are as follows:**

**R/** First and foremost, we sincerely appreciate your interest in the concept shared in our paper, as well as your kind recognition of our writing and organization. We hold your constructive comments in high regard and believe it is instrumental in enhancing the quality of our paper. These comments have been addressed point by point below, and revisions have made in the manuscript to the best of our ability.

**Major Comments:**

**RC2-2/ I do not agree with the two underlying assumptions of the PHPM-MDCF method, nor with the significance of using Equation 4 to calculate omission errors. My main reasons are as follows:**

**Firstly, the errors in hydrological models are non-negligible and represent the sum of both omission errors and data errors, rather than omission errors alone. The paper assumes that hydrological models have no data errors (inconsistency errors) and only omission errors, which is evidently unreasonable. This assumption is particularly problematic because hydrological models are typically validated against observed runoff, often neglecting the validation of ET (Evapotranspiration) and TWSC (Terrestrial Water Storage Change) simulation accuracy. As a result, using Equation 4 to calculate omission errors is not justified. Due to the complexity of hydrological models and the impact of errors in driving variables, the water imbalance caused by errors in the hydrological model may be substantial. Even if the inputs to the hydrological model are observational data and the model itself is developed based on the principle of water budget, the primary contributor to water imbalance errors between input and output might still be data errors.**

**Secondly, the total residual is calculated using multiple sources of data, and omission errors are calculated using data that drive the hydrological model as per Equation 4. The difference between these is then used to calculate data inconsistency errors. However, this approach might introduce uncertainties due to data inconsistency.**

**R/** Thank you for your comment. We acknowledge that employing hydrological models to constrain measurements and thereby enhance water budget closure among them is an ambitious idea, as it has not been previously presented in the literature. We also recognize that accepting this idea is challenging. However, this idea is not proposed arbitrarily; rather, it is developed progressively along a specific logical path.

First, the errors in hydrological model that we describe as ignorable refer to inconsistencies occurring within the input, output, and state, rather than those between measurements. This distinction is important to emphasize. In other words, each variable in Eq. (4) originates from the model itself, and from this perspective, these variables are independent of measurements. Such consistency in hydrological model

has been described in numerous studies. For example, DeChant and Moradkhani, (2014) provided reduced structural equations for general distributed hydrological models from a state-space view:

$$s_{i,t} = f(x_{i,t}, s_{i,t-1}, \theta_i), \tag{RC2-2}$$

where $f()$ represents the model structure, $x_{i,t}$ is the forcing of the $i$th grid at time $t$. $\theta_i$ is the parameter of the $i$th grid. In this equation, a quantitative balance is maintained between the input/forcing and output/state variables. In the general hydrological models, whether distributed or lumped, water balance serves as a fundamental governing equation to constrain the model, which is a well-established practice (Beven., 2001). The above constitutes the logical basis for our assumption that the hydrological model satisfies water balance, ensuring physical consistency. This also aligns with our definition of inconsistency residuals, which refer to non-closure arising from physical inconsistency.

However, given our current understanding of the water cycle, Eq. (4) may still be prone to omission residuals. It can be challenging to be aware of all water components, certain omissive components result in omission residuals. This portion of the residuals can be identified through variables derived from the hydrological model, as these variables are consistent with water balance.

In extreme cases, if all components are considered in water budget equation, the omission residual can be reduced to zero. At this point, no water imbalance exists within the simulation system (i.e., Eq. (4)), and any remaining residuals in the measurement system would be the potential inconsistency residual.

Return to your question, the "data errors" you refer to are more likely the differences between simulated and measured values (e.g., simulated versus gauged runoff). This pertains to model performance, specifically whether the model can accurately represent hydrological process. This does not conflict with the water balance feature of the model itself. It is important to emphasize once again that all variables used in Eq. (4) are derived from the model, not from measurements.

I hope the above response provides some clarity on the issues related to water balance in the hydrological model and the potential neglect of inconsistency residuals in Eq. (4). In addition, we would like to further address the question of the relationship between measurements and simulations in this method. We believe that clarifying this point may help address your concerns.

In the PHPM-MDCF method, measurements are used not only calculate the total residuals (i.e., Eq. (5)), but also to constrain the model through a multi-objective calibration process (i.e., tuning parameters). As you emphasized, using only observed runoff to validate the model is insufficient. In this work, we considered five different variables—streamflow, ET, SMS (soil moisture storage), GRS (groundwater reservoir storage), and SWE—to validate the performance of the model. After model performance evaluation, we selected 475 basins with reliable simulation for all variables for subsequent analysis. The first paragraph of Sect. 4.1 and Appendix C provide detailed information. We present the main information here:

*To ensure the robustness of the results, as mentioned previously, it is essential that hydrological model reliably represent hydrological processes. With reference to previous studies (Clark et al., 2021), we have adopted KGE≥-0.41 and r statistically significant at the 5% level as criteria for guaranteeing reliable simulations. The multi-objective simulation performances of the HBV model are detailed in Appendix C. In general, the majority of basins (475, accounting for 72.24% of the total basins) achieved reliable simulations across all variables.*

[Figure]

**Figure C1.** The multi-objective simulation performances of the HBV model across the CAMELS basins. Results are based on (a) runoff, (b) evaporation, (c) soil moisture storage and groundwater reservoir storage, and (d) snow water equivalent. Red dots represent unreliable simulation performance, and the size of points is proportional to the basin area. The unit of RMSE is "mm".

In general, this helps ensure simulation accuracy to some extent and reduces the uncertainty in the residual decomposition. Furthermore, the multi-objection calibration process is repeatedly applied during multisource datasets correction to ensure that, after each iteration of data correction, the model can produce reliable simulations corresponding to the dataset.

Based on the response to this concern, we recognize the importance of further emphasizing the water balance assumption in hydrological model used in this method, particularly with respect to Eq. (4). Therefore, we have added the following statements to the manuscript (Line 230-232 in tracked version):

It is crucial to clarify that all variables in Eq. (4) are derived from the model itself, rather than from measurement, and can therefore be considered physically consistent.

Further examination on the physical meaning of $Res_o$ is presented in **RC2-8** derived from experimental comparisons that involve the removal of SWE from the water budget equation.

**RC2-3/ The validation of results should include a comparison between the PHPM-MDCF method and existing methods. The paper repeatedly emphasizes the inadequacy of current methods in distributing residuals, yet no comparison with existing methods is provided in the results to verify the accuracy of the PHPM-MDCF method. The goal of closing the water budget is to reduce residuals while improving the accuracy of water cycle variables. Therefore, the credibility of the model should not be judged solely by the reduction of residuals (Figure 6). A comparison with existing methods would be more convincing. I strongly recommend supplementing the results with a comparison against existing correction methods, particularly CKF, PR, and MCL methods. For instance, the accuracy of the datasets after calibration using these methods, including P (Precipitation), ET (Evapotranspiration), Q (Runoff), and TWSC (Terrestrial Water Storage Change).**

**R/** Following the two interactive discussions with you, we recognized the importance of comparing our framework with existing methods. This comparison can effectively substantiate the validity of the PHPM-MDCF from a relatively objective perspective.

Therefore, we collected multisource datasets from in-situ observations, remote sensing retrievals, and model simulations. This includes 11 precipitation, 14 evaporation, 11 streamflow and 2 terrestrial water storage datasets (see Table S3). We have implemented two existing correction methods: the PR and CEnKF methods (Luo et al., 2023). According to your suggestion, such comparison was conducted on several selected representative basins.

A new subsection has been added to the manuscript to clarify the comparison between the PHPM-MDCF and existing methods (Sect. 4.3.3, Line 496-533 in tracked version). This revision is also attached at the end of this response.

In general, the comparison results from several representative basins indicate that the PHPM-MDCF can produce reliable correction results, reflected in several aspects: (1) a consistent over trend with existing method; (2) the absence of unreasonable corrections in streamflow; (3) the correction was also applied to TWSC (compared to CEnKF); and (4) a good consistency between the retrieved TWSC (from SM and SWE change) and GRAEC TWSC.

This comparison indeed further demonstrates the reliability of PHPM-MDCF, with detailed results presented below (in red). Due to time constraints, we have conducted experiments to the best of our ability. Therefore, it is worth mentioning that this comparison still includes potential uncertainties from scale and spatial mismatch issues, as we discussed in the Sect. 4.3.3.

Regardless, the PHPM-MDCF retains advantages in generating high-resolution corrections (daily), as it does not rely on multi-source datasets for the every variable but rather utilizes physical processes characterized by hydrological models as constraints. Theoretically, we can perform this correction at any model time step and for any model output variable.

In addition to the above experiments, the noise experiments in Sect. 4.3.2 can also provide a theoretical indirect analysis for comparing the PHPM-MDCF with existing methods.

When extreme single-point noise is present in streamflow measurement (NS1 and NS2), it is expected that, to ensure water balance closure, existing correction methods will impose constraints across all variable by referencing "true values". Typically, streamflow measurements are considered to have the least uncertainty, leading to the smallest correction. As a result, extreme bias in streamflow can propagate to other variables by correction process, such as ET and TWSC. This is also the reason why the correction process, as previously discussed, can lead to a reduction in the accuracy of individual variables.

Figures 9 and S10 indicate that the PHPM-MDCF can effectively reduce residuals without causing such bias to propagate across different variables, thereby avoiding the aforementioned issues. This indirect analysis also provides some explanation for the differences between PHPM-MDCF and existing methods, and, to some extent, supports its reliability.

The added new section is as follows:

**4.3.3 Comparison with existing correction methods**

[revised manuscript text omitted]

**RC2-4/ The description of the reference datasets is unclear. It is necessary to specify which observational system datasets were used for P (Precipitation), ET (Evapotranspiration), Q (Runoff), and TWSC (Terrestrial Water Storage Change), and why these datasets can be considered observational data. I recommend clarifying this in the text.**

**R/** Thank you for your suggestion. We have revised Table 1 in accordance with your suggestions and provide the explanation for the selection of these datasets for each variable. Here is the revised version (Line 178-187, 212-213 in tracked version):

Specifically, daily precipitation estimation derived from the Tropical Rainfall Measuring Mission (TRMM 3B42V7) is used in this study. The well-known international NASA project aims to comprehensively estimate all forms of precipitation, including rain, drizzle, snow, graupel, and hail, through the integration of satellite data and ground-based rain gauge measurements (Huffman et al., 2016). The accuracy of TRMM dataset has validated by many studies through comparisons with observation data and other reanalysis datasets (Kittel et al., 2018; Villarini et al., 2009). For evaporation, we utilized the third version of Global Land Evaporation Amsterdam Model (GLEAM v3) product (https://www.gleam.eu/), which employs a set of algorithms to separately estimate the different components of land evaporation (Miralles et al., 2011). Several studies have demonstrated that this product aligns well with flux measurements and multisource product ensemble (Munier et al., 2014; Robinson and Clark, 2020). And, as mentioned above, the runoff measurements on a basin scale are provided by the CAMELS dataset, which is derived from site observations.

**Table 1.** Overview of the products for constructing water balance equation used in this study.

| Variable | Product | Original Resolution | | Original Period | Reference |
| --- | --- | --- | --- | --- | --- |
| | | Spatial | Temporal | | |
| Precipitation | TRMM 3B42V7 | 0.25 °×0.25 ° | Daily | 1998-2019 | *Huffman et al. (2016)* |
| Evaporation | GLEAM v3.8a | 0.25 °×0.25 ° | Daily | 1980-2022 | *Martens et al. (2017)* |
| Soil moisture layer 1/2/3/4 | EAR5 Land | 0.1 °×0.1 ° | Hourly | 1950-present | *Muñoz Sabater et al. (2021)* |
| Snow water equivalent | GlobSnow v3.0 | 25km×25km | Daily | 1979-2018 | *Luojus et al. (2021)* |
| Runoff | CAMELS USGS | Basin scale | Daily | 1980-2010 | *Newman et al. (2015)* |

**RC2-5/ Only a single product was selected for each water cycle variable. I believe that selecting multiple products is crucial for validating the proposed PHPM-MDCF method. This is because different datasets have different sources of error, leading to varying inconsistency residuals depending on the data combination. If the proposed method can be used to identify inconsistency residual error, using multiple data combinations would better verify the reliability of the proposed method in this study.**

**R/** Thank you for your comment. We acknowledge that a common practice in previous water budget assessments is to use a range of products for each water components, evaluating the availability of different product combinations to closure the water budget. For example, Lorenz et al. (2014) compared 180 combinations of datasets for P, ET, TWS, and Q to access the degree of atmospheric-land water

balance achieved. Lehmann et al. (2022) investigated the budget closure at catchment scales using 11 P, 14 ET, and 11 Q datasets together with GRACE.

However, almost all similar studies have reached the same conclusion that no single combination can close the water budget well across all regions (Lv et al., 2017). This implies that while introducing multiple products for ranking may be meaningful for specific regions, it holds limited significance for the correction framework of this study, which focuses on broader spatial scales (large sample basins). As Petch et al. (2023) handled in their optimization-based correction method, a single product was used for each water budget component, and they emphasize:

*In this study, we use only a single data product for each component, which we account for in our uncertainty calculations. We aimed to use Earth observation data where possible and sought global gridded products to ensure the uniformity of the uncertainties across all basins.*

*Overall, the specific datasets chosen were not critical, as our primary goal was to evaluate our new optimisation methodology and its ability to bring independent products into consistency.*

In this study, we do not consider multisource products for each variable for two additional reasons. First, different products are processed at varying spatiotemporal scales and has regional applicability. The data sources used in this paper have been selected based on previous research, and incorporating additional data may introduce uncertainty. Secondly, the PHPM-MDCF is implemented on a daily scale, aligned with the model's time step. This presents a significant challenge in sourcing daily data from various sources within the same spatiotemporal coverage.

A possible realization in the current study is to use different precipitation datasets (i.e., TRMM and Daymet datasets) to force the hydrological model and conduct correction, which has been implemented in Sect. 5.2.1. The results indicated that the correction is not sensitive to the choice of precipitation data.

*In summary, the above results suggest that the correction is minimally sensitive to the choice of forcing, demonstrating the robustness of the correction results.*

For the reasons mentioned above, we are currently not considering the introduction of additional products for each water component. Instead, we validated the reliability of the PHPM-MDCF by comparing with existing methods and examining the physical meaning of the omission residuals; for details, please see **RC2-3** and **RC2-8**.

**RC2-6/ In Step 2 at line 250, please explain why is it reasonable to allocate residuals based on the difference between simulated values and reference values? It is worth noting that the simulated ET (Evapotranspiration) and TWSC (Terrestrial Water Storage Change) by the hydrological model may not have been validated for accuracy and may contain significant uncertainties. If their errors are used to allocate residuals, substantial uncertainties could lead to unreasonable allocation of residuals to ET and TWSC. The formula for residual allocation needs to be supplemented. Additionally, if Step 3 determines that the residual allocation is unreasonable, can simply halving the residual solve the issue? The underlying principles need to be clarified, or an example should be provided.**

**R/** Thank you for your careful review. For clarity, we have reorganized the questions in this comment and analyzed them individually.

**(a) Why allocate residuals based on the distance between measurements and simulations?**

As we discussed earlier, in this study, the simulations from the hydrological model are considered a physically consistent system that satisfies the water balance (See **RC2-2**). Therefore, the Eq. (4) based on the simulations inevitably leads to $Res_i$ being 0. In other words, when all measurements are corrected to equal the simulations, the $Res_i$ in the measurements are corrected to 0. This determines the correction direction for measurements of each variable.

However, directly correcting the measurements to equal the simulation at once can also introduce uncertainty, as the simulation system is not precise (i.e., model parameters). Therefore, we considered an iterative approach for correction.

From the perspective of hydrological processes, the simulations reflect an ideal system that is physically consistent and strongly physically interrelated. On the contrary, the measurements reflect a system that variables are relatively loosely connected and physically inconsistent. To facilitate the convergence of the measurement system towards the ideal simulation system, it is important to determine the relative magnitude of the corrections for each water component.

The different water components cannot be corrected to the same extent, as their physical connections must be taken into account. For example, consider a region with high evaporation and low streamflow. Typically, it is reasonable to apply more correction to evaporation. However, if measurement of streamflow exhibits extreme high values, it would be more reasonable to apply more correction to streamflow. This is because our understanding of hydrological process suggests that the likelihood of such extreme high streamflow in this region is very low. Such understanding is reflected in the hydrological process, that is, in the simulations. Given this, we allocate the correction of $Res_i$ based on the distance between measurements and simulations. In other words, the greater the distance between the measurement and the expected values, the more correction we will apply. This idea is illustrated in Fig. 3.

[Figure]

**Figure 3.** Illustration of the correction process advancing convergence between the simulation and measurement systems. The measurement system is corrected to approach the simulation system, while the simulation system is refined via parameter calibration to better approximate the measurement system. As a result, the distance between the two systems is reduced, leading to better physical consistency in the corrected measurement system.

To better assist readers in understanding this idea, we have revised the statement in Step2 to (Line 270-278 in tracked version):

• Step 2: Correction for the inconsistency residuals. Allocate inconsistency residuals based on the magnitude of differences (i.e., the distance between simulation and measurement systems) between simulated and measured values for each variable in Eq. (5) and (6). This difference indicates the correction direction and magnitude for each variable, which facilitates the convergence of the measurement system toward the simulation system. Here, an initial correction rate of 0.5 is set to gradually correct the multisource datasets, thereby avoiding potential uncertainties that arise from excessive correction. Formally, the allocation of inconsistency residuals can be described by the following equation:

$$M_c^v = M_o^v - Res_i \times \frac{d_v}{d_{all}}, \tag{7}$$

where $M_c^v$ is the corrected measurements of variable $v$, and $M_o^v$ is the original measurements; $d_v$ is the difference between simulation and measurement of variable $v$, and $d_{all}$ represents the aggregate of differences for all variables.

**(b) Were the simulations of ET and TWSC validated?**

Yes, we validated the simulation results across five variables (i.e., streamflow, ET, SMS, GRS, and SWE) to ensure reliable simulations, where the SMS and GRS are used to represent TWS. We have provided a detailed explanation in our response to **RC2-2** above. Through model performance evaluation, we have ensured that all basins undergoing multisource dataset correction exhibit reliable simulation. Additionally, the simulation performance has significantly improved after correction, as evidenced by the changes in the Pareto front shown in Fig. 8.

[Figure]

**Figure 8.** Comparison of multivariable simulation performance before and after correction at basin 1013500. Light grey and dark grey indicate population solution sets before and after correction, and blue and red indicate Pareto fronts before and after correction. Metrics evaluating SWE simulation performance have been normalized for consistency. The subplot

in the second row, second column shows that the evaporation simulation maintains highly accurate at this basin, due to the alignment between the HBV algorithm and measurements.

Another strong evidence demonstrating the validity of the ET and TWSC simulations is presented in the newly added Sect. 4.3.3. It can be observed that the TWSC retrieved from SWE and SM is consistent with GRACE TWSC, and the simulated ET (i.e., corrected ET) also aligns with ET measurements from various sources. Please refer to **RC2-3** for this detail.

**(c) Supplement the residual allocation formula.**

Thank you for pointing out this. According tor your suggestion, we have added the corresponding formula as shown blow (Line 275-278 in tracked version):

Formally, the allocation of inconsistency residuals can be described by the following equation:

$$M_c^v = M_o^v - Res_i \times \frac{d_v}{d_{all}}, \tag{7}$$

where $M_c^v$ is the corrected measurements of variable $v$, and $M_o^v$ is the original measurements; $d_v$ is the difference between simulation and measurement of variable $v$, and $d_{all}$ represents the aggregate of differences for all variables.

**(d) If Step 3 determines that the residual allocation is unreasonable, can simply halving the residual solve the issue? What is the principle behind this?**

In Step 3, a judgment will be made to determine whether the previous correction was reasonable based on whether the model can provide a reliable simulation. A misunderstanding that needs to be clarified here is that if the simulation proves unreliable, we will discard the previous correction, return to Step 2, halve the correction rate rather than directly halving $Res$, and then proceed with the correction again. Naturally, after this correction, the judgment in Step 3 will be re-evaluated until the correction or inconsistency residual falls below a pre-set threshold.

In other words, this iterative process involves continual trial and error, with each error prompting us to approach the next correction more cautiously. The underlying consideration is that the convergence of the measurement system and the simulation system is a mutual process. Measurements approach the simulated system through correction, while the simulation system, through re-calibration after each correction, aligns more closely with the measurement system. As described in the process shown in Fig. 3 above. Excessive correction may lead to the measurement system going out of bounds, preventing further convergence of the two systems. Specifically, this manifests as producing unreliable simulations, and further model calibration will not enable the two system to converge.

We have noted that our expression might lead to misunderstandings; therefore, we have revised the phrasing in Step 3 to (Line 279-285 in tracked version):

• Step 3: Calibration and evaluation of the model. Recalibrate and evaluate the hydrological model using the datasets corrected in the previous step to assess the reliability of this correction. If the recalibrated model yields unreliable simulations, consider this correction excessive, halve the correction rate, and repeat Step 2. Otherwise, maintain the correction rate and proceed with the next iteration of correction. The consideration behind this step is that excessive correction may lead to the measurement

system going out of bounds, preventing further convergence of the two systems. This is to say, the iterative process involves continual trial and error, with each error prompting us to approach the next correction more cautiously.

In addition, the issue related to the selection of parameters for the PHPM-MDCF (e.g., initial correction rate, decay rate of the correction rate, and correction termination threshold), which is behind this question, will be addressed in detail in ***RC2-9.***

**RC2-7/ Please clearly state the scope and spatiotemporal scale of this study. Most studies investigate water budget closure at the monthly scale rather than the daily scale. Aside from data availability, I believe this is mainly due to larger data errors and the lag effect of hydrological processes at the daily scale. If this study focuses on water budget closure at the daily scale, how were these issues addressed?**

**R/** Your perspective is very insightful. As you commented, the scale of the water budget study is crucial. The water budget non-closure phenomenon exhibits different behaviors at varying spatial and temporal scales. It is widely recognized that achieving water budget closure is much easier at relatively larger spatial and temporal scales.

On the one hand, at lager temporal scales, the TWSC exert a smaller influence on water budget closure. In relatively long time periods, TWSC can be assumed to negligible, making precipitation approximately equal to the sum of streamflow and evaporation. This is a common assumption in water budget assessment studies when TWSC measurements are unavailable. For example, Weligamage et al. (2023) suggested a 10-year period during which changes in water storage were considered negligible. Other several studies suggested that TWSC can be disregarded at the annual scale (Cooper et al., 2011; Kauffeldt et al., 2013; Hoeltgebaum et al., 2023). On the other hand, at larger spatial scales, inter-basin water exchanges can be considered negligible (Lv et al., 2017). Therefore, in most previous studies, it has been more feasible to conduct water budget studies at larger spatial and temporal scales. Additionally, another important reason for the choice of a monthly scale in much of the prior research is the reliance on GRACE TWSC measurements, which are only available at this temporal resolution.

In this study, TWSC is represented by a combination of observed soil moisture storage (SMS), groundwater reservoir storage (SMS), and snow water equivalent (SWE), avoiding the resolution constraints of GRACE TWSC, thus can be conducted at a daily scale. This is detailed in Sect. 2.2, where the main information is as follows:

*Assuming that TWSC can be retrieved through a combination of different water storages, we obtained the four-layer soil moisture from ERA5 Land and Snow Water Equivalent (SWE) from GlobSnow to estimate overall TWSC. This approach has been implemented in the investigation of Hoeltgebaum and Dias (2023), yield a high consistency between estimated TWSC and GRACE observation (i.e., correlation coefficient exceeding 0.71). Another consideration in this method is that the decomposed TWSC products (i.e., soil moisture and SWE) can correspond to the results simulated by hydrological model, thereby allowing us to correct water budget residuals, as discussed later.*

*Overall, all datasets were resampled to a daily time step, and then aggregated over basins through simple averaging to perform analysis of water budget closure on a basin scale from 1998 to 2010.*

Although the primary temporal scale of this study is daily, we also performed statistical analyses at monthly and annual scales. For example, Figure 4-5 aggregate the residuals to the monthly scale to illustrate their spatiotemporal distribution. Figure 6 displays the correction results at daily, monthly and annual scales. This was done for both of visualization purposes and facilitating potential comparisons with previous studies.

Through a comparison of water budget at different timescales, we observed distinct behaviors of residuals across these scales. Specifically, at smaller scale (daily), residuals show greater variability but smaller magnitudes. As aggregation occurs at lager scales (monthly and annual), the magnitude increase while the variability decreases, demonstrating a filtering behavior. The primary mechanism behind such behavior is the positive and negative offset and accumulation of residuals and biases in different water components. Figure 6 provides an example to illustrate this:

[Figure]

**Figure 6**. Correction results of water budget residuals for multisource datasets at basin 1013500. (a-c) Time series of water budget residuals ($Res$), inconsistency residuals ($Res_i$), and omission residuals ($Res_o$) at daily, monthly and yearly scales, grey line represents residuals during the correction process. (d-f) Variation of long-term mean absolute values of three residuals with correction iterations at the monthly scale. The unit of residuals is "mm".

According tor your comment, we have further emphasized the temporal scope and scale used in this study by adding the following statements (Line 205-206, 249-250, 296 in tracked version):

Overall, all datasets were resampled to a daily time step, and then aggregated over basins through simple averaging to perform analysis of water budget closure on a basin scale from 1998 to 2010.

Then, the residuals are calculated at daily scale and subsequently aggregated to the monthly and annual scales for further analysis.

Notably, the correction is performed at the daily scale, aligning with the model step.

**RC2-8/ At line 320, it is necessary to explain the reasons behind the spatial distribution of Res. How does the difference in spatial patterns indicate that inconsistency residuals and omission residuals are driven by different factors? Please provide a detailed explanation. The most likely reason for Resi and Res having the same spatial pattern is that the former was calculated based on the latter.**

**Their difference from Reso is due to the different error sources used in calculating Reso and Res, which does not necessarily demonstrate the reliability of the method for separating inconsistency residuals from omission residuals. Additionally, the residual values in Figure 4 differ significantly from those reported in previous studies. What is the reason for this discrepancy?**

**R/** Thank you for your comment. For clarity, we reorganized the questions in the comment into two separate points and address each one individually.

(a) **What are the reasons behind the spatial distribution of Res? Does its distribution show significant differences compared to previous studies? If so, what are the reasons for these differences?**

This is a good question. Indeed, as we discussed in our manuscript, the spatial distribution of $Res$ in Fig. 4 exhibits very pronounced clustering characteristics.

*$Res$ and $Resi$ both present an east-west gradient for three statistical measures (i.e., min, median, max), with low values occur along the western coastline and high values primarily concentrated in eastern inland basins. The exception is a cluster of low median values located in the central CONUS.*

From a geo-statistical perspective, the spatial heterogeneity of $Res$ likely involves multiple direct and indirect influences from basin characteristics. Clarifying these potential influencing factors is crucial for understanding the formation of $Res$. Therefore, we conducted an exploratory analysis to investigate the potential factors influencing the spatial distribution of $Res$ and compared it with previous studies. This has been presented as a separate section in this revision (Sect. 4.4.1, Line 535-564 in tracked version, see below).

4.4.1 Factors influencing spatial distribution

In this section, we conducted a preliminary exploration of the potential factors influencing the formation and distribution of water budget residuals. As shown in Fig. 4, all three water budget residuals are subject to strong spatial organization, and these patterns are in agreement with previous studies. For example, Kauffeldt et al. (2013) found negative residuals (i.e., runoff coefficient > 1) along the western coastline of CONUS, while the eastern region showed notable positive residuals (i.e., P-R > ET). Other studies investigating water budget residuals with diverse dataset combinations have similarly revealed similar spatial patterns (Zhang et al., 2016; Gordon et al., 2022). Therefore, we speculate that the spatial distribution of water budget closure is predominantly influenced by the characteristics of the basin.

Here we focus on the total water budget residuals (i.e., $Res$) and attempt to relate it with the hydro-meteorological conditions and the basin area. To bring out these relationships, from Fig. 12, three regression curves are obtained by correlating mean absolute residuals at different timescale with basin areas over 475 CAMELS basins. The negative gradients of the curves imply a scale effect in the water budget non-closure phenomenon that as basin area increases, the water balance constructed from multisource datasets can be enhanced. Moreover, as expected, hydro-meteorological conditions within the basin play a crucial role in controlling the distribution of water budget residuals. The clear delineation between different levels of daily precipitation and runoff coefficient revealed in Fig. 12 strongly supports this reasoning, where multisource datasets yield larger water budget residuals in basins with high precipitation and runoff coefficients—large red spots are located in the upper portion of the figure. These results highlight the risks of using multisource datasets for hydrological inference in humid and smallscale basins—specifically, potential physical inconsistencies—and underscore the need to carefully test the water balance assumption.

[Figure]

**Figure 12**. Relationship between the mean absolute of water budget residuals, basin area, long-term average daily precipitation, and runoff coefficient (RC) over 475 CAMELS basins with reliable simulations. The respective red lines represent the linear regression of residuals with basin area for each timescale.

We have found that $Res$ is closely related to basin area and hydro-meteorological conditions. Specifically, we found that achieving water budget closure with multisource datasets is more challenging in large and humid basins (characterized by high precipitation and runoff coefficient). Figure 12 provide the corresponding evidence.

Additionally, the comparison of the spatial distribution of $Res$ with previous studies is also presented in Sect. 4.4.1. The results indicate that the pattern of $Res$ identified in this study is consistent with previous research (Line 537-541 in tracked version).

We noticed a loose connection between Sect 4.1 and Sect 4.4; thus we added the following statement in the former section to strengthen the linkage between the two sections (Line 368-369 in tracked version):

The potential factors affecting the spatiotemporal distribution and proportion of $Res$ will be further investigated in Sect. 4.4.

In summary, we divided Sect. 4.4 into three subsections to ensure a clear structure. The titles of the three subsections are:

4.4.1 Factors influencing spatial distribution
4.4.2 Factors influencing temporal distribution
4.4.3 Factors influencing the proportions of residuals components

**(b) Why are the differences between the spatial patterns of Resi and Reso driven by different factors? What is the theoretical basis for residual decomposition? How can the reliability of this decomposition be demonstrated?**

In previous studies, $Res$ (water budget residuals) have typically been used as a whole to measure the degree to which the measurements achieve water budget closure. The cause of $Res$ is often simply attributed to inconsistencies in the processing of different products (refer to the review provided by Lv et al., 2017). Few studies have thoroughly discussed the causes of $Res$ formulation.

An exception is the study by Gordon et al., (2022), where they qualitatively decomposed $Res$ into data inconsistency error ($e$) and groundwater exchange ($G$) not accounted for in the water budget equation (see Eq. (2)). We extended Eq. (2) to incorporate additional source of potential water omission, and further attempted a quantitative decomposition of $Res$ into $Res_i$ and $Res_o$ to elucidate the distinct factors contributing to the observed water budget non-closure.

In our opinion, using measurements to describe the theoretical water balance requires two key conditions: (1) physically consistent measurements, and (2) comprehensive description of the water budget equation. Correspondingly, the causes of water budget non-closure ($|Res| > 0$) can be attributed to two factors: (1) physical inconsistency in the measurements ($Res_i$), potentially arising from discrepancies in data production process mentioned in previous studies; and the incomplete description of the water budget equation ($Res_o$).

Indeed, as you noted, the decomposition of $Res$ is fundamentally based on the following sample equation, which capture the essence of our decomposition method:

$$Res_i = Res - Res_o \qquad\qquad (\text{RC2-8})$$

However, the similar spatiotemporal distribution of $Res_i$ and $Res$ cannot be simply attributed the calculation. Essentially, this similar pattern is attributed to the relative small proportion of $Res_o$, suggesting that our description of the water budget equation is comparatively comprehensive.

Consider that if our description of the water budget equation were incomplete and omitted a significant water component, $Res_o$ would likely exert a greater influence on $Res$, resulting in a more pronounced discrepancy between $Res$ and $Res_i$.

To examine this, we intentionally exclude the SWE component from the water budget equation to access its impact on the decomposition of $Res$. This is a plausible scenario in practice, as it is likely that this component was not considered when reconstructing the TWSC. This experiment is conducted and analyzed in the new Sect. 4.4.3 (Line 596-629 in tracked version, which is also attached in the end of this response).

The results indicate that the proportion of $Res_o$ obtained from residuals decomposition after excluding SWE increases significantly, with this effect being more pronounced in high-latitude regions, high elevations, and during the cold season (see the revisions and figure below). This is consistent with physical principles, as the impact of omitting SWE on water balance is greater under these situations. These findings align with our definition of $Res_o$ which refers to the water imbalance caused by omitted water. It also, to some extent, supports the validity of our decomposition method, and highlights the importance of a comprehensive water budget equation.

The added new Sect. 4.4.3, which is used to explain the potential factors for the proportion of $Res$ components, is shown below:

**4.4.3 Factors influencing the proportions of residuals components**

Another interesting finding in Sect. 4.1 is that the magnitude of $Res_o$ is significantly smaller than that of $Res_i$. As a result, $Res$ is dominated by $Res_i$, leading to a highly consistent spatiotemporal distribution between them. However, the underlying question is what this implies and which factors drive the proportions of the residuals components.

$Res$ reflects the degree to which the measurements achieve water budget closure. In this study, we argue that two key conditions are necessary for using measurements to describe theoretical water balance. The first one is that measurements of different water components must be physically consistent. In practice, however, this condition is often challenging to meet due to inconsistencies and uncertainties in data production processes from different sources, which can result in non-zero $Res_i$ (Luo et al., 2020). The second crucial, yet frequently overlooked, condition is the completeness of the water budget equation. Building on the work of Gordon et al. (2022), we developed a more generalized water budget equation (Eq. (3)) and use $Res_o$ to account for the water imbalances caused by omitted water. From this perspective, $Res$ results from the interplay between $Res_i$ and $Res_o$, either through their accumulation or mutual cancellation. Therefore, the low proportion of $Res_o$ essentially suggests that our description of the water budget equation is comparatively comprehensive.

Consider that if our description of the water budget equation is incomplete and omits a significant water component, $Res_o$ would likely exert a greater influence on $Res$, resulting in a more pronounced discrepancy between $Res$ and $Res_i$. To examine this, we intentionally exclude the SWE component from the water budget equation to evaluate its impact on the decomposition of $Res$. This is a plausible scenario in practice, as it is likely that this component was not considered when reconstructing the TWSC. Figure 14 illustrates the comparison between $Res_o$ derived from the decomposition method excluding SWE (hereafter $Res_o^{NSWE}$), and its original values. It is evident that $Res_o^{NSWE}$ exhibits greater variability compare to the original values (i.e., with smaller minimum values and larger maximum values). The median differences indicate that the likelihood of increased omission residuals is higher after excluding SWE (Fig. 14b). Such differences reveal that omitting crucial SWE storage component results in a greater degree of water imbalance, and, as expected, this effect is more pronounce in high-latitude and high-elevation regions (Fig. 14d-f). Moreover, the spatiotemporal distribution of $Res_o$ has changed (Fig. S13-14). Notably, during the cold season (December to February), the proportion of $Res_o$ is much higher and exhibits a significant positive trend. These findings align with our definition of $Res_o$, which refers to the water imbalance caused by omitted water. It also supports the validity of our decomposition method to some extent, and highlights the importance of a comprehensive water budget equation in evaluating water balance.

[Figure]

**Figure 14.** Comparison of $Res_o$ obtained from residuals decomposition excluding SWE with the original values. (a-c) Spatial distribution of monthly mean $Res_o$ excluding SWE minus its original values. (d-f) Time series of $Res_o$ excluding SWE and its original values at the southern basin (02198100, 32.96°N), northern basin (12358500, 48.33°N), and high-elevation basin (07083000, elevation of 3.56 km) at monthly scale. The unit of residuals is "mm".

[Figure]

**Figure S13.** Same as Fig. 4, but for residuals decomposition excluding SWE.

[Figure]

**Figure S14.** Same as Fig. 5, but for residuals decomposition excluding SWE.

**RC2-9/ In the multi-source dataset correction framework for achieving water budget closure, what is the rationale for setting the initial correction rate to 0.5? Why is the correction rate halved when the model produces unreliable simulations? Is there a potential proportional relationship between the adjustment of the correction rate and the magnitude of bias in unreliable simulations that could allow for more efficient correction rate adjustments? Additionally, what is the basis for setting the conditions for iteration and termination of the correction process as "the inconsistency residuals decreases to 10% of its initial value or the correction rate falls below 4%"?**

**R/** This is a very insightful comment. What you mentioned are precisely three key issues we encountered during the implementation process. Just in our response to the fourth question in **RC2-6**, the iterative process involves continuous trial and error to prevent over-correction and ensure that measurement remain within the appropriate range.

The first issue is determining the initial correction rate ($r_0$). At the beginning, to ensure a high correction speed, we set the initial correction rate to 1 and 0.7. However, for most basins, this often resulted in measurements exceeding a reasonable range after the first iteration of the correction, leading to unreliable simulations and unreasonable corrected measurements. Through experimentation, we found that 0.5 is a suitable initial correction rate, as it ensures that the first iteration of the correction is effective in most cases.

The second key issue is determining the decay rate of correction rate ($\Delta r$) following the occurrence of unreliable simulations. The generation of unreliable simulations suggests that the current correction is excessive. Effectively reducing the correction magnitude and re-correcting may further facilitate the convergence of measurement system with the simulation system. Linear decay is a conventional approach, which aligns with our perception. For example, reducing the correction rate by 0.1 or 0.2 each time. However, testing has shown that such linear decay results in excessively long correction times, making the application of the PHPM-MDCF across a wide range of basins (i.e., 475 basins) difficult. On the other hand, exponential decay can cause the correction rate to quickly fall into a small value range, thereby reducing the correction efficiency. Given the above, we chose a multiplicative decay approach, where the

correction rate is halved each time for re-correction. The results indicate that this approach is effective, as shown in the iterative process depicted in Figures 6 and S3-6. For illustration, we provide a case here:

[Figure]

**Figure RC2-9.** The decline of *Res* with the number of correction iterations for basin 1013500. The unit of residuals is "mm".

The final issue is determining when to terminate the correction, as this criterion significantly affects the final correction efficiency. Here we consider two points.

(a) The first is that the correction has achieved satisfactory results, with the final *Res* being relatively small ($Res_t$). This threshold must be appropriately set; it cannot be too large, as this would indicate insufficient correction, nor too small, since the PHPM-MDCF, as a soft constraint, has limited correction capacity. An excessively small final *Res* threshold could result in an infinite number of correction iteration. Based on comparative experiments, we believe that reducing it to 10% of the initial value is appropriate. As shown in Fig. RC2-9, *Res* stabilizes and no longer changes once it decreases to around 10% of the initial value (from 40 to 4 mm).

(b) The second point is that the correction rate should not be too small, as this would imply excessively low calibration efficiency. This is closely related to the initial correction rate and decay rate (here, 0.5 and halving, respectively). A threshold of 4% means that the correction will cease once the correction rate, decayed four times from 0.5 to 0.03125, is reached. This threshold setting is relatively subjective, but it has proven to be reasonable based on testing results.

Notably, although the parameters for the three issues mentioned above are set subjectively, the choice follow a certain logic and have passed a series of tests. At least, cautiously speaking, they are suitable for the current study area, as shown in Fig. 7. Further adjustments are possible, but they have minimal impact on the current results (based on testing).

We have added the following statement in Sect. 3.2 to further emphasize the issues mentioned above (Line 297-300 in tracked version):

Notably, the correction is performed at the daily scale, aligning with the model step. In the subsequent application of the PHPM-MDCF, the measurements are derived from the data provided in Sect. 2.2. In addition, through experimentation, the parameter settings in the PHPM-MDCF (i.e., initial correction rate, decay rate of the correction rate, and correction termination threshold) have been tailored to suit the

current study area (Table S2). When applying this framework to other regions, additional adjustments and testing may be required.

**Table S2.** Summary of the parameters settings in the PHPM-MDCF.

| Parameters | Reference value | Reference range | Description |
|---|---|---|---|
| $r_0$ | 0.5 | 0.3~0.6 | Initial correction rate. |
| Decay approach | Multiplicative | Linear, exponential, and multiplicative decay | The method of reduction in correction rate following an unreliable simulation. |
| $\Delta r$ | 50% | 30%~70% | Decay rate of the correction rate. |
| $Res_t$ | 10% | 5%~20% | Correction termination threshold for inconsistency residuals. |
| $r_t$ | 4% | 1%~10% | Correction termination threshold for correction rate. |

**Minor Comments:**

**RC2-10/ Please provide additional explanation on how Section 4.3.1 demonstrates the reliability of the PHPM-MDCF method.**

**R/** Thank you for your suggestion. We have added scatter plots comparing measurements and simulation before and after correction to further illustrate the convergence of the measurement and simulation systems, thereby demonstrating the reliability of the PMPH-MDCF method. The following revisions have been added to Section 4.3.1 (Line 430-432 in tracked version).

More intuitively, Fig. S7 presents a comparison of measurements and simulations for each variable before and after correction. It is evident that the relationship between measurements and simulation is significantly strengthened after correction.

[Figure]

**Figure S7.** Scatter plots comparing measurements and simulation before and after correction at basin 1013500.

**RC2-11/ The paper does not validate the accuracy of the Reso, Resi, and Res separation method in the results.**

**R/** Thank you for your comment. We have addressed this issue in detail in our response to the second question of ***RC2-8*** and included a new subsection (Sect. 4.4.3) to demonstrate the reliability of the residuals decomposition method. Please review the response above.

**RC2-12/ At line 310, can KGE ≥ −0.41 really indicate that the hydrological model accurately represents the observed hydrological system?**

**R/** Thanks for your comment. The Kling-Gupta Efficiency (KGE) metric, introduced by Gupta et al. (2009), provides a method for achieving a balanced improvement of simulated mean, variability, and correlation (see Eq. B1). Many studies have demonstrated the effectiveness of KGE, which is currently a popular metric in hydrological modelling (Knoben et al., 2020; Clark et al., 2021). The KGE is bound by $(-\infty, 1]$ with 1 being the ideal value. For such a metric, it is challenging to give a benchmark value to determine whether the simulation is reliable. Thus, to ensure caution, we opt to reference previous literature for guidance. For instance, Aerts et al. (2022) use the -0.41 of KGE as the benchmark to evaluate the performance of wflow_sbm in simulating streamflow:

*Ideal model performance has a KGE score of 1 and a KGE score of −0.41 is equal to taking the mean flow as a benchmark.*

Bruno et al. (2002) noted that a KGE of -0.41 serves as the threshold for no skill:
*(KGE $\in$ (-∞, 1], optimal value = 1, no-skill threshold over mean flow as predictor = -0.41).*

The notable example is Knoben et al. (2019), who, by comparing the NSE and KGE metrics, established a KGE value of -0.41 as the threshold for evaluating whether model simulations outperform the mean flow:

*Here we show that using the mean flow as a predictor does not result in KGE = 0, but instead KGE =1- √2≈-0.41. Thus, KGE values greater than −0.41 indicate that a model improves upon the mean flow benchmark – even if the model's KGE value is negative.*

Based on the aforementioned literature, we used a KGE value greater than -0.41 as the threshold for reliable simulations. Although this threshold may still be somewhat subjective, evaluating simulation reliability across five variables (i.e., streamflow, ET, SMS, GRS, SWE) simultaneously can help mitigate this uncertainty.

For better address the question, we have included the above references in the manuscript (Line 343-345 in tracked version).

With reference to previous studies (Knoben et al. 2019; Clark et al., 2021; Aerts et al., 2022), we have adopted KGE≥-0.41 and r statistically significant at the 5% level as criteria for guaranteeing reliable simulations.

**RC2-13/ In Figure 5, Reso is closer to 0. Can we attribute this to the principle of water budget in the development of the hydrological model, rather than merely to omission errors? Since Resi = Res - Reso, and Reso is relatively small, it is evident that the values and spatial patterns of Resi and Res are more similar. What does this imply?**

**R/** Thank you for your comment. Our response to the second question of **RC2-8** provides some clarification on this issue. Specifically, $Res_o$ approaching zero indicates that our description of the water budget equation is relatively comprehensive and cannot be simply attributed to the water balance features of the hydrological model.

When the SWE component is omitted without changing the model, $Res_o$ increases significantly, with this effect being more pronounced at high elevations, high latitudes, and during the cold season (Fig. 14).

The equation ($Res_i = Res - Res_o$) is indeed the essence of our decomposition method, but it is not the sole reason for the similarity between $Res_i$ and $Res$. The fundamental reason lies in the completeness of the water budget equation description, which results in a smaller contribution of $Res_o$ to the formation of $Res$.

**RC2-14/ Please explain from a theoretical standpoint why the PHPM-MDCF method has such advantages over previous methods: "It suggests that the soft constraints based on physical hydrological processes will not lead to compensatory errors, as seen in traditional methods due to the rigid allocation of water budget residuals.".**

**R/** Thank you for your suggestion. We have added the following statement to theoretically demonstrate the advantages of the PHPM-MDCF (Line 468-473 in tracked version).

From a theoretical perspective, the PHPM-MDCF assigns the weights of residual correction based on the distance between measurements and simulation for each variable. In the presence of a single extreme bias, the large distance between the measurement and simulation of the corresponding variable leads to a larger correction being applied to that variable, while the weights for other variables remain unaffected. However, in traditional methods, the correction weight for each variable remain constant over time, and the final residuals are constrained to zero. This leads to the propagation of extreme biases across different variables.

**RC2-15/ I do not find this statement reasonable: "When the hydrological model calibrated against multiple variables measured by the multisource datasets and achieves reliable performance, we consider the simulation system approaching the measurement system.".**

**R/** Thank you for your comment. We have revised this inappropriate statement to (Line 234-236 in tracked version):

When the hydrological model calibrated against multiple variables measured by the multisource datasets and achieves reliable performance, we consider the water budget represented by the simulation and measurement systems to be comparable.

**RC2-16/ At line 255, please clarify the data sources for the observed values of P, ET, Q, and TWSC used in this study. Without this information, it is difficult to judge whether the deviation between the simulation system and the measurement system is calculated reasonably.**

**R/** Thank you for pointing out the unclear aspects of our manuscript. According to your suggestion, we reiterated the data sources (see our response in *RC2-4*) and further emphasized them in this section as follows (Line 296-297 in tracked version):

In the subsequent application of the PHPM-MDCF, the measurements are derived from the data provided in Sect. 2.2.

**RC2-17/ I personally feel that the discussion in Section 5.1 would be more effective if it were more closely aligned with the scope of this study.**

**R/** Thank you for your suggestion. According to your suggestion, we enhanced Sect. 5.1 with more arguments relevant to this study and reduce unnecessary statements. The revisions are as follows:

Remove this sentence from the penultimate paragraph (Line 669-671 in tracked version):

The last paragraph is revised to (Line 673-683 in tracked version): The proposed correction framework (PHPM-MDCF) capitalizes on this concept by iteratively advancing the convergence between the knowledge system (i.e., hydrological model and water balance equation) and the measurement system, thus enhancing the credibility of the measurements. Although our current knowledge may not be entirely precise—for example, the depiction of hydrological processes in models may lack accuracy—it remains foundation upon which we can rely and strive to refine in the future. Furthermore, several underlying concepts in this framework, such as residuals decomposition and advancing water budget closure through correction, aligns with a recent study (Wang and Gupta, 2024). They introduced a novel hybrid model (i.e., Mass-Conserving-Perceptron) and discussed its potential application, including the bias correction (lacking confidence for the measurements) and examination of non-observed interactions with the environment (corresponding to the omission errors). Coupling the PHPM-MDCF with hydrological models that provide stronger interpretability is a valuable and promising research effort, as it can offer insights into the physical attribution of water budget non-closure and enable more reasonable correction.

**RC2-18/ The limitations discussed in Section 5.2 are not explained from a theoretical perspective. I hope that some convincing explanations can be supplemented from this standpoint.**

**R/** Thanks for your suggestion. We added the following statement to Sect. 5.2 to further explain the theoretical basis of the adaptability to forcing datasets of the framework (Line 703-709 in tracked version).

Theoretically, the consistency of correction stems from two aspects. Firstly, it is attributed to the adaptability of hydrological model to the input data, specifically the calibration compensation capability we described in the introduction (Wang et al., 2023). This enables the hydrological model to generate

reasonable representation of hydrological process even with imprecise forcing. Secondly, as discussed in Sect. 4.3.2, the PHPM-MDCF serves as a soft constraint and utilizes the distance between measurements and simulations to allocate residuals correction, thereby mitigating the propagation of bias between variables. These two features ensure that stability of the correction, rendering it less susceptible to interference from uncertainties in the forcing datasets.

**RC2-19/ The structure of the article lacks a keywords section. Please add keywords.**

**R/** Thank you for your careful review. According to the current HESS official template and guidelines, the keywords section is not a required option. Please the following URLs:

https://www.hydrology-and-earth-system-sciences.net/submission.html#templates
https://www.hydrology-and-earth-system-sciences.net/submission.html#manuscriptcomposition

**RC2-20/ Please add references related to the water budget equation.**

**R/** Thank you for pointing out the omissions in our manuscript. We have added the relevant reference (Lehmann et al., 2022) for Eq. 1 as (Line 52-55 in tracked version):

For a closed basin, the water budget can be mathematically expressed as (Lehmann et al., 2022),

$$\frac{dTWS}{dt} = P - ET - R, \tag{1}$$

where $\frac{dTWS}{dt}$ is change in terrestrial water storage, P is precipitation, ET is evaporation, R is runoff at the outlet.

**RC2-21/ The text states "as illustrated in Fig. 3" but the caption provided is "Figure 3". The authors should ensure that all figure captions are consistent with the text descriptions. Please carefully check the rest of the article for similar errors and make the necessary corrections.**

**R/** Thank you for your careful review. For the abbreviation format, we referred to the official guidelines provided by HESS. Please see the following URL and explanation:

https://www.hydrology-and-earth-system-sciences.net/submission.html#figurestables

*Figure composition: ...*
*...*
*The abbreviation 'Fig.' should be used when it appears in running text and should be followed by a number unless it comes at the beginning of a sentence, e.g.: "The results are depicted in Fig. 5. Figure 9 reveals that.*

**In the end, I would like to express my sincere gratitude for your prompt reply and for the time and effort you have so generously devoted to reviewing our paper.**

**Responses to RC3:**

We sincerely appreciate your thorough review and constructive discussion of our paper. All comments from RC3 are addressed below with point-by-point responses.

**RC3-1/ The author's approach of studying water balance closure from the perspective of physical mechanisms does indeed have academic value.**

**R/** Thank you very much for recognizing the value of our work; this is a great encouragement for us.

**RC3-2/ However, the core issues I raised have not been fully addressed. The author mainly provided some explanations without offering experimental evidence to demonstrate the reliability of the proposed method.**

**R/** In this response, we have adopted your suggestions, given them careful consideration, and made every effort to conduct related experiments within the limited time available. Detailed experimental results are provided in *RC2-3*, presented as a new subsection that has been added to the manuscript.

**RC3-3/ I maintain that a comparison with existing methods is necessary to validate the accuracy and reliability of the proposed approach. The purpose of achieving water balance closure has two main components: improving data consistency and accuracy. Regarding data consistency, the author's method does not fully achieve water budget closure (I agree with the principle behind the author's approach). Therefore, if the method's performance cannot be verified in terms of data accuracy, its overall effectiveness and reliability remain questionable. I recommend that the author select some representative basins with measurements of budget components for validation.**

**R/** We are very pleased that you recognize the principles behind our methods, and we greatly appreciate the valuable suggestions you have provided. As you mentioned, further validating the calibration results through comparisons with existing methods can emphasize the reliability of our proposed approach. The approach of selecting representative basins for validation is also feasible, therefore we proceed with experiment in this regard. Detailed results about the experiment are provided in *RC2-3*.

**RC3-4/ As for the author's claim that a comparison with existing methods is not appropriate, I disagree. Some current methods estimate the distribution weights of water imbalance based on fused values (some methods are not such as PR and MCL), rather than using the fused values as exact reference points. I recommend validating the proposed method by comparing it with existing methods based on in-situ measurements of budget components (in regions with in-situ measurements, such as P and Q). Additionally, considering multiple datasets for each hydrological variable would be beneficial for validating the proposed method. The author argues that errors in hydrological model simulations only represent physical inconsistency errors, while datasets capture comprehensive errors. If multiple datasets consistently identify omission errors, this would demonstrate the reliability of the method. I recommend that the author select some representative basins for validation.**

**R/** We acknowledge your perspective. Considering multi-source data for each hydrological component,

along with comparisons of the corrected results from existing methods, will effectively demonstrate the reliability of our approach.

Therefore, we collected multisource datasets from in-situ observations, remote sensing retrievals, and model simulations. This includes 11 precipitation, 14 evaporation, 11 streamflow and 2 terrestrial water storage datasets (see Table S3). We have implemented two existing correction methods: the PR and CEnKF methods (Luo et al., 2023).

A new subsection has been added to the manuscript to clarify the comparison between the PHPM-MDCF and existing methods (Sect. 4.3.3, Line 496-533 in tracked version). This revision is also attached at the end of this response.

In general, the comparison results from several representative basins indicate that the PHPM-MDCF can produce reliable correction results, reflected in several aspects: (1) a consistent over trend with existing method; (2) the absence of unreasonable corrections in streamflow; (3) the correction was also applied to TWSC (compared to CEnKF); and (4) a good consistency between the retrieved TWSC (from SM and SWE change) and GRAEC TWSC.

For details about the experiment, see ***RC2-3***.

**RC3-5/ Finally, the observational data referenced by the author is not in-situ measurements, and attention should be given to the terminology used.**

**R/** Thank you for pointing this out, we have emphasized the scope of this term's usage in this revision. The following content has been added to the data description section (Line 175-176 in tracked version).

Notably, the term "measurements" referred in this work are derived from multisource datasets and do not specifically refer to in-situ measurements.

**We are very pleased to engage in academic discussions with you, which are highly meaningful and significantly improve the quality of our paper.**

**Responses to CC1:**

Thank you very much for your interest in our paper and for taking time and effort to review it. All comments from CC1 are addressed below with point-by-point responses.

**CC1-1/ In the context of fast development of measurement techniques, it is our mission to develop methods to leverage the advantages of the measured variables and thus promote the hydrological simulation. This study is a valuable try, which proposed a multisource datasets correction framework, the PHPM-MDCF, to achieve water budget closure with calibration of various variables. This experiment was carried out in 475 COUNS basins, showing great potential to reduce the inconsistency residuals.**

**R/** We appreciate your recognition of the importance of our work. We hope that this paper can contribute to the data foundations in various fields, such as earth system science and hydrology, within the context of big data. Your comments are very valuable in enhancing the quality of our manuscript. Below, we will provide point-by-point responses to these comments and make the corresponding revisions in the manuscript.

**Major concerns:**

**CC1-2/ There are PTRMM in both Eq. (5) and (6), then how do we reduce the inconsistency residuals brought by P in the water budget?**

**R/** This is a crucial point, but cannot be solved within the current framework. As we have assumed that "*(2) the uncertainties associated with the model forcing and structure can be considered negligible during the modelling process*" in the methods section. In fact, the PHPM-MDCF employs the distance between simulations and measurements to allocate residuals corrections among variables. As a forcing or boundary condition, precipitation cannot be corrected within this framework, or in other words, it cannot be simulated.

Nevertheless, we consider that the uncertainty, or residuals, in precipitation has a minimal impact on the correction of other variables measurements. Some evidences are provided in Sect. 5.2.1, where a comparison of correction results under different precipitation forcing (i.e., TRMM and Daymet) reveals that the correction shows minimal sensitivity to the precipitation forcing.

*In summary, the above results suggest that the correction is minimally sensitive to the choice of forcing, demonstrating the robustness of the correction results.*

Theoretically, such behavior stems from the adaptability of hydrological model to the input data, specifically the calibration compensation capability we described in the introduction (Wang et al., 2023). This enables the model to generate reasonable representation of hydrological process even with imprecise forcing.

However, can the current results offer any guidance or insights for precipitation correction? The answer is affirmative. It is the comparison of corrections with different precipitation presented in Sect. 5.2.1 that highlights the impacts associated with varying precipitation inputs. Starting from this point, we can

discern some potential clues.

It is evident that different precipitation products do not impact the correction of inconsistency residuals (Fig. 15c-d) but do results in varying omission residuals (Fig. 15e). On the one hand, discrepancies in precipitation products are compensated by model calibration, result in similar representation of hydrological process and thus similar inconsistency residuals corrections.

[Figure]

**Figure 15**. Comparison of correction results based on different forcing datasets (TRMM and Daymet) at basin 1013500. (a-b) Corrected time series of five water budget variables. (c-e) Variation of long-term mean absolute values of three residuals with correction iterations at the monthly scale. The unit of residuals is "mm".

On the other hand, the precipitation products exhibit a systematic bias. In particular, Daymet reports significantly lower precipitation in this basin compared to TRMM (see Fig. S15 below). Such bias will manifest in the water budget equation, leading to different total input water volumes. Consequently, with the inconsistency residuals of other variables unchanged, maintaining the water balance would require an increase in $Res_o$ (Fig. 15e). Note that the $Res_o$ presented in Fig. 15e represents the mean of absolute values.

Fig. S15 was added to our manuscript along with the corresponding explanation (Line 698-699 in tracked version).

The comparison of the two precipitation products is presented in Fig. S15, where Daymet precipitation is significantly lower.

[Figure]

**Figure S15.** Comparison of TRMM and Daymet precipitation products.

Therefore, it can be inferred that, with other variables unchanged, TRMM precipitation demonstrates superior water budget closure compared to Daymet precipitation, which contains larger inconsistency residuals. This difference in inconsistency residuals is directly reflected in the variations in omission residuals after correction (Fig. 15e). In other words, this portion of the omission residuals (i.e., the difference between the two omission residuals after correction) can be directly corrected in the precipitation.

Note that not all omission residuals can be corrected in the precipitation data, as it still contains residuals from some unknown omitted water content. In other words, such correction must be relative and based on comparisons between different precipitation products, as the true values and perfect water balance equation are unattainable. Only through comparisons can the discrepancies in $Res_o$ arising from precipitation inconsistencies be identified.

To focus the attention of this paper on the PHPM-MDCF framework, we have not conducted actual experiments here. Instead, we introduced is idea in the discussion section (Line 725-739 in the tracked version):

It is noted that the PHPM-MDCF has limitations in addressing inconsistency residuals in forcing. The reasons are twofold. On the one hand, this is due to our neglect of uncertainties in the forcing, which, as indicated by the above analysis, appears to have limited impact on the correction for other variables. On the other hand, this is because the PHPM-MDCF allocates residuals based on the distance between simulations and measurement, while the forcing cannot be simulated within the hydrological model. In this case, is there a potential to correct the inconsistency residuals in the forcing? Clues to this possibility are hidden in the above analysis. Systematic biases in precipitation products are directly reflected in the water budget equation, leading to different total input water volumes. Consequently, with the inconsistency residuals of other variables unchanged, maintaining the water balance would require an increase in omission residuals (Fig. 15e). Therefore, it can be inferred that, with other variables unchanged, TRMM demonstrates superior water budget closure compared to Daymet, which contains smaller inconsistency residuals. In other words, the difference in the two omission residuals reflects the discrepancy in inconsistency residuals contained within the two precipitation products. This portion of the omission residuals difference can be directly corrected in the precipitation. However, it is worth noting that not all omission residuals can be corrected in the precipitation, as it still contains residuals from some

unknown omitted water content. Such correction must be relative and based on comparisons between different precipitation products, as true values and perfect water balance equation are unattainable. In subsequent work, we will explore the approach and try to extend the PHPM-MDCF based on this idea.

**CC1-3/ This paper focuses on the terrestrial water balance (Eq. (1)). However, whether this framework is applicable to broader water balances, such as atmospheric water balance or local water balance, or if any adjustments are needed?**

**R/** The ideas in your comment are very interesting. Although this paper primarily focuses on terrestrial water cycle systems, exploring broader water balance applications is highly valuable for extending the scope of this research.

Through a review of the literature, we found several water balance equations designed for other systems. For example:

- The steady-state hydrological budget equation of the proglacial zone (Cooper et al., 2011):

$$W_{PZ} = W_P + W_R - W_E - W_{SSS} - W_{SR} \pm \Delta W_S, \qquad \text{(RRCC1-31)}$$

    where $W_{PZ}$ is the net proglacial water flux, $W_P$ is the precipitation water flux, $W_R$ is the channel recharge water flux, $W_E$ is the evaporation water flux, $W_{SSS}$ is the sub-surface seepage water flux, $W_{SR}$ is the surface runoff water flux, and $\Delta W_S$ is the change in water storage.

- The atmospheric water vapor budget with a focus on the oceans (Penning et al., 2021):

$$\frac{\Delta W}{\Delta t} = E - P - \nabla \cdot (vq), \qquad \text{(RRCC1-32)}$$

    where $W$ being the total column water vapor and $\nabla \cdot (vq)$ the moisture flux divergence.

- The coupled atmospheric–terrestrial water balance equation (Lorenz et al., 2014):

$$\frac{\Delta W}{\Delta t} + \nabla \cdot Q = ET_a - P, \qquad \text{(RRCC1-33)}$$

    where $W$ denotes the total column water content in the atmosphere and $\nabla \cdot Q$ is the net balance of moisture flux (i.e., moisture flux divergence).

Regardless of the water balance system under consideration, the key to applying the PHPM-MDCF is whether the utilized model can represent the components of the water budget equation. The core principle of the PHPM-MDCF is to characterize the physical relationships among water budget components through the model, thereby imposing closure constraints on the measurements. As we noted in the last paragraph of Sect. 4.3.2:

*The physical relationships among various water budget variables, as representation by the model, are also imposed onto the measurements through the correction process. This reflects the core principle of PHPM-MDCF.*

In other words, the application of the PHPM-MDCF to more complex systems to conduct correction can be achieved by replacing the hydrological model (HBV) in the framework with other more suitable

models that can output more variables, such as physically distributed models (VIC model; Liang et al., 1994), coupled models (WRF-TOPMODEL; Rogelis and Werner, 2018), or even deep learning models (MCP; Wang and Gupta., 2024).

We added a statement at the end of Sect. 5.2.2 to emphasize this issue (Line 787-790 in tracked version).

By employing models that generate additional output variables, we can more comprehensively represent the water budget equation and extend the application of the PHPM-MDCF to more complex water budget systems. Additionally, multiple models can be utilized for "ensemble correction", which aids in quantifying uncertainty and providing more robust correction results.

**CC1-4/ Uncertainty plays a crucial role, and this study qualitatively address the uncertainty associated with the model structure. A pertinent question is whether this uncertainty can be quantified. While we know that validating this uncertainty through multiple models may be both challenging and unnecessary within the scope of the current work, it would be valuable if the authors could suggest potential avenues for future research and development.**

R/ Thank you for your insightful comment. We would like to address this question from the perspective of Bayesian philosophy. In practical Bayesianism, all models are inherently flawed, yet each model can be assigned a level of confidence that indicates the degree to which we trust it (Hoang, 2020). Only by considering more than one theory and model can we more effectively approach the truth. This is also the core idea of the Beven's Alternative Blueprint (Beven, 2002). As they stated:

*Why should there be any expectation of a single 'real' description when the direct observation of the responses of the most important part of hydrological systems is quite beyond our current capabilities and will be until there is a dramatic improvement in the available geophysical techniques?*

*The fact that there may be no unique answer does not mean that the approach is not science or scientific. Indeed, such an approach has then the additional advantage that we will work more naturally with the many potential worlds of future (and therefore unknown) boundary condition scenarios and the uncertain predictions that should ensue (e.g. Cameron et al., 2001).*

We strongly align with this scientific perspective. The "uncertainty" should be regarded as varying descriptions of the assumed "truth" and the associated confidence levels. Relying on a single theory alone is insufficient.

Due to limitations in data and resources, this study employs only one model for measurements correction, and we acknowledge that this introduces uncertainty. In future work, an effective method for quantifying uncertainty is to use an "ensemble" approach. Specifically, employing multiple models (theories) to describe the same hydrological process enables the range of ensemble corrections to be used for quantifying uncertainty. This is very similar to the ensemble forecasting (Nicolle et al., 2014). Additionally, confidence can be assigned to each correction based on the simulation accuracy of each model, resulting in a unique weighted correction outcome.

The description of the possible approach has been added into the Sect. 5.2.2 as follows (Line 788-790 in tracked version):

Additionally, multiple models can be utilized for "ensemble correction", which aids in quantifying uncertainty and providing more robust correction results.

**Minor comments:**

**CC1-5/ Please check carefully of the text, to avoid grammatic errors, e.g. km2 in Line 183.**

**R/** Thank you for your thorough review. We have conducted a comprehensive check and made revision throughout the manuscript. Below is an example of the revision made (Line 193-194 in tracked version).

However, the assumption is fragile when applied to small basin, leading to significant uncertainty in estimating TWSC for basins with areas less than $63,000 \text{ km}^2$.

**CC1-6/ Line 75-76: The semantics are repetitive; it is recommended to delete "to ensure data consistency".**

**R/** Thanks for your comment. We have deleted the redundant expression. The revised sentence is as follows (Line 76-78 in tracked version):

Other approaches, such as post-Processing Filtering technique (PF) and bias correction method (Munier et al., 2014; Weligamage et al., 2023), can also be helpful in closing water budget.

**CC1-7/ Line 80: "residuals" is more precise than "bias".**

**R/** Yes, thank you for pointing that out. The revised sentence is as follows (Line 78-80 in tracked version):

However, the closure constraints imposed by the above methods (hereafter referred to as traditional methods) have been questioned, with Abolafia-Rosenzweig et al. (2020) arguing about the potential incorrect assignment of residuals.

**CC1-8/ Line 131-134: It seems that these sentences should be changed to the past tense.**

**R/** Thank you for your comment. We have revised the sentence as follows (Line 134-137 in tracked version):

Furthermore, we developed a multisource datasets correction framework based on decomposition of water budget residuals and multi-objective calibration within hydrological modeling. The presented framework, providing the capability to enhance the water budget closure and hydrological connections among multisource datasets, was applied to a large-sample basins dataset across CONUS.

**CC1-9/ Line 165: "One of the main aims" might be more appropriate.**

**R/** Thank you for your suggestion. We have revised the manuscript according to your suggestion (Line 169-170 in tracked version):

One of the main aims of this study is to investigate the decomposition of water budget residuals and correction to datasets, rather than comparing the differences and rankings of closure residuals across different dataset combinations.

**CC1-10/ Line 167: This sentence should be in the past tense.**

**R/** Thanks for your comment. We have made the revisions as follows (Line 170-172 in tracked version):

In line with this objective, referring to the work of Petch et al. (2023), we strategically selected single product for each water component to construct water budget equation, thereby laying the foundation for further research.

**CC1-11/ Fig. 3: It is recommended to add further explanations in the caption of Fig. 3.**

**R/** Thank you for your valuable suggestion. We have added further explanations to the caption of Fig.3 as follows (Line 305-308 in tracked version):

**Figure 3.** Illustration of the correction process advancing convergence between the simulation and measurement systems. The measurement system is corrected to approach the simulation system, while the simulation system is refined via parameter calibration to better approximate the measurement system. As a result, the distance between the two systems is reduced, leading to better physical consistency in the corrected measurement system.

**CC1-12/ Line 458: I suggest emphasizing the spatial distribution of water balance closure.**

**R/** Thank you for your suggestion. Based on your advice, we have revised the sentence to: (Line 541-542 in tracked version):

Therefore, we speculate that the spatial distribution of water budget closure is predominantly influenced by the characteristics of the basin.

**CC1-13/ Line 619: A "." is missing before the "The major".**

**R/** Thank you for pointing out this oversight. We have added a period between the sentences. (Line 803-804 in tracked version).

**Thank you again for your thorough review and the valuable suggestions provided.**

**Finally, we would like to once again thank the Editor and all Reviewers for their thorough review of our paper. If there are any questions, suggestions, or discussions, please feel free to contact us.**

---

## Author Response (AR2)

**Response to comments on Manuscript hess-2024-230**

**Title**: Achieving water budget closure through physical hydrological processes modelling: insights from a large-sample study

**Authors:** Xudong Zheng, Dengfeng Liu*, Shengzhi Huang*, Hao Wang, Xianmeng Meng

**Manuscript ID**: hess-2024-230

Dear Editor and Reviewers,

Please find enclosed our responses to the manuscript assessment entitled "Achieving water budget closure through physical hydrological processes modelling: insights from a large-sample study".

We would like to express our sincere gratitude to the Editor, the two anonymous reviewers for their invaluable support and constructive suggestions, as well as for the opportunity afforded to us to revise our work. We have given full attention to all comments and suggestions and made all revisions accordingly. It has resulted in an improved manuscript that fully addresses all concerns.

Concerning the revision of the manuscript, the following changes have been made:

(1) The correction of Eq. 7 and the redrawing of Figs. 10 and 15 to enhance readability.
(2) Section. 5.2.1 has been updated to highlight the impact of forcing uncertainty and potential solutions.
(3) Section 5.2.2 has been updated to emphasize the defects of HBV model.
(4) Several spelling errors and grammatical issues have been corrected.

We believe the paper quality has significantly improved through this review process. We are also happy to address any comments that may further strengthen the paper quality. We are uploading our point-by-point response to the comments, an updated manuscript with red highlighting indicating changes, and a clean updated manuscript without highlights.

In the point-by-point responses below, the original comments are displayed in **bold**. **EC-n/**, **RC3-n/** correspond to the Editor, Referee 3, respectively. The corresponding responses begin with **R/** and the revisions is highlighted in red, while important sections are marked in blue. The quoted content is displayed in *italics*.

We thank you for your consideration,

Dengfeng Liu
Email: liudf@xaut.edu.cn

**Responses to Editor:**

**EC-1/ Publish subject to minor revisions (review by editor). Please address the minor comments from referee #3.**

**R/** We greatly appreciate your timely handling of our manuscript, as well as the recognition of our work and the opportunity to publish after minor revisions. All comments from referee #3 have been addressed point-by-point below. The updated manuscript, which includes a tracked version with changes highlighted in red and a clean version without highlights, will be submitted alongside this response file.

We believe that our manuscript has significantly improved through this review process, and we are open to addressing any comments that may further enhance the quality of our paper. If there are any questions or suggestions, please feel free to contact us.

**Responses to RC3:**

We sincerely appreciate your time and effort in reviewing our manuscript, and providing valuable suggestions. All comments from RC3 are addressed below with point-by-point responses.

**RC3-1/ This is a very interesting and valuable study, but it has to be said that it faces great challenges. I'm trying to understand the whole study, and some confusion and doubts need to be further explained based on the previous reviewers.**

**R/** Thanks for your positive feedback and recognition of our work. It must be said that, as you pointed out, the goal of this work is indeed challenging. In the era of big data, we believe that effectively integrating our knowledge (water balance) and real-world observation (measurements) holds significant value. This is the primary motivation behind this work. While it may still have some limitations, we are confident that by addressing the issues you kindly raised and making the necessary revisions, it can become a meaningful contribution to the field.

Your comments are valuable for revising and improving our paper. Below, we provide detailed responses to each of your concerns. The corresponding revisions are attached after the responses and have been incorporated into the revised manuscript.

**RC3-2/ First of all, the essential purpose of this study is to try to correct the errors of basic data through the principle of water balance using the HBV model, to reduce the balance residual of water in closed basins. The two residual hypotheses seem reasonable, but I found that data correction mainly targets sink terms other than precipitation which is the source term in water balance. Generally speaking, the errors that are difficult to overcome in hydrological model simulation are mainly from precipitation rather than others such as ET, SWE, etc. According to equations 3 and 4, inconsistency residuals of precipitation are attributed to omission residuals, which is also discussed in section 5.2.1. However, Figure 5 shows that the omission residual does not seem to dominate compared to the inconsistency residual. Does this indicate that TRMM data is working well across the COUNS? In my opinion, its inconsistency residual is largely offset by the correction of other data. Moreover, the corrected inconsistency residuals in this study are essentially caused by the inconsistencies of precipitation.**

**R/** Excellent point! Clearly, you have have gained a profound understanding of the scientific question addressed in this work. We are sincerely grateful for your thoughtful analysis.

The first point we acknowledge is that this framework does not explicitly address the residuals in the precipitation products, which have been considered one of the main sources of residuals in many previous studies (Sahoo et al., 2011; Ansari et al., 2022). As a primary source term and the main forcing of hydrological models, the absence of precipitation correction appears counterintuitive. This was also mentioned in a previous community comment:

*CC1-2/ There are PTRMM in both Eq. (5) and (6), then how do we reduce the inconsistency residuals brought by P in the water budget?*

However, in our opinion, assuming the uncertainty (residuals) in precipitation product can be neglected

is both necessary and feasible. This is also why we explicitly emphasized this assumption at the beginning of the method section. From a necessity perspective, this assumption is made to ensure the operation of the correction framework. Since the current setup does not couple with an atmospheric model, and precipitation, as a forcing, cannot be simulated, it leaves us without a reference within a physically consistent simulation system. From a feasibility perspective, the reason this assumption holds due to the model's inherent calibration compensation capability. Specifically, the calibration compensation capability allows the model to accommodate uncertainties in the forcing (namely, precipitation) through calibration, producing relatively reliable simulations, as mentioned in the introduction (Line 95-100 in the tracked version):

*Another distinctive feature of hydrological models, known as error adaptability or calibration compensation capability, underscores their pivotal role as innovative solutions for addressing challenges in achieving water budget closure. The feature emphasizes that hydrological models can, to some extent, compensate for biases in model inputs, outputs and structure, allowing satisfactory performance even when the utilized datasets exhibit certain inaccuracies (Wang et al., 2023). This provides hydrological models with the potential to integrate forcing and evaluation datasets into a unified water balance system under the soft constraint paradigm.*

This capability is further enhanced under the constraints of multi-objective calibration. As shown in Fig. C1, despite the potential uncertainty in precipitation, most basins still yield reliable simulation across all variables. All of our analyses exclude basins with unreliable simulations (i.e., the remaining 475 basins), thereby offering a certain level of confidence in this assumption. In other words, given the HBV model, TRMM performs well in these basins.

[Figure]

**Figure C1.** The multi-objective simulation performances of the HBV model across the CAMELS basins. Results are based on (a) runoff, (b) evaporation, (c) soil moisture storage and groundwater reservoir storage, and (d) snow water equivalent. Red dots represent unreliable simulation performance, and the size of points is proportional to the basin area. The unit of RMSE is "mm".

Therefore, under this assumption, the residual in precipitation included in Eq. 4 is eliminated through model calibration. In other words, the TRMM precipitation forms a physically consistent simulation system with the remaining simulated variables (Eq. 6), although it still contains uncertainty. In extreme case, when the measurement system is corrected to be identical to the simulation system, all measurements

would become physically consistent. The correction amount, which is the sum of the changes occurring in the measurements (without precipitation), corresponds to the inconsistency residuals. This process can be seen as a collapse from Eq. 5 to Eq. 6.

Returning to the uncertainty inherent in the precipitation products, it is undoubtedly always present. However, when different precipitation products are used, the correction process ensures that the final corrected results are both similar and physically consistent (Fig. 15). This is achieved by maintaining similar inconsistency residuals—corresponding to a similar correction amount—as long as differences in precipitation do not result in substantial variations in the hydrological processes. In general situation, the impact of the uncertainty in precipitation is mitigated through the calibration process, resulting in similar hydrological process. It is worth highlighting that the iterative calibration process (applying a correction rate less than 1) also plays a crucial role (see the response to RC3-4).

On the other hand, the omission residuals are dynamically adjusted. Since the precipitation serves as the source term in the water balance equation, it will lead to different overall water amounts. In the context of a relatively complete water balance equation, the impact of omission residuals is minimal, as their source is solely precipitation. In contrast, the inconsistency residuals arise from uncertainty in the other five terms (Eq. 5). This is why the overall residuals in Fig. 5 are predominantly driven by inconsistency residuals.

Nevertheless, the results in Sect. 5.2.1 suggest that, in this study, the impact of precipitation on the correction of other variables in minimal (likely because both precipitation products are relatively accurate). The difference in omission residuals can provide guidance for the assessment and correction of precipitation uncertainty, as discussed in this section (Line 721-728).

*Clues to this possibility are hidden in the above analysis. Systematic biases in precipitation products are directly reflected in the water budget equation, leading to different total input water volumes. Consequently, with the inconsistency residuals of other variables unchanged, maintaining the water balance would require an increase in omission residuals (Fig. 15e). Therefore, it can be inferred that, with other variables unchanged, TRMM demonstrates superior water budget closure compared to Daymet, which contains smaller inconsistency residuals. In other words, the difference in the two omission residuals reflects the discrepancy in inconsistency residuals contained within the two precipitation products. This portion of the omission residuals difference can be directly corrected in the precipitation.*

The inability to correct precipitation forcing is a major limitation of this framework. We discuss this issue in Sect. 5.2.1, and in response to your comment, we have added the following points based on our reflections.

Line 292-293 in the tracked version:
In extreme case, when the measurement system is corrected to be identical to the simulation system, all measurements would become physically consistent. This process can be seen as a collapse from Eq. 5 to Eq. 6.

Line 713-715 in the tracked version:
This is achieved by maintaining similar inconsistency residuals—corresponding to a similar correction amount—as long as differences in precipitation do not result in substantial variations in the hydrological processes.

Line 731-733 in the tracked version:
Another strategy is to couple an atmospheric model with this framework to generate simulated precipitation, allowing for the correction of precipitation products. In subsequent work, we will explore these approaches and try to extend the PHPM-MDCF based on these ideas.

**RC3-3/ In addition, there are some issues that need to be improved or resolved:**

**R/** Thank you for your careful review, which has been extremely valuable in enhancing the quality of our manuscript. We have addressed and made revisions for all of these issues, which are provided below.

**Specific comments**

**RC3-4/ "Here, an initial correction rate of 0.5 is set to gradually correct the multisource datasets, thereby avoiding potential uncertainties that arise from excessive correction." Why is there an initial correction? Isn't it calibrated according to the equation 7.**

**R/** Thank you for pointing out the discrepancies in our equations. It has been revised to (Line 274-277 in the tracked version):

$$M_c^v = M_o^v - Res_i \times \frac{d_v}{d_{all}} \times \alpha, \tag{7}$$

where $M_c^v$ is the corrected measurements of variable $v$, and $M_o^v$ is the original measurements; $d_v$ is the difference between simulation and measurement of variable $v$, and $d_{all}$ represents the aggregate of differences for all variables; $\alpha$ is the correction rate, with an initial value of 0.5.

In addition, it should be noted that setting a correction rate effectively helps mitigate uncertainty caused by over-correction. This correction rate can be adjusted based on the simulation results in correction step 3.

**RC3-5/ The study compared the effects of Daymet and TRMM. It seems that R is consistent, but E is very different (Figure 15). However, E is a corrected result and should not theoretically differ greatly. I'm not sure if this is due to the calibration method or the data itself.**

**R/** Thank you for pointing out the potential confusion in Fig. 15. In fact, evaporation shows high consistency between the two correction. In subplot Fig. 15a, the results of the two correction results are represented by the blue line and red dots, respectively, which may give the appearance of inconsistency. In addition, the evaporation points are hidden by other points in Fig. 15b, which further leads to a misunderstanding.

Recognizing this, we have modified Fig. 15b to position the evaporation points in the top layer, as shown below (Line 734-738 in tracked version). It can be seen that the scatter points from both corrections are

located near the 1:1 line, indicating that the results of the two evapotranspiration corrections are consistent.

[Figure]

**Figure 15**. Comparison of correction results based on different forcing datasets (TRMM and Daymet) at basin 1013500. (a-b) Corrected time series of five water budget variables. (c-e) Variation of long-term mean absolute values of three residuals with correction iterations at the monthly scale. The unit of residuals is "mm".

**RC3-6/ Section 5.2.2 mentions the uncertainty of the model structure, but does not cover the defects of HBV. HBV has some shortcomings in both snowmelt simulation and groundwater simulation.**

**R/** Thank you for your suggestion. According to your suggestion, we have included the shortcoming of the HBV model in Sect. 5.2.2 of the revised manuscript (Line 775-778 in the tracked version). For your convenience and review, we present the revised content below.

It is worth noting that, while we have validated the reliability of the HBV model in the current study, its simplistic physics and lumped design structure lead to significant limitations in simulating several processes such as snow and groundwater (Brunner et al., 2021). In other words, the HBV model may not be suitable for accurately representing the reality of these specific processes.

**RC3-7/ The spatial resolution of the gridded data used in this study is very rough, and some of the basins selected are very small. It will cause great uncertainty if the basic data is not reliable.**

**R/** Thank you for your comment. We agree with this point; the disparity between scale of data and that of research object is an important source of uncertainty. This may be the potential cause of the scale effect in the residuals discussed in Sect. 4.4.1 (Fig. 12, we present it below).

[Figure]

**Figure 12.** Relationship between the mean absolute of water budget residuals, basin area, long-term average daily precipitation, and runoff coefficient (RC) over 475 CAMELS basins with reliable simulations. The respective red lines represent the linear regression of residuals with basin area for each timescale.

In this study, to establish the water balance equation and drive the hydrological model at the daily scale, while ensuring a certain level of data reliability, we conducted a data selection process from a broad range of available datasets. With reference to previous studies, we ultimately selected the datasets presented in Table. 1. While this does introduce some uncertainty, it is a decision made by balancing data availability with the research objective.

Nevertheless, we believe that the uncertainty introduced by scaling process remains limited within a certain range. This is for the following two reasons. First, all datasets are aggregated to the basin scale in depth units. This is to say, the grid volume data are divided by the grid area, which helps reduce the impact of the mismatch between the grid and basin boundaries, even for very small basins. Second, as we mentioned in the response to RC3-2, the hydrological model's calibration compensation capability further reduces the impact of the mismatch. Since the lumped hydrological model do not account for spatial heterogeneity, the impact of grid distribution is minimal. These factors significantly reduce the uncertainty caused by scale differences. The accurate simulation of runoff at basin outlet support this, as it effectively represents the basin scale, while the forcing is at the grid scale (Fig. C1).

Building on the discussion of this issue, we have added the following points in the discussion section (i.e., Sect. 5.2.1) to highlight the potential uncertainty arising from scale mismatch.

Line 685-687 in the tracked version:

First, the uncertainty in the forcing may arise from two aspects, one is the inaccuracy of the datasets themselves, and the other is the uncertainty introduced by the scaling process (i.e., the conversion from grid scale to basin scale).

Line 699-700 in the tracked version:
Thirdly, the uncertainty caused by the mismatch between the grids and basin boundaries is effectively alleviated through the unit conversion (i.e., from volume to depth units).

**Reference**

Sahoo, A., M. Pan, T. Troy, R. Vinukollu, J. Sheffield, and E. Wood (2011), Reconciling the global terrestrial water budget using satellite remote sensing, Remote Sensing of Environment, 115, 1850-1865.

Ansari, R., M. U. Liaqat, and G. Grossi (2022), Evaluation of gridded datasets for terrestrial water budget assessment in the Upper Jhelum River Basin-South Asia, Journal of Hydrology, 613, 128294.

**Finally, we would like to once again thank the Editor and all the Reviewers for their thorough review and support of our paper. If you have any questions, suggestions, or discussions, please feel free to contact us.**